# Pre-processing visualization of hyperspectral fluorescent data with Spectrally Encoded Enhanced Representations

Wen Shi [1,2,8], Daniel E.S. Koo[1,2,8], Masahiro Kitano[1,3], Hsiao J. Chiang[1,2], Le A. Trinh[1,3,4], Gianluca Turcatel[5,6], Benjamin Steventon[7], Cosimo Arnesano[1,3], David Warburton[5,6], Scott E. Fraser [1,2,3] & Francesco Cutrale [1,2,3]*

Hyperspectral fluorescence imaging is gaining popularity for it enables multiplexing of spatio-temporal dynamics across scales for molecules, cells and tissues with multiple fluorescent labels. This is made possible by adding the dimension of wavelength to the dataset. The resulting datasets are high in information density and often require lengthy analyses to separate the overlapping fluorescent spectra. Understanding and visualizing these large multi-dimensional datasets during acquisition and pre-processing can be challenging. Here we present Spectrally Encoded Enhanced Representations (SEER), an approach for improved and computationally efficient simultaneous color visualization of multiple spectral components of hyperspectral fluorescence images. Exploiting the mathematical properties of the phasor method, we transform the wavelength space into information-rich color maps for RGB display visualization. We present multiple biological fluorescent samples and highlight SEER's enhancement of specific and subtle spectral differences, providing a fast, intuitive and mathematical way to interpret hyperspectral images during collection, pre-processing and analysis.

[1] Translational Imaging Center, University of Southern California, 1002 West Childs Way, Los Angeles, CA 90089, USA. [2] Biomedical Engineering, University of Southern California, 1002 West Childs Way, Los Angeles, CA 90089, USA. [3] Molecular and Computational Biology, University of Southern California, 1002 West Childs Way, Los Angeles, CA 90089, USA. [4] Department of Biological Sciences, University of Southern California, 1050 Childs Way, Los Angeles, CA 90089, USA. [5] Developmental Biology and Regenerative Medicine Program, Saban Research Institute, Children's Hospital, 4661 Sunset Blvd, Los Angeles, CA 90089, USA. [6] Keck School of Medicine and Ostrow School of Dentistry, University of Southern California, Los Angeles, CA, USA. [7] Department of Genetics, University of Cambridge, Downing Street, Cambridge CB2 3EH, UK. [8] These authors contributed equally: Wen Shi, Daniel E.S. Koo.
*email: cutrale@usc.edu

Fluorescence hyperspectral imaging (fHSI) has become increasingly popular in recent years for the simultaneous imaging of multiple endogenous and exogenous labels in biological samples[1–5]. Among the advantages of using multiple fluorophores is the capability to simultaneously follow differently labeled molecules, cells or tissues space- and time-wise. This is especially important in the field of biology where tissues, proteins and their functions within organisms are deeply intertwined, and there remain numerous unanswered questions regarding the relationship between individual components[6]. fHSI empowers scientists with a more complete insight into biological systems with multiplexed information deriving from observation of the full spectrum for each point in the image[7].

Standard optical multi-channel fluorescence imaging differentiates fluorescent protein reporters through band-pass emission filters, selectively collecting signals based on wavelength. Spectral overlap between labels limits the number of fluorescent reporters that can be acquired and background signals are difficult to separate. fHSI overcomes these limitations, enabling separation of fluorescent proteins with overlapping spectra from the endogenous fluorescent contribution, expanding to a fluorescent palette that counts dozens of different labels with corresponding separate spectra[7–9].

The drawback of acquiring this vast multidimensional spectral information is an increase in complexity and computational time for the analysis, showing meaningful results only after lengthy calculations. To optimize experimental time, it is advantageous to perform an informed visualization of the spectral data during acquisition, especially for lengthy time-lapse recordings, and prior to performing analysis. Such preprocessing visualization allows scientists to evaluate image collection parameters within the experimental pipeline as well as to choose the most appropriate processing method. However, the challenge is to rapidly visualize subtle spectral differences with a set of three colors, compatible with displays and human eyes, while minimizing loss of information. As the most common color model for displays is RGB, where red, green, and blue are combined to reproduce a broad array of colors, hyper- or multispectral datasets are typically reduced to three channels to be visualized. Thus, spectral information compression becomes the critical step for proper display of image information.

Dimensional reduction strategies are commonly used to represent multidimensional fHSI data[10]. One strategy is to construct fixed spectral envelopes from the first three components produced by principal component analysis (PCA) or independent component analysis (ICA), converting a hyperspectral image to a three-band visualization[10–14]. The main advantage of spectrally weighted envelopes is that it can preserve the human-eye perception of the hyperspectral images. Each spectrum is displayed with the most similar hue and saturation for tri-stimulus displays in order for the human eye to easily recognize details in the image[15]. Another popular visualization technique is pixel-based image fusion, which preserves the spectral pairwise distances for the fused image in comparison to the input data[16]. It selects the weights by evaluating the saliency of the measured pixel with respect to its relative spatial neighborhood distance. These weights can be further optimized by implementing widely applied mathematical techniques, such as Bayesian inference[17], by using a filters-bank[18] for feature extraction[19] or by noise smoothing[20].

A drawback to approaches such as Singular Value Decomposition to compute PCA bases and coefficients or generating the best fusion weights is that it can take numerous iterations for convergence[21]. Considering that fHSI datasets easily exceed GigaBytes range and many cross the TeraBytes threshold[6], such calculations will be both computationally and time demanding. Furthermore, most visualization approaches have focused more on interpreting spectra as RGB colors and not on exploiting the full characterization that can be extracted from the spectral data. Our approach is based on the belief that preserving most spectral information and enhancing the distinction of spectral properties between relevant pixels, will provide an ideal platform for understanding biological systems. The challenge is to develop tools that allow efficient visualization of multidimensional datasets without the need for computationally demanding dimensionality reduction, such as ICA, prior to analysis.

In this work, we build maps based on Phasors (Phase Vectors). The Phasor approach to fluorescence microscopy has multiple advantages deriving from its properties for fluorescent signals[22–25]. After transforming the spectrum at each pixel into its Fourier components, the resulting complex value is represented as a two-dimensional histogram where the axes represent the real and imaginary components. Such histogram has the advantage of providing a representative display of the statistics and distributions of pixels in the image from a spectral perspective, simplifying identification of independent fluorophores. Pixels in the image with similar spectra generate a cluster on the phasor plot. While this representation is cumulative across the entire image, each single point on the phasor plot is easily remapped to the original fluorescent image[26–29].

Exploiting the advantages of the phasor approach, Hyper-Spectral Phasors (HySP) has enabled analysis of 5D hyperspectral time-lapse data semi-automatically as similarly colored regions cluster on the phasor plot. These clusters have been characterized and exploited for simplifying interpretation and spatially lossless denoising of data, improving both collection and analysis in low-signal conditions[29]. Phasor analysis generally explores the 2d histogram of spectral fingerprints by means of geometrical selectors[22,23,25,27,28,30], which is an effective strategy but requires user involvement. While capable of imaging multiple labels and separating different spectral contributions as clusters, this approach is inherently limited in the number of labels that can be analyzed and displayed simultaneously. Prior works directly utilize phase and modulation for quantifying, categorizing, and representing features within Fluorescence Lifetime and Image Correlation Spectroscopy data[31–33]. Our method differs from previous implementations[22,23,25,27,28,30], as it focuses instead on providing a mathematically constructed, holistic preprocessing visualization of large hyperspectral data.

The solution we propose extracts from both the whole denoised phasor plot and image to reconstruct a one shot view of the data and its intrinsic spectral information. Spectrally Encoded Enhanced Representations (SEER) is a dimensionality reduction-based approach, achieved by utilizing phasors, and automatically creating spectrally representative color maps. The results of SEER show an enhanced visualization of spectral properties, representing distinct fluorophores with distinguishable pseudo-colors and mathematically highlighted differences between intrinsic signals during live imaging. SEER has the potential of optimizing the experimental pipeline, from data collection during acquisition to data analysis, greatly improving image quality and data size.

## Results

**Spectrally Encoded Enhanced Representations (SEER).** The execution of SEER has a simple foundation. Each spectrum is assigned a pseudo-color, based on its real and imaginary Fourier components, by means of a reference color map.

This concept is illustrated in detail in Fig. 1 using an example of Zebrabow[34] embryo dataset, where cells within the sample express different ratios of cyan, yellow, and red fluorescent proteins, resulting in a wide-ranging pallet of discrete spectral differences. The data are acquired as a hyperspectral volume ($x$, $y$,

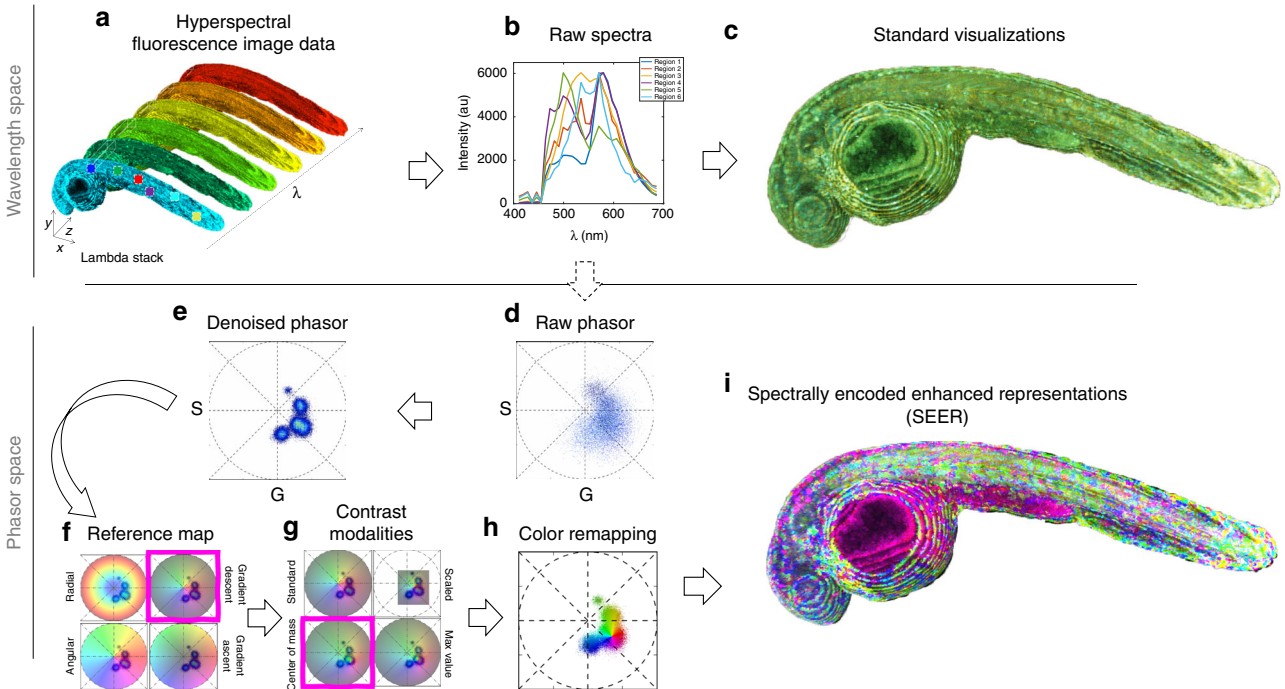

**Fig. 1 Spectrally Encoded Enhanced Representations (SEER) conceptual representation. a** A multispectral fluorescent dataset is acquired using a confocal instrument in spectral mode (32 channels). Here we show a Tg(ubi:Zebrabow)[34] dataset where cells contain a stochastic combination of cyan, yellow, and red fluorescent proteins. **b** Average spectra within six regions of interest (colored boxes in **a**) show the level of overlap resulting in the sample. **c** Standard multispectral visualization approaches have limited contrast for spectrally similar fluorescence. **d** Spectra for each voxel within the dataset are represented as a two-dimensional histogram of their Sine and Cosine Fourier coefficients S and G, known as the phasor plot. **e** Spatially lossless spectral denoising is performed in phasor space to improve signal[29]. **f** SEER provides a choice of several color reference maps that encode positions on the phasor into predetermined color palettes. The reference map used here (magenta selection) is designed to enhance smaller spectral differences in the dataset. **g** Multiple contrast modalities allow for improved visualization of data based on the phasor spectra distribution, focusing the reference map on the most frequent spectrum, on the statistical spectral center of mass of the data (magenta selection), or scaling the map to the distribution. **h** Color is assigned to the image utilizing the chosen SEER reference map and contrast modality. **i** Nearly indistinguishable spectra are depicted with improved contrast, while more separated spectra are still rendered distinctly.

$z$, $\lambda$) (Fig. 1a), providing a spectrum for each voxel. The spectra obtained from multiple regions of interest are complex, showing both significant overlap and the expected difference in ratios (Fig. 1b). Discriminating the very similar spectra within the original acquisition space is challenging using standard multispectral dataset visualization approaches (Fig. 1c).

SEER was designed to create usable spectral contrast within the image by accomplishing five main steps. First, the Sine and Cosine Fourier transforms of the spectral dataset at one harmonic (usually 1st or 2nd owing to Riemann surfaces) provide the components for a 2D phasor plot (Fig. 1d). The phasor transformation compresses and normalizes the image information, reducing a multidimensional dataset into a 2D-histogram representation and normalizing it to the unit circle.

Second, the histogram representation of the phasor plot provides insight on the spectral population distribution and improvement of the signal through summation of spectra in the histogram bins. Pixels with very similar spectral features, for example expressing only a single fluorophore, will fall within the same bin in the phasor plot histogram. Because of the linear property of the phasor transform, if an image pixel contains a mixture of two fluorophores, its position on the phasor plot will lie proportionally along the line connecting the phasor coordinates of those two components. This step highlights importance of geometry and distribution of bins in the phasor representation.

Third, spatially lossless spectral denoising, previously presented[29], is performed 1–2 times in phasor space to reduce spectral error. In short, median filters are applied on both the Sine

and Cosine transformed images, reducing the spectral scatter error on the phasor plot, while maintaining the coordinates of the spectra in the original image (Fig. 1e). Filters affect only the phasor space, producing an improvement of the signal.

Fourth, we designed multiple SEER maps exploiting the geometry of phasors. For each bin, we assign RGB colors based on the phasor position in combination with a reference map (Fig. 1f). Subtle spectral variations can be further enhanced with multiple contrast modalities, focusing the map on the most frequent spectrum, the statistical center of mass of the distribution or scaling the colors to the extremes of the phasor distribution (Fig. 1g).

Finally, colors in the original dataset are remapped based on SEER results (Fig. 1h). This permits a dataset in which spectra are visually indistinguishable (Fig. 1a–c) to be rendered so that even these subtle spectral differences become readily discernible (Fig. 1i). SEER rapidly produces three channel color images (Supplementary Fig. 1) that approximate the visualization resulting from a more complete spectral unmixing analysis (Supplementary Fig. 2).

**Standard reference maps.** Biological samples can include a multitude of fluorescent spectral components, deriving from fluorescent labels as well as intrinsic signals, each with different characteristics and properties. Identifying and rendering these subtle spectral differences is the challenge. We found that no one rendering is sufficient for all cases, and thus created four

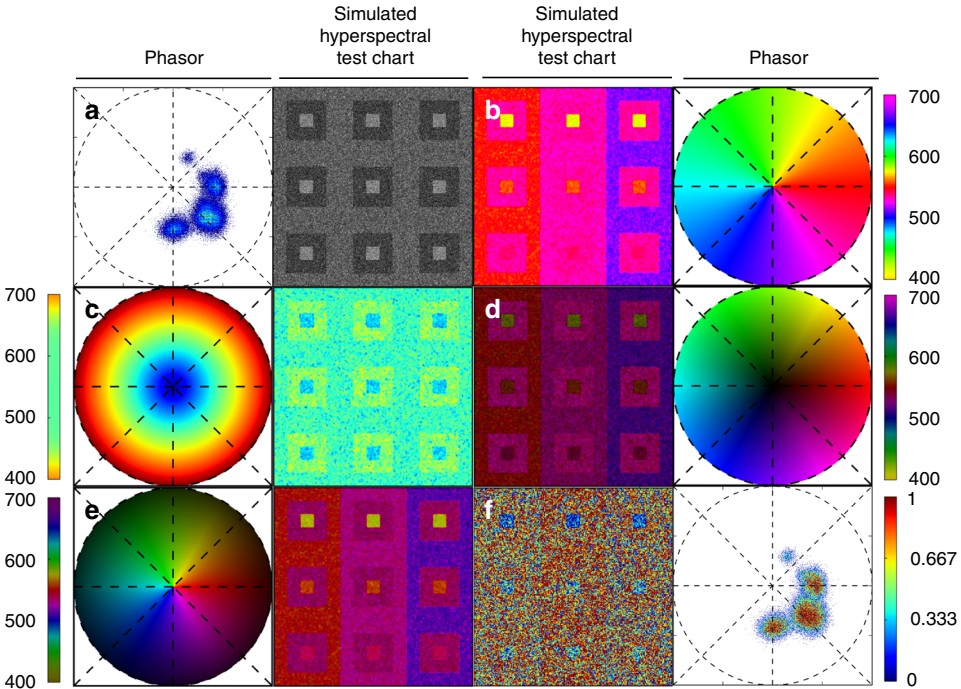

**Fig. 2 Spectrally Encoded Enhanced Representation (SEER) designs.** A set of standard reference maps and their corresponding result on a Simulated Hyperspectral Test Chart (SHTC) designed to provide a gradient of spectral overlaps between spectra. **a** The Standard phasor plot with corresponding average grayscale image provides the positional information of the spectra on the phasor plot. The phasor position is associated to a color in the rendering according to a set of standard reference maps, each highlighting a different property of the dataset. **b** The angular map enhances spectral phase differences by linking color to changes in angle (in this case, with respect to origin). This map enhances changes in maximum emission wavelength, as phase position in the plot is most sensitive to this feature, and largely agnostic to changes in intensity. **c** The radial map, instead, focuses mainly on intensity changes, as a decrease in the signal-to-noise generally results in shifts towards the origin on the phasor plot. As a result, this map highlights spectral amplitude and magnitude, and is mostly insensitive to wavelength changes for the same spectrum. **d** The gradient ascent map enhances spectral differences, especially within the higher intensity regions in the specimen. This combination is achieved by adding a brightness component to the color palette. Darker hues are localized in the center of the map, where lower image intensities are plotted. **e** The gradient descent map improves the rendering of subtle differences in wavelength. Colorbars for **b**, **c**, **d**, **e** represent the main wavelength associated to one color in nanometers. **f** The tensor map provides insights in statistical changes of spectral populations in the image. This visualization acts as a spectral edge detection on the image and can simplify identification of spectrally different and infrequent areas of the sample such as the center of the SHTC. Colorbar represents the normalized relative gradient of counts.

specialized color map references to enhance color contrast in samples with different spectral characteristics. To simplify the testing of the color map references, we designed a Simulated Hyperspectral Test Chart (SHTC), in which defined areas contain the spectra we obtained from CFP, YFP, and RFP zebrafish embryos. Each section of the test chart offers different image contrast, obtained by shifting the CFP and RFP spectra maxima position with respect to the YFP spectrum (Supplementary Fig. 3, see Methods). We render the SHTC in grayscale image and with SEER for comparison (Fig. 2a). These representations can be rapidly shown separately to determine which has the highest information content.

A reference map is defined as an organization of the palette where each spectrum is associated with a color based on its phasor position. The color distribution in each of the reference maps is a function of the coordinates of the phasor plot. In the angular map (Fig. 2b), hue is calculated as a function of angle, enhancing diversity in colors when spectra have different center wavelengths (phases) on the phasor plot. For the radial map (Fig. 2c), we assign colors with respect to different radii, highlighting spectral amplitude and magnitude. The radial position is, in general, related to the intensity integral of the spectrum, which in turn can depend on the shape of the spectrum, with the null-intensity localizing at the origin of the plot (Supplementary Fig. 4). In our simulation (Fig. 2c), the colors obtained with this map mainly represent differences in shape, however, in a scenario with large

dynamic range of intensities, colors will mainly reflect changes in intensity, becoming affected, at low signal-to-noise, by the uncorrelated background (Supplementary Fig. 5). In the gradient ascent and descent models (Fig. 2d, e), the color groups differ according to angle as seen in the angular map with an added variation of the color intensity strength in association with changes in the radius. Gradient maps enhance similar properties as the angular map. However, the gradient ascent (Fig. 2d) map puts a greater focus on distinguishing the higher intensity spectra while de-emphasizing low intensity spectra; whereas, the gradient descent (Fig. 2e) map does the opposite, highlighting the spectral differences in signals with low intensity. The complementary attributes of these four maps permits renderings that distinguish a wide range of spectral properties in relation to the phasor positions. It is important to note that the idea of Angular and Radial maps have been previously utilized in a variety of applications and approaches[32,33] and are usually introduced as Phase and Modulation, respectively. Here, we have recreated and provided these maps for our hyperspectral fluorescence data as simpler alternatives to our more adaptable maps.

The standard reference maps simplify comparisons between multiple fluorescently labeled specimens as the palette representation is unchanged across samples. These references are centered at the origin of the phasor plot, hence their color distributions remain constant, associating a predetermined color to each phasor coordinate. Fluorophores positions are constant on the

phasor plot[29], unless their spectra are altered by experimental conditions. The ability of the standard reference maps to capture either different ratios of labels or changes in a label, such as a calcium indicator, offers the dual advantage of providing a rapid, mathematically improved overview and simplifying the comparison between multiple samples.

**Tensor map**. The SEER approach provides a straightforward means to assess statistical observations within spectral images. In addition to the four standard reference maps, we designed a tensor map that recolors each image pixel based on the gradient of counts relative to its surrounding spectra (Fig. 2f). Considering that the phasor plot representation is a two-dimensional histogram of real and imaginary Fourier components, then the magnitude of each histogram bin is the number of occurrences of a particular spectrum. The tensor map is calculated as a gradient of counts between adjacent bins, and each resulting value is associated a color based on a color map (here we use a jet color map).

The image is recolored according to changes in spectral occurrences, enhancing the spectral statistical fluctuations for each phasor cluster. The change in frequency of appearance can provide insights in the dynamics of the population of spectra inside dataset. A visible result is a type of spectral edge detection that works in the wavelength dimension, facilitating detection of chromatic changes in the sample. Alternatively, tensor map can aid in identifying regions which contain less frequent spectral signatures relative to the rest of the sample. An example of such a case is shown in the upper left quadrant of the simulation (Fig. 2f) where the center part of each quadrant has different spectrum and appears with lower frequency compared with its surroundings.

**Modes (scale and morph)**. We have implemented two different methods to improve our ability to enhance spectral properties: scaled mode and morphed mode.

Scaled mode provides an adaptive map with increased color contrast by normalizing the standard reference map extreme values to the maximum and minimum phasor coordinates of the current dataset, effectively creating the smallest bounding unit circle that contains all phasor points (Fig. 3b). This approach maximizes the number of hues represented in the rendering by resizing the color map based on the spectral range within the image. Scaled mode increases the difference in hue and the contrast of the final false-color rendering. These characteristics set the scaled mode apart from the standard reference maps (Fig. 3a), which constantly cover the full phasor plot area and eases comparisons between datasets. Scaled mode sacrifices this uniformity, but offers spectral contrast stretching that improves contrast depending on the values represented in individual image datasets. The boundaries of the scaled mode can be set to a constant value across different samples to facilitate comparison.

Morph mode exploits the dataset population properties captured in the image's phasor representation to enhance contrast. From the phasor histogram, either the most frequent spectral signature or the center of mass (in terms of histogram counts) are used as the new center reference point of the SEER maps. We call this new calculated center the apex of the SEER. The result is an adaptive palette that changes depending on the dataset. In this representation mode, the edges of the reference map are held anchored to the phasor plot circular boundary, while the center point is shifted, and the interior colors are linearly warped (Fig. 3c, d). By shifting the apex, contrast is enhanced for datasets with off-centered phasor clusters. A full list of the combination of standard reference maps and modes is

reported (Supplementary Figs. 6, 7) for different levels of spectral overlap in the simulations and for different harmonics. The supplement presents results for SHTC with very similar spectra (Supplementary Fig. 3), using second harmonic in the transform (see Methods), and for an image with frequently encountered level of overlap (Supplementary Figs. 8 and 9) using first harmonic. In both scenarios, SEER improves visualization of multispectral datasets (Supplementary Figs. 6, 7, 9, 10) compared with standard approaches (Supplementary Figs. 3 and 8). A detailed description of the choice of harmonic for visualization is presented in Supplementary Note 1. Implementation of 1x to 5x spectral denoising filters[29] further enhances visualization (Supplementary Figs. 11, 12).

**Color maps enhance different spectral gradients**. To demonstrate the utility of SEER and its modes, we present four example visualizations of images taken from unlabeled mouse tissues and fluorescently tagged zebrafish.

In live samples, a number of intrinsic molecules are known to emit fluorescence, including NADH, riboflavin, retinoids, and folic acid[36]. The contribution of these generally low signal-to-noise intrinsic signals to the overall fluorescence is generally called autofluorescence[7]. Hyperspectral imaging and HySP[29] can be employed to diminish the contribution of autofluorescence to the image. The improved sensitivity of the phasor, however, enables autofluorescence to become a signal of interest and allows for exploration of its multiple endogenous molecules contributions. SEER is applied here for visualizing multispectral autofluorescent data of an explant of freshly isolated trachea from a wild-type C57Bl mouse. The tracheal epithelium is characterized by a very low cellular turnover, and therefore the overall metabolic activity is attributable to the cellular function of the specific cell type. Club and ciliated cells are localized in the apical side of the epithelium and are the most metabolically active as they secrete cytokines and chemically and physically remove inhaled toxins and particles from the tracheal lumen. Contrarily, basal cells which represent the adult stem cells in the upper airways, are quiescent and metabolically inactive[37,38]. Because of this dichotomy in activity, the tracheal epithelium at homeostasis constituted the ideal cellular system for testing SEER and validating with FLIM imaging. The slight bend on the trachea, caused by the cartilage rings, allowed us to visualize the mesenchymal collagen layer, the basal and apical epithelial cells and tracheal lumen in a single focal plane.

The explant was imaged with 2-photon laser scanning microscopy in multispectral mode. We compare the state of the art true-color image (Fig. 4a, see Methods), and SEER images (Fig. 4b, c). The gradient descent morphed map (Fig. 4b) enhances the visualization of metabolic activities within the tracheal specimen, showing different metabolic states when moving from the tracheal airway apical surface toward the more basal cells and the underlying collagen fibers (Fig. 4b). The visualization improvement is maintained against different implementations of RGB visualization (Supplementary Fig. 13) and at different depths in volumetric datasets (Supplementary Fig. 28). The tensor map increases the contrast of cell boundaries (Fig. 4c). Changes in autofluorescence inside live samples are associated to variations in the ratio of NAD+/NADH, which in turn is related to the ratio of free to protein bound NADH[39]. Despite very similar fluorescence emission spectra, these two forms of NADH are characterized by different decay times (0.4 ns free and 1.0–3.4 ns bound)[35,40–44]. FLIM provides a sensitive measurement for the redox states of NADH and glycolytic/oxidative phosphorylation. Metabolic imaging by FLIM is well established and has been applied for characterizing disease

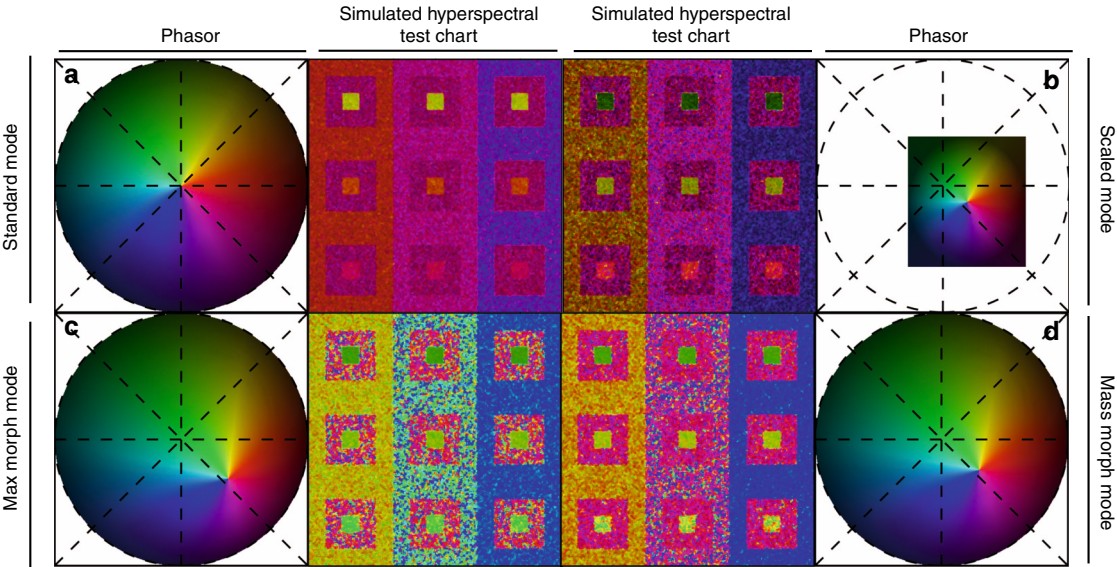

**Fig. 3 Enhanced contrast modalities.** For each SEER standard reference map design, four different modes can provide improved contrast during visualization. As a reference we use the gradient descent map applied on a Simulated Hyperspectral Test Chart (SHTC). **a** Standard mode is the standard map reference. It covers the entire phasor plot circle, centering on the origin and anchoring on the circumference. The color palette is constant across samples, simplifying spectral comparisons between datasets. **b** Scaled mode adapts the gradient descent map range to the values of the dataset, effectively performing a linear contrast stretching. In this process the extremities of the map are scaled to wrap around the phasor representation of the viewed dataset, resulting in the largest shift in the color palette for the phase and modulation range in a dataset. **c** Max morph mode shifts the map center to the maximum of the phasor histogram. The boundaries of the reference map are kept anchored to the phasor circle, while the colors inside the plot are warped. The maximum of the phasor plot represents the most frequent spectrum in the dataset. This visualization modality remaps the color palette with respect to the most recurring spectrum, allowing insights on the distribution of spectra inside the sample. **d** Mass morph mode, instead, uses the histogram counts to calculate a weighted average of the phasor coordinates and uses this color-frequency center of mass as a new center for the SEER map. The color palette now maximizes the palette color differences between spectra in the sample.

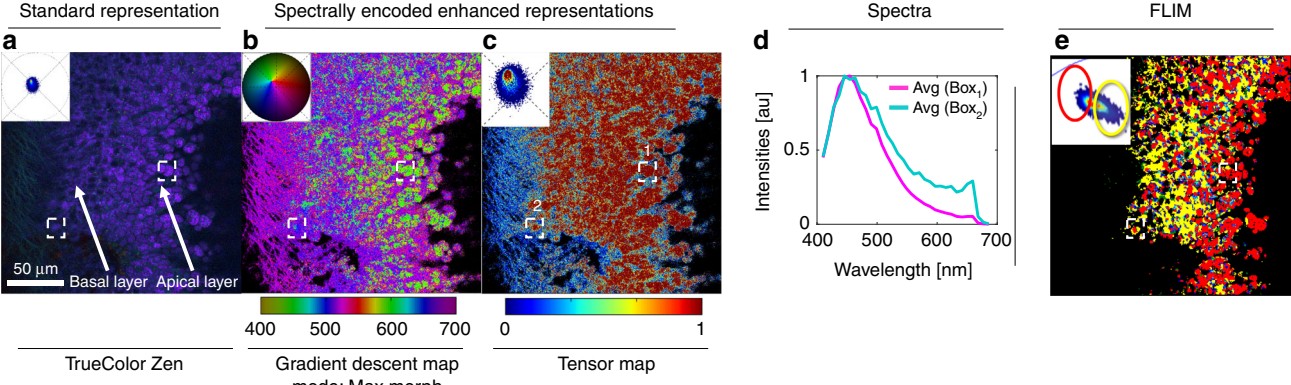

**Fig. 4 Autofluorescence visualization comparison for unlabeled freshly isolated mouse tracheal explant.** The sample was imaged using multispectral two-photon microscopy (740 nm excitation, 32 wavelength bins, 8.9 nm bandwidth, 410–695 nm detection) to collect the fluorescence of intrinsic molecules including folic acid, retinoids and NADH in its free and bound states. These intrinsic molecules have been used as reporters for metabolic activity in tissues by measuring their fluorescence lifetime, instead of wavelength, due to their closely overlapping emission spectra. This overlap increases the difficulty in distinguishing spectral changes when utilizing a (**a**) TrueColor image display (Zen Software, Zeiss, Germany). **b** The gradient descent morphed map shows differences between apical and basal layers, suggesting different metabolic activities of cells based on the distance from the tracheal airway. Cells on the apical and basal layer (dashed boxes) are rendered with distinct color groups. Colorbar represents the main wavelength associated to one color in nanometers. **c** The tensor map image provides an insight of statistics in the spectral dataset, associating image pixels' colors with corresponding gradient of phasor counts for pixels with similar spectra. The spectral counts gradients in this sample highlights the presence of fibers and edges of single cells. Colorbar represents the normalized relative gradient of counts. **d** Average spectra for the cells in dashed boxes (1 and 2 in panel **c**) show a blue spectral shift in the direction of the apical layer. **e** Fluorescence Lifetime Image Microscopy (FLIM) of the sample, acquired using a frequency domain detector validates the interpretation from panel (**b**), Gradient descent map, where cells in the apical layer exhibit a more oxidative phosphorylation phenotype (longer lifetime in red) compared with cells in the basal layer (shorter lifetime in yellow) with a more glycolytic phenotype. The selections correspond to areas selected in phasor FLIM analysis (**e**, top left inset, red and yellow selections) based on the relative phasor coordinates of NAD +/NADH lifetimes[35].

progression in multiple animal models, in single cells and in human as well as to distinguish stem cells differentiation and embryo development[41–48]. Previous work has shown that both Hyperspectral Imaging and FLIM correlate with metabolic changes in cells from retinal organoids[49]. Here, the dashed squares highlight cells with distinct spectral representation through SEER, a difference which the FLIM image (Fig. 4d, Supplementary Fig. 14) confirms.

**The improvement of SEER in visualizing intrinsic signals is clear when compared with standard approaches.** Microscopic imaging of fluorophores in the cyan to orange emission range in tissues is challenging due to intrinsic fluorescence. A common problem is bleed-through of autofluorescent signals into the emission wavelength of the label of interest. Bleed-through is the result of two fluorophores overlapping in emission and excitation profiles, so that photons from one fluorophore fall into the detection range of the other. While bleed-through artifacts can be partially reduced with a stringent choice of the emission filters, this requires narrow collection channels, which reject any ambiguous wavelength and greatly decreases collection efficiency. This strategy generally proves difficult when applied to broad-spectra autofluorescence. mKusabira-Orange 2 (mKO2) is a fluorescent protein whose emission spectrum significantly over-laps with autofluorescence in zebrafish. In a *fli1:mKO2* zebrafish, where all of the vascular and endothelial cells are labeled, the fluorescent protein, mKO2, and autofluorescence signals due to pigments and yolk are difficult to distinguish (Fig. 5a, boxes). Grayscale renderings (Supplementary Fig. 15) provide information on the relative intensity of the multiple fluorophores in the sample but are not sufficient for specifically detecting the spatial distribution of the mKO2 signal. True-color representation (Fig. 5a, Supplementary Fig. 16) is limited in visualizing these spectral differences. SEER's angular map (Fig. 5b) provides a striking contrast between the subtly different spectral components inside this 4D $(x, y, z, \lambda)$ dataset. The angular reference map enhances changes in phase on the phasor plot which nicely dis-criminates shifts in the center wavelength of the spectra inside the sample (Supplementary Movie 1). Autofluorescence from pig-ment cells is considerably different from the *fli1:mKO2* fluores-cence (Fig. 5c–h). For example, the dorsal area contains a combination of mKO2 and pigment cells (Fig. 5e–f) not clearly distinct in the standard approaches. The angular map permits SEER to discriminate subtle spectral differences. Distinct colors represent the autofluorescence from yolk and from pigment cells (Fig. 5g, h), enriching the overall information provided by this single-fluorescently labeled specimen and enhancing the visuali-zation of mKO2 fluorescently labeled pan-endothelial cells.

Imaging and visualization of biological samples with multiple fluorescent labels are hampered by the overlapping emission spectra of fluorophores and autofluorescent molecules in the sample, complicating the visualization of the sample. A triple-labeled zebrafish embryo with Gt(desm-Citrine)$^{ct122a/+}$;Tg(kdrl: eGFP), H2B-Cerulean labeling, respectively, muscle, vasculature and nuclei, with contributions from pigments autofluorescence is rendered with standard approaches and SEER in 1D and 3D (Fig. 6). TrueColor representation (Fig. 6a, Supplementary Fig. 17) provides limited information on the inner details of the sample. Vasculature (eGFP) and nuclei (Cerulean) are highlighted with shades of cyan whereas autofluorescence and muscle (Citrine) are in shades of green (Fig. 6a) making both pairs difficult to distinguish. The intrinsic richness of colors in the sample is an ideal test for the gradient descent and radial maps.

The angular map separates spectra based mainly on their central (peak) wavelength, which correspond to phase differences in the phasor plot. The gradient descent map separates spectra with a bias on subtle spectral differences closer to the center of the phasor plot. Here we applied the mass morph and max morph modes to further enhance the distinction of spectra (Fig. 6b, c). With the mass morph mode, the muscle outline and contrast of the nuclei are improved by increasing the spatial separation of the fluorophores and suppressing the presence of autofluorescence from skin pigment cells (Fig. 6e). With the max morph mode (Fig. 6c), pixels with spectra closer to skin autofluorescence are visibly separated from muscle, nuclei and vasculature.

The enhancements of SEER are also visible in volumetric visualizations. The angular and gradient maps are applied to the triple-labeled 4D $(x, y, z, \lambda)$ dataset and visualized as maximum intensity projections (Fig. 6d–f). The spatial localization of fluorophores is enhanced in the mass morphed angular map, while the max morphed gradient descent map provides a better separation of the autofluorescence of skin pigment cells (Supplementary Movie 2). These differences are also maintained in different visualization modalities (Supplementary Fig. 18).

SEER helps to discern the difference between fluorophores even with multiple contributions from bleed though between labels and from autofluorescence. Particularly, morphed maps demonstrate a high sensitivity in the presence of subtle spectral differences. The triple-labeled example (Fig. 6) shows advantage of the morphed map, as it places the apex of the SEER map at the center of mass of the phasor histogram and compensates for the different excitation efficiencies of the fluorescent proteins at 458 nm.

**Spectral differences visualized in combinatorial approaches.** Zebrabow[34] is the result of a powerful genetic cell labeling technique based on stochastic and combinatorial expression of different relative amounts of a few genetically encoded, spectrally distinct fluorescent proteins[34,50,51]. The Zebrabow (Brainbow) strategy combines the three primary colors red, green and blue, in different ratios, to obtain a large range of colors in the visual palette, similar to modern displays[52]. Unique colors arise from the combination of different ratios of RFP, CFP, and YFP, achieved by a stochastic Cre-mediated recombination[50].

This technique has been applied multiple applications, from axon and lineage tracing[51–55] to cell tracking during develop-ment[56,57], in which a specific label can be used as a cellular identifier to track descendants of individual cells over time and space.

The challenge is acquiring and analyzing the subtle differences in hues among these hundreds of colors. Multispectral imaging provides the added dimension required for an improved acquisition; however, this modality is hampered by both instrumental limitations and spectral noise. Furthermore, current image analysis and visualization methods interpret the red, yellow, and cyan fluorescence as an RGB additive combination and visualize it as a color picture, similar to the human eye perception of color. This approach is not well poised for distinguishing similar, yet spectrally unique, recombination ratios due to our difficulty in reliably identifying subtly different colors.

SEER overcomes this limitation by improving the analysis' sensitivity using our phasor-based interpretation of colors. Recombinations of labels belong to separate areas of the phasor plot, simplifying the distinction of subtle differences. The standard reference maps and modes associate a color easily distinguishable by eye, enhancing the subtle spectral recombina-tion. SEER simplifies the determination of differences between cells for combinatorial strategies, opening a novel window of analysis for brainbow samples.

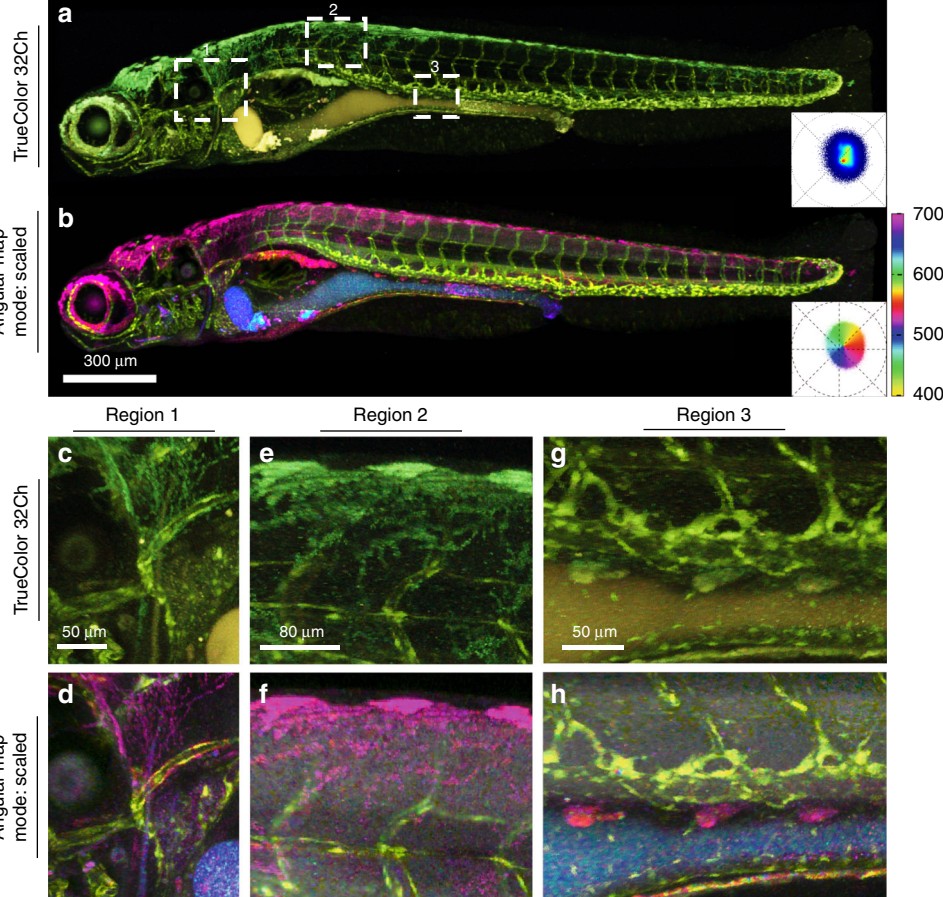

**Fig. 5 Visualization of a single fluorescence label against multiple autofluorescences.** Tg(*fli1:mKO2*) (pan-endothelial fluorescent protein label) zebrafish was imaged with intrinsic signal arising from the yolk and xanthophores (pigment cells). Live imaging was performed using a multispectral confocal (32 channels) fluorescence microscope with 488 nm excitation. The endothelial mKO2 signal is difficult to distinguish from intrinsic signals in a (**a**) maximum intensity projection TrueColor 32 channels Image display (Bitplane Imaris, Switzerland). The SEER angular map highlights changes in spectral phase, rendering them with different colors (reference map, bottom right of each panel). **b** Here we apply the angular map with scaled mode on the full volume. Previously indistinguishable spectral differences (boxes 1, 2, 3 in panel **a**) are now easy to visually separate. Colorbar represents the main wavelength associated to one color in nanometers. **c–h** Zoomed-in views of regions 1–3 (from **a**) visualized in TrueColor (**c**, **e**, **g**) and with SEER (**d**, **f**, **h**) highlight the differentiation of the pan-endothelial label (yellow) distinctly from pigment cells (magenta). The improved sensitivity of SEER further distinguishes different sources of autofluorescence arising from yolk (blue and cyan) and pigments.

We imaged an *Tg(ubi:Zebrabow)* sample and visualized its multiple genetic recombinations using SEER. The results (Fig. 7, Supplementary Fig. 19) highlight the difficulty of visualizing these datasets with standard approaches as well as how the compressive maps simplify the distinction of both spectrally close and more separated recombinations.

## Discussion
Standard approaches for the visualization of hyperspectral datasets trade computational expense for improved visualization. In this work, we show that the phasor approach can define a new compromise between computational speed and rendering performance. The wavelength encoding can be achieved by proper conversion and representation by the spectral phasor plot of the Fourier transform real and imaginary components. The phasor representation offers effortless interpretation of spectral information. Originally developed for fluorescence lifetime analysis[22] and subsequently brought to spectral applications[26,28,29], here the phasor approach has been applied to enhance the visualization of multi- and hyperspectral imaging. Because of the refined spectral discrimination achieved by these phasor-based tools, we call this approach Spectrally Enhanced Encoded Representations (SEER).

SEER offers a computationally efficient and robust method that converts spectral $(x, y, \lambda)$ information into a visual representation, enhancing the differences between labels. This approach makes more complete use of the spectral information. Prior analyses employed the principal components or specific spectral bands of the wavelength dimension. Similarly, previous phasor analyses interpreted the phasor using selected regions of interest. Our work explores the phasor plot as a whole and represents that complete information set as a color image, while maintaining efficiency and minimizing user interaction. The function can be achieved quickly and efficiently even with large data sizes, circumventing the typical computational expense of hyperspectral processing. Our tests show SEER can process a 3.7 GB dataset with $1.26 \times 10^8$ spectra in 6.6 s and a 43.88 GB dataset with $1.47 \times 10^9$ spectra in 87.3 s, including denoising of data. Comparing with the python module, scikit-learn's implementation of fast independent component analysis (fastICA), SEER provides up to a 67-fold speed increase (Supplementary Fig. 1) and lower virtual memory usage.

Processing speed comparison between SEER and fastICA for the multispectral fluorescent data shown in Figs. 4–7 is presented in Supplementary Table 2. SEER's computation time ranged between 0.44 (for Fig. 4) and 6.27 s (for Fig. 5) where the corresponding

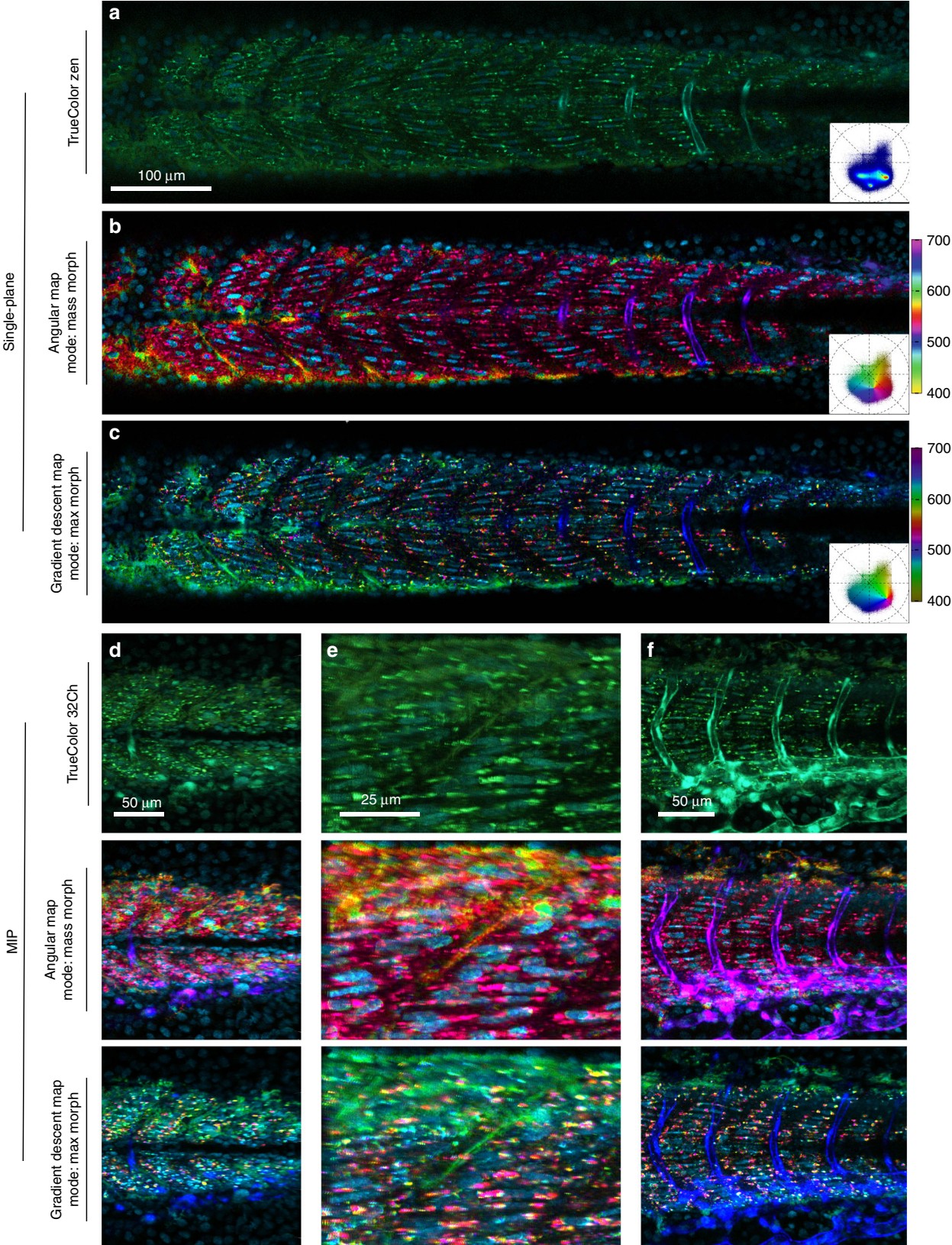

timing for fastICA was 3.45 and 256.86 s, respectively, with a speed up in the range of 7.9–41-folds (Supplementary Fig. 20), in accordance with the trend shown in Supplementary Fig. 1. These results were obtained using Python, an interpreted language. Implementation of SEER with a compiled language could potentially increase speed by one order of magnitude. The spectral maps presented here reduce the dimensionality of these large datasets and assign colors to a final image, providing an overview of the data prior to a full-scale analysis.

A simulation comparison with other common visualization approaches such as Gaussian kernel and peak wavelength selection (Supplementary Fig. 21, see Methods) shows an increased

**Fig. 6 Triple label fluorescence visualization.** Zebrafish embryo *Tg(kdrl:eGFP); Gt(desmin-Citrine);Tg(ubiq:H2B-Cerulean)* labeling, respectively, vasculature, muscle, and nuclei. Live imaging with a multispectral confocal microscope (32-channels) using 458 nm excitation. Single plane slices of the tiled volume are rendered with TrueColor and SEER maps. **a** TrueColor image display (Zen, Zeiss, Germany). **b** Angular map in center of mass morph mode improves contrast by distinguishable colors. The resulting visualization enhances the spatial localization of fluorophores in the sample. **c** Gradient descent map in max morph mode centers the color palette on the most frequent spectrum in the sample, highlighting the spectral changes relative to it. In this sample, the presence of skin pigment cells (green) is enhanced. 3D visualization of SEER maintains these enhancement properties. Colorbars represent the main wavelength associated to one color in nanometers. Here we show (**d**, **e**, **f**) TrueColor 32 channels Maximum Intensity Projections (MIP) of different sections of the specimen rendered in TrueColor, angular map center of mass mode and gradient descent max mode. The selected views highlight SEER's performance in the **d** overview of somites, **e** zoom-in of somite boundary, and **f** lateral view of vascular system.

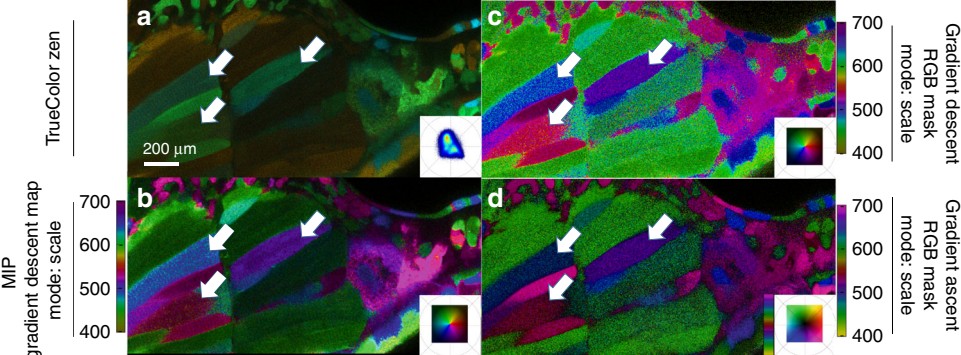

**Fig. 7 Visualization of combinatorial expression on Zebrabow samples.** Maximum intensity projection renderings of *Tg(ubi:Zebrabow)* muscle[34] acquired live in multispectral confocal mode with 458 nm excitation. **a** The elicited signal (e.g., white arrows) is difficult to interpret in the TrueColor Image display (Zen Software, Zeiss, Germany). **b** Discerning spectral differences is increasingly simpler with gradient descent map scaled to intensities, while compromising on the brightness of the image. **c** Gradient descent and **d** gradient ascent RGB masks in scale mode show the color values assigned to each pixel and greatly improve the visual separation of recombined CFP, YFP, and RFP labels. Colorbars represent the main wavelength associated to one color in nanometers.

spectral separation accuracy (see Methods) for SEER to associate distinct colors to closely overlapping spectra under different noise conditions (Supplementary Fig. 22). The spectral separation accuracy improvement was 1.4–2.6-fold for highly overlapping spectra, 0–8.9 nm spectra maxima distance, and 1.5–2.7-fold for overlapping spectra with maxima separated by 17.8–35.6 nm (Supplementary Figs. 23, 24, see Methods). Additional discussion is available in the Supplementary Peer Review file.

Quantification of RGB images by colorfulness, contrast and sharpness show that SEER generally performs better than standard visualization approaches (Supplementary Fig. 25, see Methods). SEER's average enhancement was 2–19% for colorfulness, 11–27% for sharpness and 2–8% for contrast (Supplementary Table 3) for the datasets of Figs. 4–7. We then performed a measure of the Color Quality Enhancement (CQE)[58], a metric of the human visual perception of color image quality (Supplementary Table 4). The CQE score of SEER was higher than the standard, with improvement of 11–26% for Fig. 4, 7–98% for Fig. 5, 14–25% for Fig. 6, and 12–15% for Fig. 7 (Supplementary Fig. 25, Supplementary Note 3, see Methods).

Flexibility is a further advantage of our method. The users can apply several different standard reference maps to determine which is more appropriate for their data and enhance the most important image features. The modes provide a supplementary enhancement by adapting the References to each dataset, in terms of size and distribution of spectra in the dataset. Scaling maximizes contrast by enclosing the phasor distribution, it maintains linearity of the colormap. Max and center of mass modes shift the apex of the distribution to a new center, specifically the most frequent spectrum in the dataset or the weighted color-frequency center of mass of the entire dataset. These modes adapt and improve the specific visualization properties for each map to the dataset currently being analyzed. As a result, each map offers increased sensitivity to

specific properties of the data, amplifying, for example, minor spectral differences or focusing on major wavelength components. The adaptivity of the SEER modes can prove advantageous for visually correcting the effect of photobleaching in samples, by changing the apex of the map dynamically with the change of intensities (Supplementary Fig. 26, Supplementary Movie 3).

SEER can be applied to fluorescence, as performed here, or to standard reflectance hyper- and multispectral imaging. These phasor remapping tools can be used for applications in fluorescence lifetime or combined approaches of spectral and lifetime imaging. With multispectral fluorescence, this approach is promising for real-time imaging of multiple fluorophores, as it offers a tool for monitoring and segmenting fluorophores during acquisition. Live-imaging visualization is another application for SEER. The gradient descent map, for example, in combination with denoising strategies[29] can minimize photobleaching and -toxicity, by enabling the use of lower excitation power. SEER overcomes the challenges in visualization and analysis deriving from low signal-to-noise images, such as intrinsic signal autofluorescence imaging. Among other complications, such image data can result in a concentrated cluster proximal to the phasor center coordinates. The gradient descent map overcomes this limitation and provides bright and distinguishable colors that enhance subtle details within the dim image.

It is worth noticing that the method is generally indifferent to the dimension being compressed. While in this work we explore the wavelength dimension, SEER can be utilized, in principle, with any *n*-dimensional dataset where *n* is larger than two. For instance, it can be used to compress and compare the dimension of lifetime, space or time for multiple datasets. Some limitations that should be considered are that SEER pseudo-color representation sacrifices the true color of the image, creating inconsistencies with the human eyes' expectation of the original image

and does not distinguish identical signals arising from different biophysical events (Supplementary Note 2). SEER is currently intended as a preprocessing visualization tool and, currently, is not utilized for quantitative analysis. Combining SEER with novel color-image compatible segmentation algorithms[59,60] might expand the quantitative capabilities of the method.

New multidimensional, multi-modal instruments will more quickly generate much larger datasets. SEER offers the capability of processing this explosion of data, enabling the interest of the scientific community in multiplexed imaging.

## Methods

**Simulated hyperspectral test chart**. To account for the Poisson noise and detector noise contributed by optical microscopy, we generated a simulated hyperspectral test chart starting from real imaging data with a size of $x$: 300 pixels, $y$: 300 pixels, and lambda: 32 channels. S1, S2, and S3 spectra were acquired, respectively, from zebrafish embryos labeled only with CFP, YFP, and RFP, where the spectrum in Fig. 1a is identical to the center cell of the test chart Fig. 1d. In each cell three spectra are represented after shifting the maxima by d1 or d2 nanometers with respect to S2. Each cell has its corresponding spectra of S1, S2, and S3 (Supplementary Fig. 3).

**Standard RGB visualizations**. The TrueColor RGB image (Supplementary Figs. 3, 8, 21) is obtained through compression of the hyperspectral cube into the RGB 3 channels color space by generating a Gaussian radial basis function kernel[61] $K$ for each RGB channel. This kernel $K$ acts as a scaling factor and is defined as:

$$K_i(x_i, x') = e^{\frac{-|x_i - x'|^2}{2\sigma^2}} \tag{1}$$

where $x'$ is the center wavelength of R or B or G. For example, when $x' = 650$ nm, the associated RGB color space is (R:1, G:0, B:0). Both $x$ and $K$ are defined as $32 \times 1$ vectors, representing, respectively, the 32-channel spectrum of one single pixel and the normalized weight of each R, G, and B channel. $i$ is the channel index of both vectors. $K_i$ represents how similar channel $i$ is to each R/G/B channel, and $\sigma$ is the deviation parameter.

We compute RGB color space $c$ by a dot product of the weight vector $K$ and $\lambda$ at corresponding channel R/G/B:

$$c = \sum_{i=1}^{i=32} \lambda_i \times K_i \tag{2}$$

where $\lambda$ is a vector of the wavelengths captured by the spectral detector in an LSM780 inverted confocal microscope with lambda module (Zeiss, Jena, Germany) and $\lambda_i$ is the center wavelength of channel $i$. Gaussian kernel was set at 650, 510, 470 nm for RGB, respectively, as default (Supplementary Figs. 3s, 8, 13e, 16e, 17e, 19j).

The same Gaussian kernel was also changed adaptively to the dataset to provide a spectral contrast stretching on the visualization and focus the visualization on the most utilized channels. The average spectrum for the entire dataset is calculated and normalized. The intersect at 10% (Supplementary Figs. 13f, 16f, 17f, 19f), 20% (Supplementary Figs. 13g, 16g, 17g, 19g), and 30% (Supplementary Figs. 13h, 16h, 17h, and 19h). of the intensity is obtained and used as a center for the blue and red channels. The green channel is centered halfway between red and blue. Representations of these adaptations are reported in Supplementary Fig. 21g, h, i.

The TrueColor 32 Channels image (Figs. 1c, 5a, c, e, g, 6a–d, e, f, Supplementary Figs. 13c, 16c, 17c, and 19c) was rendered as a 32 channels Maximum Intensity Projection using Bitplane Imaris (Oxford Instruments, Abingdon, UK). Each channel has a known wavelength center (32 bins, from 410.5 to 694.9 nm with 8.9 nm bandwidth). Each wavelength was associated with a color according to classical wavelength-to-RGB conversions[62] as reported in Supplementary Fig. 21f. The intensity for all channels was contrast-adjusted (Imaris Display Adjustment Settings) based on the channel with the largest information. A meaningful range for rendering was identified as the top 90% in intensity of the normalized average spectrum for the dataset (Supplementary Figs. 13b, 16j, 17j, 19b). Channels outside of this range were excluded for rendering. Furthermore, for 1 photon excitation, channels associated to wavelength lower than the laser excitation (for example, channels 1–5 for laser 458 nm) were excluded from rendering.

Peak Wavelength representation (Supplementary Figs. 13d, 16d, 17d, 19d, 21, and 23) reconstructs an RGB image utilizing, for each pixel, the color associated to the wavelength at which maximum intensity is measured. Wavelength-to-RGB conversion was performed using a python function (http://bioimaging.usc.edu/software.html#HySP) adapted from Dan Bruton's work[62]. A graphical representation is reported in Supplementary Fig. 21f.

**Spectral separation accuracy calculation**. We utilize the Simulated Hyperspectral Test Chart to produce different levels of spectral overlap and signal-to-noise ratio (SNR). We utilize multiple RGB visualization approaches for producing compressed RGB images (Supplementary Figs. 21, 22). Each panel of the simulation is constructed by three different spectra, organized as three concentric squares $Q_1$, $Q_2$, $Q_3$

(Supplementary Fig. 3). The maximal contrast visualization is expected to have three well separated colors, in this case red, green, and blue. For quantifying this difference, we consider each (R, G, B) vector, with colors normalized [0, 1], in each pixel as a set of Euclidean coordinates $(x, y, z)$ and for each pixel calculate the Euclidean distance:

$$l_{12} = \sqrt{\sum_{i=R}^{B} (p_{Q1} - p_{Q2})_i^2} \tag{3}$$

where $l_{12}$ is the color distance between square $Q_1$ and $Q_2$. $p_{Q1}$ and $p_{Q2}$ are the (R, G, B) vectors in the pixels considered, $i$ is the color coordinate R, G, or B. The color distances $l_{13}$ and $l_{23}$ between squares $Q_1 Q_3$ and $Q_2 Q_3$, respectively, are calculated similarly. The accuracy of spectral separation (Supplementary Fig. 23) is calculated as:

$$Sp.sep.acc = \frac{(l_{12} + l_{13} + l_{23})}{l_{red-green} + l_{red-blue} + l_{green-blue}} \tag{4}$$

where the denominator is the maximum color distance which can be achieved in this simulation, where $Q_1$, $Q_2$, and $Q_3$ are, respectively, pure red, green, and blue, therefore:

$$l_{red-green} + l_{red-blue} + l_{green-blue} = 3\sqrt{2} \tag{5}$$

**Compressive spectral algorithm and map reference design: phasor calculations**. For each pixel in an image, we acquire the sequence of intensities at different wavelengths $I(\lambda)$. Each spectrum $I(\lambda)$ is discrete Fourier transformed into a complex number $g_{x,y,z,t} + is_{x,y,z,t}$. Here $i$ is the imaginary unit, while $(x, y, z, t)$ denotes the spatio-temporal coordinates of a pixel in a 5D dataset.

The transforms used for real and imaginary components are

$$g_{x,y,z,t}(k)_{|k=2} = \frac{\sum_{\lambda_0}^{\lambda_N} I(\lambda) * \cos\left(\frac{2\pi k\lambda}{N}\right) * \Delta\lambda}{\sum_{\lambda_0}^{\lambda_N} I(\lambda) * \Delta\lambda} \tag{5}$$

$$s_{x,y,z,t}(k)_{|k=2} = \frac{\sum_{\lambda_0}^{\lambda_N} I(\lambda) * \sin\left(\frac{2\pi k\lambda}{N}\right) * \Delta\lambda}{\sum_{\lambda_0}^{\lambda_N} I(\lambda) * \Delta\lambda} \tag{6}$$

Where $\lambda_0$ and $\lambda_N$ are the initial and final wavelengths, respectively, $N$ is the number of spectral channels, $\Delta\lambda$ is the wavelength bandwidth of a single channel. $k$ is the harmonic. In this work, we utilized harmonic $k = 2$. The effects of different harmonic numbers on the SEER visualization are reported in (Supplementary Note 1).

**Standard map reference**. Association of a color to each phasor coordinate $(g, s)$ is performed in two steps. First, the reference system is converted from Cartesian to polar coordinates $(r, \theta)$.

$$(r, \theta) = \left(\sqrt{g^2 + s^2}, \frac{s}{g}\right) \tag{7}$$

These polar coordinate values are then transformed to the hue, saturation, value (HSV) color model utilizing specific settings for each map, as listed below. Finally, any color generated outside of the $r = 1$ boundary is set to black.

Gradient descent:

$$hue = \theta$$

$$saturation = 1$$

$$value = 1 - 0.85 * r$$

Gradient ascent:

$$hue = 0$$

$$saturation = 1$$

$$value = r$$

Radius:

Each $r$ value from 0 to 1 is associated to a level in the jet colormap from the *matplotlib* package

Angle:

$$hue = 0$$

$$saturation = 1$$

$$value = 1$$

**Tensor map**. Visualization of statistics on the phasor plot is performed by means of the mathematical gradient. The gradient is obtained in a two-step process.

First, we compute the two-dimensional derivative of the phasor plot histogram counts by utilizing an approximation of the second order accurate central difference.

Each bin $F(g, s)$ has a central difference: $\frac{\partial F}{\partial g}, \frac{\partial F}{\partial s}$, with respect to the differences in $g$ (horizontal) and $s$ (vertical) directions with unitary spacing $h$. The approximation becomes

$$\begin{aligned}\frac{\partial F}{\partial s} &= \frac{F\left(s+\frac{1}{2}h, g\right) - F\left(s-\frac{1}{2}h, g\right)}{h} \\ &= \frac{\frac{F(s+h,g)+F(s,g)}{2} - \frac{F(s,g)+F(s-h,g)}{2}}{h} \\ &= \frac{F(s+h,g) - F(s-h,g)}{2h}\end{aligned} \qquad (8)$$

Similarly

$$\frac{\partial F}{\partial g} = \frac{F(s, g+h) - F(s, g-h)}{2h} \qquad (9)$$

Second, we calculate the square root of the sum of squared differences $D(s,g)$ as

$$D(g,s) = \sqrt{\left(\frac{\partial F}{\partial g}\right)^2 + \left(\frac{\partial F}{\partial s}\right)^2} \qquad (10)$$

obtaining the magnitude of the derivative density counts. With this gradient histogram, we then connect the phasor coordinates with same $D(s, g)$ gradients with one contour. All gradients are then normalized to (0,1). Finally, pixels in the hyperspectral image corresponding to the same contour in phasor space will be rendered the same color. In the reference map, red represents highly dense gradients, usually at the center of a phasor cluster. Blue, instead, represents the sparse gradient that appears at the edge circumference of the phasor distributions.

**Scale mode**. In this mode, the original square standard reference maps are transformed to a new boundary box adapted to each dataset's spectral distribution.

The process of transformation follows these steps. We first determine the boundary box (width $\omega$, height $h$) based on the cluster appearance on the phasor plot. We then determine the largest ellipsoid that fits in the boundary box. Finally, we warp the unit circle of the original map to the calculated ellipsoid.

Using polar coordinates, we represent each point $P$ of the standard reference map with phasor coordinates $(g_i, s_i)$ as

$$P(g_i, s_i) = P(r_i * \cos\theta_i, r_i * \sin\theta_i) \qquad (11)$$

The ellipsoid has semi-major axes

$$a = \frac{\omega}{2} \qquad (12)$$

and semi-minor axes

$$b = \frac{h}{2} \qquad (13)$$

Therefore, the ellipse equation becomes

$$\left(\frac{g_i}{\omega/2}\right)^2 + \left(\frac{s_i}{h/2}\right)^2 = rad^2 \qquad (14)$$

Where $rad$ is a ratio used to scale each radius $r_i$ in the reference map to a proportionally corresponding distance in the boundary box-adapted ellipse, which in polar coordinates becomes

$$rad^2 = r_i^2 * \left(\left(\frac{\cos\theta_i}{\frac{\omega}{2}}\right)^2 + \left(\frac{\sin\theta_i}{\frac{h}{2}}\right)^2\right) \qquad (15)$$

Each point $P(g_i, s_i)$ of the standard reference map is geometrically scaled to a new coordinate $(g_o, s_o)$ inside the ellipsoid using forward mapping, obtaining the equation

$$(r_o, \theta_o) = \left(\sqrt{g_i^2 + s_i^2}/rad, \tan^{-1}\frac{g_i}{s_i}\right) \qquad (16)$$

This transform is applied to all standard reference maps to generate the respective scaled versions.

**Morph mode**. We linearly morph each point $P(g_i, s_i)$ to a new point $P'(g_o, s_o)$ by utilizing a shifted-cone approach. Each standard map reference is first projected on to a 3D conical surface centered on the phasor plot origin and with unitary height (Supplementary Fig. 27a-c, Supplementary Movie 4). Each point $P$ on the standard map is given a $z$ value linearly, starting from the edge of the phasor universal circle. The original standard map (Supplementary Fig. 27c) can, thus, be interpreted as a top view of a right cone with $z = 0$ at the phasor unit circle and $z = 1$ at the origin (Supplementary Fig. 27a).

We then shift the apex $A$ of the cone to the computed weighted average or maxima of the original 2d histogram, producing an oblique cone (Supplementary Fig. 27b) centered in $A'$.

In this oblique cone, any horizontal cutting plane is always a circle with center $O'$. Its projection $O'_\perp$ is on the line joining the origin $O$ and projection of the new

center $A_\perp'$ (Supplementary Fig. 27b–d). As a result, all of the points in each circle are shifted towards the new center $A_\perp'$ on the phasor plot. We first transform the coordinates $(g_i, s_i)$ of each point $P$ to the morphed map coordinates $(s_o, g_o)$, and then obtain the corresponding $(r_o, \theta_o)$ necessary for calculating Hue, Saturation, and Value.

In particular, a cutting plane with center $O'$ has a radius of $r'$ (Supplementary Fig. 27). This cross section projects on a circle centered in $O'_\perp$ with the same radius. Using geometrical calculations, we obtain:

$$OO'_\perp = \alpha * OA'_\perp, \qquad (17)$$

where $\alpha$ is a scale parameter. By taking the approximation,

$$\Delta O'OO'_\perp \sim \Delta A'OA'_\perp, \qquad (18)$$

we can obtain

$$OO' = \alpha * OA'. \qquad (19)$$

Furthermore, given a point $N'$ on the circumference centered in $O'$, Eq. (14) also implies that:

$$O'N' = (1 - \alpha) * ON'_\perp, \qquad (20)$$

which is equivalent to

$$r' = (1 - \alpha) * R. \qquad (21)$$

where $R$ is the radius of the phasor plot unit circle.

With this approach, provided a new center $A_\perp'$ with a specific $\alpha$, we obtain a collection of scaled circles with centers on line $OA'_\perp$. In boundary cases, when $\alpha = 0$, the scaled circle is the origin, while $\alpha = 1$ is the unit circle. Given any cutting plane $O'$, the radius of this cross section always satisfies this identity:

$$r'^2 = (g_i - \alpha * g_{A\perp'})^2 + (s_i - \alpha * s_{A\perp'})^2 = (1 - \alpha)^2 \cdot R^2 \qquad (22)$$

The coordinates of a point $P''(g_o, s_o)$ for a new morphed map centered in $A_\perp'$ are:

$$(g_o, s_o) = (g_i - \alpha * g_{A\perp'}, s_i - \alpha * s_{A\perp'}) \qquad (23)$$

Finally, we compute

$$(r_o, \theta_o) = \left(\sqrt{g_o^2 + s_o^2}, \tan^{-1}\frac{s_o}{g_o}\right) \qquad (24)$$

and then assign colors based on the newly calculated hue, saturation, and value to generate the morph mode references.

**Color image quality calculations: colorfulness**. Due to the inherent lack of ground truth in experimental fluorescence microscopy images, we utilized an established model for calculating the color quality of an image without a reference[63]. The colorfulness is one of three parameters, together with sharpness and contrast, utilized by Panetta et al.[58] to quantify the overall quality of a color image. Two opponent color spaces are defined as:

$$\alpha = R - G \qquad (25)$$

$$\beta = 0.5(R + G) - B \qquad (26)$$

where R, G, and B are the red, green, and blue channels, respectively, $\alpha$ and $\beta$ are red-green and yellow-blue spaces. The colorfulness utilized here is defined as:

$$\text{Colorfulness} = 0.02 \log\left(\frac{\sigma_\alpha^2}{|\mu_\alpha|^{0.2}}\right) \log\left(\frac{\sigma_\beta^2}{|\mu_\beta|^{0.2}}\right) \qquad (27)$$

With $\sigma_\alpha^2, \sigma_\beta^2, \mu_\alpha, \mu_\beta$, respectively, as the variances and mean values of the $\alpha$ and $\beta$ spaces[58].

**Sharpness**. We utilize EME[64], a Weber based measure of enhancement. EME is defined as follows:

$$\text{EME}_{\text{sharp}} = \frac{2}{k_1 k_2} \sum_{l=1}^{k_1} \sum_{l=1}^{k_2} \log\left(\frac{I_{\max,k,l}}{I_{\min,k,l}}\right) \qquad (28)$$

Where $k_1$, $k_2$ are the blocks used to divide the image and $I_{\max,k,l}$ and $I_{\min,k,l}$ are the maximum and minimum intensities in the blocks. EME has been shown to correlate with a human observation of sharpness in color images[58] when associated with a weight $\lambda_c$ for each color component.

$$\text{Sharpness} = \sum_{c=1}^{3} \lambda_c \text{EME}_{\text{sharp}} \qquad (29)$$

Where the weights for different color components used in this article are $\lambda_R = 0.299$, $\lambda_G = 0.587$, $\lambda_B = 0.114$ in accordance with NTSC standard and values reported in literature[58].

**Contrast**. We utilize Michelson-Law measure of enhancement AME[65], an effective evaluation tool for contrast in grayscale images, designed to provide larger metric value for larger contrast images. AME is defined as:

$$\text{AME}_{\text{contrast}} = \frac{1}{k_1 k_2} \sum_{l=1}^{k_1} \left( \sum_{l=1}^{k_2} \log \left( \frac{I_{\max,k,l} + I_{\min,k,l}}{I_{\max,k,l} - I_{\min,k,l}} \right) \right)^{-0.5} \quad (30)$$

where $k_1$, $k_2$ are the blocks used to divide the image, and $I_{\max,k,l}$ and $I_{\min,k,l}$ are the maximum and minimum intensities in the blocks. The value of contrast for color images was then calculated as:

$$\text{Contrast} = \sum_{c=1}^{3} \lambda_c \text{AME}_{\text{contrast}} \quad (31)$$

With the same weights $\lambda_c$ utilized for sharpness.

**Color Quality Enhancement**. We utilize Color Quality Enhancement (CQE), a polling method to combine colorfulness, sharpness, and contrast into a value that has both strong correlation and linear correspondence with human visual perception of quality in color images[58]. CQE is calculated as:

$$\text{CQE} = c_1 \text{colorfulness} + c_2 \text{sharpness} + c_3 \text{contrast} \quad (31)$$

Where the linear combination coefficients for CQE measure were set to evaluate contrast change according to values reported in literature[58], $c_1 = 0.4358$, $c_2 = 0.1722$, and $c_3 = 0.3920$.

**Mouse lines**. Mice imaging was approved by the Institutional Animal Care and Use Committee (IACUC) of the Children's Hospital of Los Angeles (permit number: 38616) and of the University of Southern California (permit number: 20685). Experimental research on vertebrates complied with institutional, national and international ethical guidelines. Animals were kept on a 13:11 h light:dark cycle. Animals were breathing double filtered air, temperature in the room was kept at 68–73 F, and cage bedding was changed weekly. All these factors contributed to minimize intra- and inter-experiment variability. Adult 8-week-old C57Bl mice were euthanized with euthasol. Tracheas were quickly harvested from the mouse, washed in PBS, and cut longitudinally alongside the muscolaris mucosae in order to expose the lumen. A 3 mm × 3 mm piece of the trachea was excised and arranged onto a microscope slide for imaging.

**Zebrafish lines**. Lines were raised and maintained following standard literature practice[66] and in accordance with the Guide for the Care and Use of Laboratory Animals provided by the University of Southern California. Fish samples were part of a protocol approved by the IACUC (permit number: 12007 USC).

Transgenic FlipTrap $Gt(desm\text{-}Citrine)$ ct122a/+ line is the result of previously reported screen[67], Tg(kdrl:eGFP)[s843] line[68] was provided by the Stainier lab (Max Planck Institute for Heart and Lung Research). The Tg(ubi:Zebrabow) line was a kind gift from Alex Schier[34]. Controllable recombination of fluorophores was obtained by crossing homozygous Tg(ubi:Zebrabow) adults with a Tg(hsp70l: Cerulean-P2A-CreER[T2]) line. Embryos were raised in Egg Water (60 μg ml⁻¹ of Instant Ocean and 75 μg ml⁻¹ of CaSO4 in Milli-Q water) at 28.5 °C with addition of 0.003% (w v⁻¹) 1-phenyl-2-thiourea (PTU) around 18 hpf to reduce pigment formation[69].

Zebrafish samples with triple fluorescence were obtained by crossing Gt(desm-Citrine)ct122a/+ with Tg(kdrl:eGFP) fish followed by injection of 100 pg per embryo of mRNA encoding H2B-Cerulean at one-cell stage[29]. Samples of Gt(desm-Citrine)ct122a/+;Tg(kdrl:eGFP); H2B-Cerulean were imaged with 458 nm laser to excite Cerulean, Citrine and eGFP and narrow 458–561 nm dichroic for separating excitation and fluorescence emission.

**Plasmid constructions: pDestTol2pA2-hsp70l:Cerulean-P2A-CreERT2 (for generating Tg(hsp70l:Cerulean-P2A-CreER[T2]) line)**. The coding sequences for Cerulean, CreERT2, and woodchuck hepatitis virus posttranscriptional regulatory element (WPRE) were amplified from the vector for Tg(bactin2:cerulean-cre)[67], using primers #1 and #2 (complete list of primers is reported in Supplementary Table 5), pCAG-ERT2CreERT2 (Addgene #13777) using primers #3 and #4, and the vector for Tg(PGK1:H2B-chFP)[70] using primers #5 and #6, respectively. Then Cerulean and CreERT2 sequences were fused using a synthetic linker encoding P2A peptide[71]. The resultant Cerulean-P2A-CreERT2 and WPRE sequences were cloned into pDONR221 and pDONR P2R-P3 (Thermo Fisher Scientific), respectively. Subsequent MultiSite Gateway reaction was performed using Tol2kit vectors according to developer's manuals[72]. p5E-hsp70l (Tol2kit #222), pDONR221-Cerulean-P2A-CreER, and pDONR P2R-P3-WPRE were assembled into pDest-Tol2pA2 (Tol2kit #394)[73,74].

**pDestTol2pA2-fli1:mKO2 (for generating Tg(fli1:mKO2) line)**. The coding sequence for mKO2 was amplified from mKO2-N1 (addgene #54625) using primers #7 and #8, and cloned into pDONR221. Then p5Efli1ep (addgene #31160), pDONR221-mKO2, and pDONR P2R-P3-WPRE were assembled into pDest-Tol2pA2 as described above.

**Microinjection and screening of transgenic zebrafish lines**. 2.3 nL of a solution containing 20 pg nL⁻¹ plasmid DNA and 20 pg nL⁻¹ tol2 mRNA was injected into the one-cell stage embryo obtained through crossing AB with Casper zebrafish[75]. The injected F0 embryos were raised and crossed to casper zebrafish for screening. The F1 embryos for prospective Tg(hsp70l:Cerulean-P2A-CreER[T2]) line and Tg(fli1: mKO2) were screened for ubiquitous Cerulean expression after heat shock for 30 min at 37 °C, and mKO2 expression restricted in vasculatures, respectively. Positive individual F1 adults were subsequently outcrossed to casper zebrafish, and their offspring with casper phenotype were then used for experiments when 50% transgene transmission was observed in the subsequent generation, indicating single transgene insertions.

**Sample preparation and multispectral image acquisition and instrumentation**. Images were acquired on a Zeiss LSM780 inverted confocal microscope equipped with QUASAR detector (Carl Zeiss, Jena, Germany). A typical dataset comprised 32 spectral channels, covering the wavelengths from 410.5 nm to 694.9 nm with 8.9 nm bandwidth, generating an $x,y,\lambda$ image cube. Detailed acquisition parameters are reported in Supplementary Table 1.

Zebrafish samples for in vivo imaging were prepared by placing 5–6 embryos at 24–72 hpf into 1% agarose (cat. 16500-100, Invitrogen) molds created in an imaging dish with no. 1.5 coverglass bottom, (cat. D5040P, WillCo Wells) using a custom-designed negative plastic mold[45]. Stability of the embryos was ensured by adding ~2 ml of 1% UltraPure low-melting-point agarose (cat. 16520-050, Invitrogen) solution prepared in 30% Danieau (17.4 mM NaCl, 210 μM KCl, 120 μM MgSO4·7H2O, 180 μM Ca(NO3)2, 1.5 mM HEPES buffer in water, pH 7.6) with 0.003% PTU and 0.01% tricaine. This solution was subsequently added on top of the mounted embryos. Upon agarose solidification at room temperature (1–2 min), the imaging dish was topped with 30% Danieau solution and 0.01% tricaine at 28.5 °C. Imaging on the inverted confocal microscope was performed by positioning the imaging dish on the microscope stage. For Tg(ubi:Zebrabow) samples, to initiate expression of CreER[T2], embryos were heat-shocked at 15 h post fertilization at 37 °C in 50 ml falcon tubes within a water bath before being returned to a 28.6 °C incubator. To initiate recombination of the zebrabow transgene, 5 μM 4-OHT (Sigma; H7904) was added to culture media 24 h post fertilization. Samples of Tg(ubi:Zebrabow) were imaged using 458 nm laser to excite CFP, YFP, and RFP in combination with a narrow 458 nm dichroic.

Mouse tracheal samples were collected from wild-type C57Bl mice and mounted on a coverslip with sufficient Phosphate Buffered Saline to avoid dehydration of the sample. Imaging was performed in 2-photon mode exciting at 740 nm with a 690+ nm dichroic.

**Non-de-scanned (NDD) multiphoton fluorescence lifetime imaging (FLIM) and analysis**. Fluorescence lifetime imaging microscopy (FLIM) data were acquired with a two-photon microscope (Zeiss LSM-780 inverted, Zeiss, Jena, Germany) equipped with a Ti:Sapphire laser system (Coherent Chameleon Ultra II, Coherent, Santa Clara, California) and an ISS A320 FastFLIM[76] (ISS, Urbana-Champaign, Illinois). The objective used was a 2-p optimized 40 × 1.1 NA water immersion objective (Korr C-Apochromat, Zeiss, Jena, Germany). Images with size of 256 × 256 pixels were collected with pixel dwell time of 12.6 μs pixel⁻¹. A dichroic filter (690+ nm) was used to separate the excitation light from fluorescence emission. Detection of fluorescence comprised a combination of a hybrid photomultiplier (R10467U-40, Hamamatsu, Hamamatsu City, Japan) and a 460/80 nm band-pass filter. Acquisition was performed using VistaVision software (ISS, Urbana-Champaign, Illinois). The excitation wavelength used was 740 nm with an average power of about 7 mW on the sample. Calibration of lifetimes for the frequency domain system was performed by measuring the known lifetime of the Coumarin 6 with a single exponential of 2.55 ns. FLIM data were collected until 100 counts in the brightest pixel of the image were acquired.

Data was processed using the SimFCS software developed at the Gratton Lab (Laboratory of Fluorescence Dynamics (LFD), University of California Irvine, www.lfd.uci.edu). FLIM analysis of intrinsic fluorophores was performed as previously described and reported in detail[23,35,43]. Phasor coordinates (g,s) were obtained through Fourier transformations. Cluster identification was utilized to associate specific regions in the phasor to pixels in the FLIM dataset according to published protocols[35].

**Reporting summary**. Further information on research design is available in the Nature Research Reporting Summary linked to this article.

## Data availability

All the relevant data are available from the corresponding author upon reasonable request. Datasets for Figs. 1–7 and simulations are available for download at http:// bioimaging.usc.edu/software.html#sampledatasets in the samples section.

## Code availability

All the relevant code is available from the corresponding author upon reasonable request. Software and instructions can be downloaded from http://bioimaging.usc.edu/software. html#HySP.

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

## Acknowledgements

The authors would like to thank S. Restrepo and J. Junge (Translational Imaging Center, University of Southern California) for helpful discussions. This work is supported by Department of Defense PR150666 and University of Southern California.

## Author contributions

W.S., D.E.S.K., and F.C. analyzed the results and wrote the software. F.C, L.A.T., D.W., and S.E.F. helped in the experimental design and data analysis. G.T. prepared the mouse samples. M.K. and B.S. prepared the zebrafish samples. M.K. generated the inducible and mKO2 zebrafish lines. F.C., H.J.C., C.A., and G.T. acquired data. C.A. performed the FLIM analysis. W.S, D.E.S.K, and F.C. wrote the paper.

## Competing interests

The authors declare no competing interests.
