## [Peer Review File · Nature Communications]

Editorial Note 1: Parts of this Peer Review File have been redacted as indicated to remove third-party material where no permission to publish could be obtained.

Editorial Note 2: Parts of this Peer Review File have been redacted as indicated to remove confidential information.

Reviewers' comments:

Reviewer #1 (Remarks to the Author):

In the manuscript "Visualization of hyperspectral fluorescent data with Spectrally Encoded Enhanced Representations (SEER)" by W. Shi, D.E.S. Koo et al, the authors describe a new method to enhance the representation of hyperspectral fluorescent data. Hyperspectral imaging is an important technique that, by adding a full spectral dimension to the fluorescence microscope, allows the imaging of a large number of fluorophores (even with overlapping spectra) simultaneously.

There are several established methods to process the hyperspectral dataset after the acquisition and, for instance, extract the multiple species contained in the specimen (e.g. linear unmixing). The same group has published recently a work on Nat Methods demonstrating the advantages of a Phasor-based analysis (HySP) with respect to linear unmixing.

Here the authors are focusing on the rapid, pre-analysis visualization of the hyperspectral dataset that can be useful in guiding the users during acquisition of the data. In this context, their main goal is to visualize the information contained in the spectral dimension by means of a compressed representation in standard RGB display. In contrast to commonly used dimensionality-reduction approaches (PCA, image-fusion), the authors propose an alternative, less computationally demanding, approach based on the use of Phasors. Essentially, they build different types of colormaps related to the pixel distribution of (g,s) values (the 'phasor plot').

The authors claim that the SEER approach:

- 1) improves computational efficiency (this is a feature of the phasor method, as is FFT based)
- 2) quantitatively enhances the analysis and interpretation of spectral information

In particular, the second claim is supported by the analysis of simulated Testcharts and experimental measurements on mouse and Zebrafish specimens.

In all cases, they compare the SEER visualization with what is considered the 'state of the art' or 'standard' visualization (also named 'True color') to highlight the benefits of SEER. According to all these comparisons, the performances of SEER are simply striking, in both simulations and experiments, as it greatly improves the visibility of spectral differences in the data.

In summary, the paper is well written and, if the claims stated by the authors are confirmed (see my major comments below), it represents an original and significant technical advance in the field of hyperspectral imaging. I think it is of interest and of potential impact for the growing number of users of this technique, now available not only on confocal microscopes but also on light sheet-based setups. In addition, the generation of 'smart' colormaps from the 2D phasor plot is a new idea with potential impact also in all the other contexts where the phasor can be applied.

For these reasons I think that the paper is of great interest for Nature Communications and well-worth of publication on this journal.

However, there are some fundamental points that must be addressed in order to understand the true advantages of the proposed method.

The phasor of a full spectrum is generally an approximation (g and s instead of n lambda points) that in many cases (FLIM autofluorescence, FLIM-FRET, FLIM superresolution, emission spectra etc) is sufficient for the description of the system under study. A second advantage is represented by the linear property which is the basis of the calculation of fractional components in many of these applications.

In the way the data are presented, I have the impression that the difference between the SEER and the 'standard' visualization might be exaggerated. I think it is important to understand how much of this difference is due to the phasor-based representations and how much is due to other effects (for instance the 'unused' spectral range in the TrueColor map). In addition, the 'quantitative' aspects of the phasor (e.g. the linearity that can be used to extract fractional components, etc) could be highlighted more in the paper; it would even be more attractive for the users of hyperspectral imaging to associate some numbers to the maps.

I ask the authors to address the following major concerns.

Major comments

1) One of the main results of the paper is the enhanced contrast provided by the SEER map with respect to the 'standard' or TrueColor map.

For instance let's consider Figure 1: the data are analyzed in spectral range 400-700 nm (Fig.1b) but the meaningful part of the fluorophores spectra (CFP to RFP) is in a smaller range, say 470-620nm. The TrueColor map should be adapted to this smaller range. In other words, the center wavelengths of R, G, B (which, by the way, are not reported except R=650nm) should be set closer to each other to render better the variation in the spectra.

In fact I also note that the phasor map (the SEER map) is constructed, in all the data, from the 2nd harmonic of the spectra. This mean that the phase of the phasor will highlight a variation of the emission maxima in the range $(700-400\text{nm})/2=150\text{nm}$.

I think a more fair comparison should be made between the SEER map and the 'standard' maps. For this reason I ask to perform the following tests:

a) Visualize all the data in the Figures with a TrueColor map adapted to the smaller range (about 150nm instead of 300nm) by selecting different center wavelengths of the R,G,B in the processing. I want to see if it produces a better contrast.

b) Similarly, what happens if we just take the emission maximum from the spectrum at each pixel and use these values to build an image? (autoscaling in a proper range)

2) Figure 2 and relative description:

Here the authors test the SEER with a simulated dataset.

a) for Suppl Fig 1, I ask the same tests as at points 1a,b

b) the description of the dataset in Supplementary fig 1 is quite confusing:

-which is middle and which is center?

-which is the spectrum kept unchanged from square to square?

-why the labels S1,S2,S3 in the figure have changing colors?

Please clarify

c) "The radial map instead, focuses on intensity changes, as intensity changes for the same spectrum generally result in radial shifts on the phasor plot. As a result, this map highlights spectral amplitude and magnitude, and is insensitive to wavelength changes"

This point should be explained better.

In this example, what we see is related to intensity changes or to the particular shapes of the spectra?

Why the grayscale image shows differences in intensity?

Is a less modulated value corresponding to a broader spectrum?

It is affected by the uncorrelated background as in FLIM?

3) SEER on the second SHTC (Supplementary Fig. 4-7)

a) Here the authors should do the same tests I asked before.

b) "with frequently encountered level of overlap (Supplementary Fig. 4)"

In SHTC II , the authors changed the shift between spectra from 2nm to 3nm, is that correct? Why only 1 nm?

4) Figure 4, mouse specimen:

a) What is the average spectrum of this specimen? It should be shown to understand what is the interesting range of emission

b) same tests as 1a,b adapted to the 'interesting' range for this sample

c) "The Gradient Descent morphed map (Figure 4b) enhances the visualization of metabolic activities within the tracheal specimen, showing different metabolic state..."

The results are very interesting. Is it a blue shift of the emission? Or a red shift? I understand that the map is more general than a simple blue or red shift but I think it is important, in a quantitative direction, trying to interpret the observed variations with changes in spectra of NADH (or other fluorophores). Is it what is expected based on previous reports?

d) FLIM data: "...where cells in apical layer exhibit a more Oxidative Phosphorylation phenotype (in red) compared to cells in basal layer (in yellow) with a more Glycolytic phenotype"

Here the FLIM is used to confirm spectral variations. The interpretation of the FLIM data in terms

of metabolic fingerprint is not so straightforward to the non-experts. I suggest they add at least a reference or explain why the lifetime data (e.g. longer lifetime in the apical layer) confirm spectral variations

5) Figure 5 : zebrafish label + autofluorescence

a) here we have a fluorophore in the 570nm range + broad autofluorescence. It would be nice to see the average spectrum of the sample

b) Same test as 1a,b also for this sample

6) Fig. 6: zebrafish with Cerulean -GFP- citrine (emission 470 to 550) + pigment autofluorescence

a) show spectrum

b) Same test described in point 1a,b also for this sample

7) Fig.7: "The results (Figure 7) highlight the difficulty of visualizing these datasets with standard approaches as well as how the compressive maps simplify the distinction of both spectrally close and more separated recombinations"

a) This is true for some cells. For others, for instance those in the upper left corner close to the arrow, the TrueColor map (a) gives a better separation than the the SSER map in c or d. Why?

b) Also here I would test points 1a,b as an alternative to the reported TrueColor map

8) What happens if the fluorophores in the specimen have emission peaks distant apart more than half the wavelength range (more than 150nm in this case)?

Can we still use the 2nd harmonic – phasor to distinguish them?

The authors do not discuss at all why they used only the 2nd harmonic

9) abstract: "reference maps that quantitatively enhance different spectral properties"

As a mentioned before, the 'quantitative' aspect could be better exploited. For instance the phase information could be translated into nanometers. In the SEER maps it would be nice to associate what I see in a given color with some spectral information.

Minor comments

1) Introduction:

"Phasor analysis is generally done by exploring the 2d-histogram of spectral fingerprints by means of geometrical selectors"

This is just the 'exploring' feature of Phasor analysis. I agree with the authors that with phasors can be done more than that.

There are also many examples of direct quantitative calculations starting from g and s values. For instance, the works of Scipioni et al (Biophys J 2006, Communications Biol 2008) are not using geometrical selectors but calculate maps directly from the g and s values

2) Standard Reference maps:

It should be noted that what the authors call 'Angular' and 'Radial' maps are very common maps in the field of Frequency domain FLIM, as they represent the 'Phase' and 'Modulation' images. They are normally visible in the softwares for Frequency domain FLIM (for instance ISS Vista).

Besides FLIM, examples of phase/modulation images can be found at:

Scipioni et al, Communications Biol 2018 (show phase images)

Sarmiento et al, Nat Comm 2018 (show modulation and phase images)

The authors should highlight more what is absolutely new in this paper, i.e. the more complex maps (Gradient, Morphing etc) generated in the phasor plot

3) Fig.7b : MIP or section?

4) Methods: The value of the deviation parameter sigma is not reported. How is it set?

Reviewer #2 (Remarks to the Author):

The authors report an innovative visualization method that builds on the spectral phasor approach. They designed a set of reference maps/modes to visualize simultaneously multiple spectral components characterizing the analyzed samples. These maps allow to highlight different spectral properties of the original acquired hyperspectral images. Therefore, by exploiting their reference maps, they can obtain a colored image in which each color is related to a different characteristic of the acquired spectral images.

In the first part of the manuscript the authors provide a description of the spectral phasor approach and the proposed method. Reference color maps and modes are described and tested under different conditions explaining the spectral characteristic that can be highlighted in the acquired spectral images.

The authors applied their method to multiple biological problems. At first they visualized the autofluorescence signal primed by two photon excitation in a mouse tracheal explant, showing the different results that can be obtained by exploiting the Gradient Descent morphed map or the tensor map. They concluded that the first map is able to visualize the metabolic activity, while the second can increase the contrast of cell boundaries.

In a second application, the authors exploit the angular map to discriminate different autofluorescence signals from the mKO2 protein fluorescence in zebrafish samples.

In a third example, the authors also show how their reference maps can be exploited to visualize different spectral properties in a triple labeled zebrafish embryo. They report that their method is useful in discerning the difference between multiple fluorophores and autofluorescence both in single plane and in volumetric visualizations.

Finally, the authors exploit their method to highlight spectral differences due to the recombination of CFP, YFP and RFP labels in a Zebrafish sample.

The manuscript shows an interesting method to visualize separate fluorescence spectra characterized by different properties. The authors report essentially a discussion of their method together with its validation by means of simulations and in standard biological samples. The potentially more interesting biological application in Zebrafish samples is not explored in details. The phasor approach has been exploited for multiple quantitative applications in spectral, lifetime, magnetic resonance imaging and second harmonic generation signal analysis. Although a change of color maps can be useful to visualize some details in the samples, I think that the major drawback of the manuscript is the lacking of quantitative evidence, which could improve the impact of the work.

I suggest to the authors to take into account and discuss some major points:

- Is their method robust with respect to typical problems of in-vivo microscopy, such as samples movement in the xy or xyz planes? It would be interesting a discussion of this problem. How can their method take into account sample movements? Does it produce an effect on the phasor plot that can be corrected?

- Which is the effect of the photobleaching in their visualization? If they acquire images as a function of time in presence of this effect, is it possible that the reconstructed image highlight this effect and not other spectral differences (for example related to combinatorial expression of different proteins) among the fluorophores? Is there a color map which can take into account this effect among those they designed? A discussion of this problem could be helpful for in-vivo time-lapse imaging.

In Supplementary Table 1 the authors report a power higher than that used in their previous work (Supplementary Table 3, "Hyperspectral phasor analysis enables multiplexed 5D in vivo imaging" Nature Methods 2017). Is the photobleaching more relevant in the applications shown in this manuscript?

- The number of acquired photons decreases rapidly while going deeper into the sample and therefore the fluorescence spectra (especially in the blue region of the visible spectrum) are characterized by lower signal to noise ratio (S/N). Is it possible that the color maps highlight this effect instead of other spectral difference related to the considered biological problems? Is it possible that their method assigns the same color to pixels with low S/N in a superficial plane (due for example to a low protein expression) and to pixels in a deeper plane? In this case the spectra would be depicted with the same color but the low S/N of each spectrum is related to two different effects. Could this eventual problem give rise to wrong tissue/cell segmentation and classification of the associated properties? Is it possible to correct this possible effect due to in-depth imaging? Maybe they could show the phasor plots and maps for different z-planes and discuss this eventual effect for different color maps/modes.

Is it possible that their color maps introduce some errors in cells segmentation and tracking along the z direction?

In my opinion the major concern about the manuscript is that the authors provide only a visualization comparison among different color maps/modes. The authors should discuss in more detail which parameters can be extracted from the images obtained by applying the different color maps and show some quantitative results. Quantitative information allowing to reach a deeper understanding of the biological problems with respect to other unmixing/color maps methods (for example Ref. 26, 29, or hue vs saturation plot in Ref.31) could greatly enhance the impact of their method.

Moreover, a comparison of the images and the information that can be retrieved by means of the color maps/modes shown in this manuscript and the unmixing method of their previous work (Ref. 29) should be interesting.

Minor point:

-In Scale and Morph modes, the apex of the map shifts and changes the reference center of the color palette depending on the dataset. Are therefore these modes useful only when applied to static images?

Review of Visualization of hyperspectral fluorescent data with Spectrally Encoded Enhanced Representations (SEER)

Authors: Wen Shi et al

This article describes an enhancement to the visualisation of a hyperspectral image processing data produced by a technique called SEER, which spectrally decomposes fluorescence hyperspectral images into spectral components to facilitate visualisation of multiple fluorophores in a single HSI image. The main contribution of this paper seems to be in the mapping of SEER data onto a standard color representation with enhanced contrast between fluorophores. The primary question is whether this is sufficient in itself, if done well, to justify publication in NatComm. The ancillary question is whether the work and the assessment of the results is sufficient. I believe that the manuscript as it stands is marginal on both counts, but could be sufficiently improved, particularly in terms of clarity and rigour to justify publication.

SEER may be considered as an alternative to established spectral decomposition techniques, such as PCA, (or better, ICA), but has the advantage of simplicity and computational speed – this is the major claimed advantage. While fast processing should certainly be true, speed is not quantified absolutely or relatively – and this is an important omission for would-be users. An increase in processing speed compared to a traditional technique (such as ICA), which may already be implemented quite easily and computed quickly (<1s) using standard software such as ENVI, might not be that important. Conversely a modest increase in speed of a slow process (minutes) could be important. Absolute processing time and improvement in processing time are both important and neither are quantified. Nevertheless the main narrative of this paper seems to be visualisation of SEER data rather than SEER itself.

The fundamental principles of SEER are described in ref 26 by Fereidouni, Bader and Gerritson of Utrecht (Optics Express, 2012) and its application to in vivo imaging of zebra fish was described in ref 27 (Nature Methods by the authors of this manuscript) in 2017. The first four pages of the manuscript describes the principles of SEER covering ground previously described in references 26 and 27 without clearly highlighting the new contribution compared to ref 27. There is additional new material as described below but the novelty of this new material is not clear.

The manuscript goes on to describe the main contribution: maps of the SEER data onto RGB colour spaces including mechanisms for spectral contrast enhancement. The techniques seem to be largely heuristic and assessed qualitatively using striking fHSI images. The manuscript demonstrates the application of these techniques for various samples such as 2p fluorescence of mouse trachea and multiply labelled zebrafish. There are several claims of enhanced performance, and also claims of confirmation by FLIM images, but given the lack of rigour in the comparison and uncontrolled nature of the biological samples this seems to be a rather weak validation and the confirmation by FLIM is not clear to this reviewer.

There are various comparisons with *TrueColor*, but the reason for this is not clear since the main contribution of the paper seems to be about enhanced visualisation of SEER images

rather the enhancement provided by SEER compared to traditional techniques (which has been reported previously by the authors).

Overall, the main new contribution claimed by this paper seems to be about the mapping of the SEER data for visualisation – although I found that the narrative lacked clarity overall. The manuscript presents some striking results and there are indications and claims of enhanced ability to discriminate and identify overlapping fluorophores that may be true, however the analysis presented did not demonstrate emphatically and quantitatively that the functionality of the data was improved in a way that could not be achieved by various other contrast-enhancing color mappings of SEER data, or indeed of ICA data, and this may discourage readers from investing in the implementation of these techniques. My main concern is that enhanced contrast, as apparent in many images, does not equate to enhanced functionality or (accuracy of) discrimination of fluorophores. For example, a simple increase in contrast in some spectral space while also increasing scatter within a cluster, or amplifying noise, might be visually appealing without giving more accurate information or functionality. This paper might achieve a real enhancement that is more than aesthetic, but this is not clearly demonstrated. A quantitative analysis of detectivity of features, or of spectral clustering and classification accuracy would be more convincing to would-be adopters.

Point by point response

- Reviewer's points
- "manuscript text"
- Modifications in the manuscript highlighted.

Reviewer #1 (Remarks to the Author):

In the manuscript "Visualization of hyperspectral fluorescent data with Spectrally Encoded Enhanced Representations (SEER)" by W. Shi, D.E.S. Koo et al, the authors describe a new method to enhance the representation of hyperspectral fluorescent data. Hyperspectral imaging is an important technique that, by adding a full spectral dimension to the fluorescence microscope, allows the imaging of a large number of fluorophores (even with overlapping spectra) simultaneously.

There are several established methods to process the hyperspectral dataset after the acquisition and, for instance, extract the multiple species contained in the specimen (e.g. linear unmixing). The same group has published recently a work on Nat Methods demonstrating the advantages of a Phasor-based analysis (HySP) with respect to linear unmixing.

Here the authors are focusing on the rapid, pre-analysis visualization of the hyperspectral dataset that can be useful in guiding the users during acquisition of the data. In this context, their main goal is to visualize the information contained in the spectral dimension by means of a compressed representation in standard RGB display. In contrast to commonly used dimensionality-reduction approaches (PCA, image-fusion), the authors propose an alternative, less computationally demanding, approach based on the use of Phasors. Essentially, they build different types of colormaps related to the pixel distribution of (g,s) values (the 'phasor plot').

The authors claim that the SEER approach:

- 1) improves computational efficiency (this is a feature of the phasor method, as is FFT based)
 - 2) quantitatively enhances the analysis and interpretation of spectral information
- In particular, the second claim is supported by the analysis of simulated Testcharts and experimental measurements on mouse and Zebrafish specimens.

In all cases, they compare the SEER visualization with what is considered the 'state of the art' or 'standard' visualization (also named 'True color') to highlight the benefits of SEER. According to all these comparisons, the performances of SEER are simply

striking, in both simulations and experiments, as it greatly improves the visibility of spectral differences in the data.

In summary, the paper is well written and, if the claims stated by the authors are confirmed (see my major comments below), it represents an original and significant technical advance in the field of hyperspectral imaging. I think it is of interest and of potential impact for the growing number of users of this technique, now available not only on confocal microscopes but also on light sheet-based setups. In addition, the generation of 'smart' colormaps from the 2D phasor plot is a new idea with potential impact also in all the other contexts where the phasor can be applied.

For these reasons I think that the paper is of great interest for Nature Communications and well-worth of publication on this journal.

However, there are some fundamental points that must be addressed in order to understand the true advantages of the proposed method.

The phasor of a full spectrum is generally an approximation (g and s instead of n lambda points) that in many cases (FLIM autofluorescence, FLIM-FRET, FLIM superresolution, emission spectra etc) is sufficient for the description of the system under study. A second advantage is represented by the linear property which is the basis of the calculation of fractional components in many of these applications.

In the way the data are presented, I have the impression that the difference between the SEER and the 'standard' visualization might be exaggerated. I think it is important to understand how much of this difference is due to the phasor-based representations and how much is due to other effects (for instance the 'unused' spectral range in the TrueColor map). In addition, the 'quantitative' aspects of the phasor (e.g. the linearity that can be used to extract fractional components, etc) could be highlighted more in the paper; it would even be more attractive for the users of hyperspectral imaging to associate some numbers to the maps.

We thank the reviewer for the positive comments, the extremely on-point summary and for acknowledging the detail in the description of the manuscript. We reply point-by-point to the specific requests.

I ask the authors to address the following major concerns.

Major comments

1) One of the main results of the paper is the enhanced contrast provided by the SEER map with respect to the 'standard' or TrueColor map.

For instance let's consider Figure 1: the data are analyzed in spectral range 400-700 nm (Fig.1b) but the meaningful part of the fluorophores spectra (CFP to RFP) is in a smaller range, say 470-620nm. The TrueColor map should be adapted to this smaller range. In other words, the center wavelengths of R, G, B (which, by the way, are not reported except R=650nm) should be set closer to each other to render better the variation in the spectra.

In fact I also note that the phasor map (the SEER map) is constructed, in all the data, from the 2nd harmonic of the spectra. This mean that the phase of the phasor will highlight a variation of the emission maxima in the range $(700-400\text{nm})/2=150\text{nm}$. I think a more fair comparison should be made between the SEER map and the 'standard' maps. For this reason I ask to perform the following tests:

a) Visualize all the data in the Figures with a TrueColor map adapted to the smaller range (about 150nm instead of 300nm) by selecting different center wavelengths of the R,G,B in the processing. I want to see if it produces a better contrast.

b) Similarly, what happens if we just take the emission maximum from the spectrum at each pixel and use these values to build an image? (autoscaling in a proper range)

We thank the reviewer for this suggestion. We realize the TrueColor rendering was not properly defined, and it is correct to show different adaptations of this "standard" modality.

We have added a more proper definition of the Standard RGB Visualizations including those suggested (in this and other questions) by this reviewer. We believe this is a fair comparison worth including in the manuscript, and we thank the reviewer for the suggestion. We have updated the labels on the corresponding main text figures.

We utilize the following definitions for computing the images requested by this reviewer.

Methods section now reads:

"Standard RGB Visualizations

The TrueColor RGB image (Supplementary Figure 3, 8, 15) is obtained through compression of the hyperspectral cube into the RGB 3 channels color space by generating a Gaussian radial basis function.. [...] .. Gaussian kernel was set at 650nm, 510nm, 470nm for RGB respectively as Default (Supplementary Figure 3s, Supplementary Figure 8, Supplementary Figure 13e, Supplementary Figure 16e, Supplementary Figure 17e, Supplementary Figure 19j).

The same Gaussian kernel was also changed adaptively to the dataset to provide a spectral contrast stretching on the visualization and focus the visualization on the most utilized channels. The average spectrum for the entire dataset is calculated and normalized. The intersect at 10%

(Supplementary Figure 13f, 16f, 17f, 19f), 20% (Supplementary Figure 13g, 16g, 17g, 19g) and 30% (Supplementary Figure 13h, 16h, 17h, 19h) of the intensity is obtained and used as a center for the blue and red channels. The green channel is centered halfway between red and blue. Representations of these adaptations are reported in Supplementary Figure 20g,h,i.

The TrueColor 32 Channels image (Figure 1c, Figure 5a,c,e,g, Figure 6a,d,e,f, Supplementary Figure 13c, 16c, 17c, 19c) was rendered as a 32 channels Maximum Intensity Projection using Bitplane Imaris (Oxford Instruments, Abingdon, UK). Each channel has a known wavelength center (32 bins, from 410.5 nm to 694.9 nm with 8.9 nm bandwidth). Each wavelength was associated with a color according to classical wavelength-to-RGB conversions [D. Bruton Approximate RGB Values for Visible Wavelengths (1996)] as reported in Supplementary Figure 20f. The intensity for all channels was contrast-adjusted (Imaris Display Adjustment Settings) based on the channel with the largest information. A meaningful range for rendering was identified as the top 90% in intensity of the normalized average spectrum for the dataset (Supplementary Figure 13b, 16j, 17j, 19b). Channels outside of this range were excluded for rendering. Furthermore, for 1 photon excitation, channels associated to wavelengths lower than the laser excitation (for example channels 1 to 5 for laser 458nm) were excluded from rendering. Peak Wavelength representation (Supplementary Figure 13d, 16d, 17d, 19d, 20 and 22) reconstructs an RGB image utilizing, for each pixel, the color associated to the wavelength at which maximum intensity is measured. Wavelength-to-RGB conversion was performed using a python function (<http://bioimaging.usc.edu/software.html#HySP>) adapted from Dan Bruton's work⁵⁹. A graphical representation is reported in Supplementary Figure 20f. “

2) Figure 2 and relative description:

Here the authors test the SEER with a simulated dataset.

a) for Suppl Fig 1, I ask the same tests as at points 1a,b

We have performed the analysis utilizing the methods described above (and now in the Methods section of the paper). We agree with the reviewer in showing a more contrast enhanced version of the figure. We have updated Supplementary Figure 3 to include the Gaussian default visualization with the appropriate contrast suggested by the reviewer. The choice of the wavelengths associated to the colors was made based on standards for eye color appearance and appears to be the same approach utilized in commercial instruments such as Zen (Zeiss) for visualization. Below we report the other approaches the reviewer suggests:

The Simulated Hyperspectral Test Chart I is represented here as (a)“Peak Wavelength Mask”, (b, h) “Gaussian Default Mask”, (c, i)“Gaussian $r = 0.1$ Mask”, (d, j)“Gaussian $r = 0.2$ Mask”, (e, k)“Gaussian $r = .2$ Mask”. (a) is a visualization method suggested by reviewer 1 which constructs an RGB image with values of the emission maximum from the spectrum at each pixel. (f) presents the normalized averaged spectrum of this simulation dataset (black line). The dashed blue and red lines show the center position for the B and R channels when a 10% threshold is applied on the normalized average intensity of the whole dataset. (b, h) was proposed in our first manuscript as “True Color”; this gaussian default setting was chosen based on how the eye interprets a multi-channel wavelength by associating to RGB coefficients to these wavelengths (Methods - Standard RGB Visualizations). (c-e, i-k) is another suggested visualization method from reviewer 1 which constrains the standard R,G,B spectrum to a “meaningful range”. The new range is adjusted by constraining the R,G,B standard spectra (Red, Green Blue Gaussian curves in plots) within 90%, 80%, 70% of the normalized average intensity.

b) the description of the dataset in Supplementary fig 1 is quite confusing:
 -which is middle and which is center?
 -which is the spectrum kept unchanged from square to square?
 -why the labels S1,S2,S3 in the figure have changing colors?
 Please clarify

We thank the reviewer for the constructive comment. We have improved both the figure labeling and legend to address the above questions. We corrected the color mismatch of S1-S3. The figure now appears as follows:

The legend for this figure (Supplementary Figure 3) has been edited and now reads:

“Supplementary Figure 3. Simulated Hyperspectral Test Chart I rendered in TrueColor shows nearly indistinguishable spectra. The simulation is represented here in “TrueColor RGB” (Methods). S_1 , S_2 , and S_3 spectra acquired respectively from CFP, YFP, RFP zebrafish embryos are used to generate a (a-i) 3-by-3 Simulated Hyperspectral Test Chart. In each panel (a-i) of the chart, three spectra (S_1 to S_3) are represented as concentric squares (see panel a) outer: S_1 - blue, intermediate: S_2 - yellow, middle: S_3 - red spectra respectively. The spectrum S_2 (intermediate square in each panel) is kept unchanged in all panels. The maximum of spectrum S_1 is shifted by d_1 (-2 wavelength bins, -17.8nm nm steps) with respect to the fixed spectrum S_2 maximum. S_3 max value is shifted by d_2 (2 wavelength bins, 17.8nm nm steps) respect to S_2 maximum. The changes are applied for 2 steps along the vertical (d_1) and horizontal (d_2) axis of the center panel assembly (a-i), starting from aligned maxima, $d_1=d_2=0$ (panel a). The spectra utilized in each panel (a-i) are represented in panels j-r. Each plot (j-r) represents the averaged normalized S_1 - S_3 spectra as 32 wavelength bins, 8.9nm bandwidth, 410-695 nm detection. Each panel has different visual contrast but is generally difficult to distinguish by eye due to significant overlap in spectra. (s) R,G,B channels used in the Gaussian Kernel for True color representation (red, green, blue lines) and average spectrum for panels (a-i) (yellow line) for reference. “

The main text has been corrected in the standard reference maps section and now reads:

“Each section of the test chart offers different image contrast, obtained by shifting the CFP and RFP spectra maxima position with respect to the YFP spectrum (Supplementary Figure 1, Methods).”

c) “The radial map instead, focuses on intensity changes, as intensity changes for the same spectrum generally result in radial shifts on the phasor plot. As a result, this map highlights spectral amplitude and magnitude, and is insensitive to wavelength changes” This point should be explained better.

In this example, what we see is related to intensity changes or to the particular shapes of the spectra?

We thank the reviewer for this question. In this example what we see is related to the particular shape of the spectra. In the more common experimental scenario, however, the radial shift is mainly affected by the decrease in intensities and the corresponding decrease of Signal to Noise Ratio. This intensity related radial shift is the result of the apparent “broadening” of spectra when SNR lowers below 2. We have added two supplementary figures to better describe this point. First figure, Supplementary Figure 4 shows the effect of spectral shape on the radial position in absence of background uncorrelated noise. The second figure, Supplementary Figure 5 shows the apparent spectral “broadening” which happens in presence of background. The corresponding captions are reported below.

We edited the main text in the Standard Reference Maps paragraphs to explain this point better. The radial map portion now reads:

“The radial position is, in general, related to the intensity integral of the spectrum, which in turn can depend on the shape of the spectrum, with the null-intensity localizing at the origin of the plot (Supplementary Figure 4). In our simulation (Figure 2c), the colors obtained with this map mainly represent differences in shape, however in a scenario with large dynamic range of intensities, colors will mainly reflect changes in intensity, becoming affected, at low signal to noise, by the uncorrelated background (Supplementary Figure 5).”

“ **Supplementary Figure 4. Effect of spectral shape with constant intensities on radial map in absence of background.** This simulation shows spectra with Gaussian shape and different standard deviations using 32 wavelength bins, 8.9nm bandwidth, and a 410-695 nm range in the absence of background. All spectra are centered on 543nm (channel 16) and the integral of intensities is kept constant. **(a-i)** For each value of the standard deviation, a grayscale image and SEER visualization are presented. The map used is the Radial map centered on the origin and extended to the border of the phasor plot. A color reference is added in the phasor plot **(m)**. Clusters on the phasor plot are distributed along the radius, with distance from the origin inversely proportional to the standard deviation. “

“Supplementary Figure 5. Effect of spectrum intensity in presence of background on radial map. In this simulation, the first panel (top-left) of the Simulated Hyperspectral Test Chart (Supplementary Figure 3) is reduced in intensity by a factor of 10^1 - 10^4 (panel 1-4 respectively) in the presence of a constant background. Background with an average intensity of 5 digital levels was generated in Matlab; poissonian noise was added using the `poissrnd()` function. **(a,d,g,j)** Grayscale images are scaled by **(a)** factor of 10, **(d)** factor of 10^2 , **(g)** factor of 10^3 , **(j)** factor of 10^4 . Radial map (original) visualization shows a shift of panel colors toward blue with decreasing intensities **(b,e,h,k)**. The phasor plots **(c,f,i,l)** (harmonic $n=2$) show a radial shift of the clusters toward the origin. Radial map reference is added in **(c)**. **(m)** Absolute intensities plot shows the average spectrum for the four panels, maximum peak values are 1780, 182, 23, 7 digital levels (panel 1-4 respectively). The normalized intensity spectra **(n)** show an apparent broadening of the shape of spectra with the decreasing signal-to-noise. “

The legend for Figure 2 has been updated, now reads:

“(c) The **radial map**, instead, focuses mainly on intensity changes, as a decrease in the signal to noise generally results in shifts towards the origin on the phasor plot. ”

Why the grayscale image shows differences in intensity?

The grayscale images are the average of the 32 spectral bands. Spectra are normalized by maximum intensity value. The total intensity under each spectrum is different. As a result, grayscale images show differences in intensity.

We have updated the legend for Figure 2 to specify the source of the grayscale image. Caption portion now reads:

“(a) The Standard phasor plot with corresponding average grayscale image provides the positional information of the spectra on the phasor plot.”

Is a less modulated value corresponding to a broader spectrum?

As we show in the images above, in general, a less modulated value corresponds to a broader spectrum. However, in the presence of background, when SNR falls below 2, spectra blend with the background, showing an apparent “broader” spectrum.

It is affected by the uncorrelated background as in FLIM?

It is affected by uncorrelated background only at low SNR. At high SNR the effect is minimal. The effect is visible in Supplementary Figure 5 (above) for SNR below 2.

We edited the main text in the radial map section of the Standard Reference Maps paragraph to add this point. Paragraph now reads:

“[...] in a scenario with large dynamic range of intensities, colors will mainly reflect changes in intensity, becoming affected, at low signal to noise, by the uncorrelated background (Supplementary Figure 5).”

3) SEER on the second SHTC (Supplementary Fig. 4-7)

a) Here the authors should do the same tests I asked before.

We thank the reviewer for this question. In answering this, we would like to point out that in this version of the simulation, the spectra are given steps of 3 channels separation, 26.7nm steps. The final block of the simulation shows spectra with peak-to-peak distance of 53.6nm, with outer and inner concentric squares well separated by 106.8nm. This distance is similar to the emission gap between CFP (475nm EM) and tdTomato (581nm). Under these conditions, most methods perform well.

A new Simulated SHTC II is represented here in (a, f) “Peak Wavelength Mask”, (b, h) “Gaussian Default Mask”, (c, i) “Gaussian $r = 0.1$ Mask”, (d, j) “Gaussian $r = 0.2$ Mask”, (e, k) “Gaussian $r = 0.3$ Mask”. The two visualization methods, “Peak wavelength mask” and “Gaussian Kernel Mask,” used here are the same as in SHTC I in our answer. Similarly represented as for SHTC I, “Peak wavelength mask” approximates the RGB components using the peak wavelength. In “Gaussian kernel mask”, the gaussian rate r acts as a threshold to constrain the interesting range suggested by the reviewer. In particular, this dataset is randomly picked pixel by pixel from the CFP, YFP, and RFP labeled samples used in SHTC I. As such, we expect each dataset to not be identical.

We have updated Supplementary Figure 8 to include a renewed version of the TrueColor RGB visualization as described in the methods section. The figure now looks like this:

Figure caption:

“Supplementary Figure 8. Simulated Hyperspectral Test Chart II and its standard overlapping spectra. Simulated SHTC II was generated from the same zebrafish embryo datasets and same design used in SHTC1 (Supplementary Figure 3) utilizing CFP, YFP, and RFP labeled samples and a 3-by-3 block chart, with each block subdivided into 3 regions corresponding to spectra S_1 , S_2 , and S_3 . The aim is to test scenarios with less overlapping spectra. We change the shifting distance in this simulation to be d_1 (-3 wavelength bins, -26.7nm steps) and d_2 (3 wavelength bins, 26.7nm steps). The channels used in the Gaussian Kernel for TrueColor RGB representation here were 650nm, 510nm, 470nm which respectively represent R, G, B. The concentric squares in the lower right side of the simulation are separated by a peak-to-peak distance of 53.6nm, with outer and inner concentric squares well separated by 106.8nm. This distance is similar to the emission gap between CFP (475nm EM) and tdTomato (581nm). Under these spectral conditions most methods are expected to perform well. “

b) “with frequently encountered level of overlap (Supplementary Fig. 4)”
In SHTC II , the authors changed the shift between spectra from 2nm to 3nm, is that correct? Why only 1 nm?

We thank the reviewer for pointing this out. The shift unit was in channels, not nanometers. The simulation utilizes spectra acquired from a Zeiss LSM780 with Quasar detectors (32 wavelength bins, 8.9nm bandwidth, 410-695 nm). The minimum shift possible (without interpolation) was 1 channel (8.9nm). Hence, the shift in SHTC II was 3 wavelength bins, 26.7nm steps, compared to SHTC I with 2 wavelength bins, 17.8nm. Considering the CFP spectrum has a width of ~79nm, SHTC I has a shift of 22%-66% and SHTC II of 33%-99% of the CFP spectral width. The aim here was to show SEER visualization performance at different overlap levels, focusing both on close overlap (SHTC I) and little overlap (SHTC II).

We have corrected the caption for Supplementary Figure 8 which now reads:

“ The aim is to test scenarios with less overlapping spectra. We change the shifting distance in this simulation to be d1 (-3 wavelength bins, -26.7nm nm steps) and d2 (3 wavelength bins, 26.7nm nm steps). “

4) Figure 4, mouse specimen:

- a) What is the average spectrum of this specimen? It should be shown to understand what is the interesting range of emission
- b) same tests as 1a,b adapted to the ‘interesting’ range for this sample

The answer to point 4a and 4b is shown in the figure below, Supplementary Figure 13. Other visualization approaches do not show any appreciable improvement.

“Supplementary Figure 13. Visualization comparison for autofluorescence with other RGB standard visualizations. The visualization of unlabeled freshly isolated mouse tracheal explant (Figure 4) is shown here with different standard approaches. Details for these visualizations are reported in the Methods section. (a) SEER RGB mask obtained using gradient descent morphed map; this mask shows the colors associated by SEER to each pixel, without considering intensity. (b) Average spectrum for the entire dataset. (c) TrueColor 32 channels maximum intensity projection. (d) Peak wavelength RGB mask. (e) Gaussian Default Kernel with RGB centered respectively at 650nm, 510nm and 470nm. (f) Gaussian Kernel at 10% threshold, RGB values centered at 659nm, 534nm and 410nm. (g) Gaussian Kernel at 20% threshold, RGB values centered at 570nm, 490nm and 410nm. (h) Gaussian kernel at 30% threshold, RGB values centered at 543nm, 472nm and 410nm. (i) wavelength-to-RGB color representation for Peak Wavelength mask in panel d. A representation of the RGB visualization parameters is reported in (j) kernel used for panel e, average spectrum of the dataset (yellow plot), (k) kernel used for panel f, average spectrum of the dataset (yellow plot), (l) kernel used for panel g, average spectrum of the dataset (yellow plot), (m) kernel used for panel h, average spectrum of the dataset (yellow plot).”

We then referenced this supplementary figure in the main text:

“The visualization improvement is maintained against different implementations of RGB visualization (Supplementary Figure 13). “

c) “The Gradient Descent morphed map (Figure 4b) enhances the visualization of metabolic activities within the tracheal specimen, showing different metabolic state...” The results are very interesting. Is it a blue shift of the emission? Or a red shift? I understand that the map is more general than a simple blue or red shift but I think it is important, in a quantitative direction, trying to interpret the observed variations with changes in spectra of NADH (or other fluorophores). Is it what is expected based on previous reports?

The sample exhibits, in general, a blue shift in the direction of the apical layer.

We have updated both figure 4 and its captions to simplify interpretation of the metabolic fingerprint.

Caption has been updated for panels d and e and now reads:

“(d) Average spectra for the cells in dashed boxes (1 and 2 in panel c) show a blue spectral shift in the direction of the apical layer.”

The tracheal epithelium comprises of 3 main cell types: basal, club, and ciliated cells. Since its turnover is small and slow, differences in the metabolic status are attributable to the specific role of the cell type. Club and ciliated cells, which are located in the apical layer are the most active cells in the tracheal epithelium. Club cells constantly secrete numerous cytokines such as Cc10 and Sftpb and chemically neutralize inhaled xenobiotics (<https://www.ncbi.nlm.nih.gov/pmc/articles/PMC4860431/>). On the other hand, ciliated cells are constantly clearing up the airways from mucus and inhaled debris and particles (<http://www.pnas.org/content/104/2/410>). Basal cells are quiescent cells located on the basal portion of the tracheal epithelium, in direct contact with the base membrane. Basal cells are only activated during the regeneration of the tracheal

epithelium, during which they quickly proliferate and differentiate into club and ciliated cells [37, 38] (<https://www.ncbi.nlm.nih.gov/pmc/articles/PMC4229034/>) . In our experiments, we used tracheal tissue from uninjured wild mice that haven't been exposed to any chemicals or toxins. Our FLIM results are in perfect agreement with the diversity in both biological activity and function of tracheal epithelial cells: apical club and club cells are indeed metabolically more engaged than basal cells.

We have edited the main text in the autofluorescence section to clarify this aspect. The text now reads:

“SEER is applied here for visualizing multispectral autofluorescent data of an explant of freshly isolated trachea from a wildtype C57Bl mouse. The tracheal epithelium is characterized by a very low cellular turnover, and therefore the overall metabolic activity is attributable to the cellular function of the specific cell type. Club and ciliated cells are localized in the apical side of the epithelium and are the most metabolically active as they secrete cytokines and chemically and physically remove inhaled toxins and particles from the tracheal lumen. Contrarily, basal cells which represent the adult stem cells in the upper airways, are quiescent and metabolically inactive [37, 38]. Because of this dichotomy in activity, the tracheal epithelium at homeostasis constituted the ideal cellular system for testing SEER and validating with FLIM imaging. The slight bend on the trachea caused by the cartilage rings allowed us to visualize the mesenchymal collagen layer, the basal and apical epithelial cells and tracheal lumen in a single focal plane. “

d) FLIM data: “..where cells in apical layer exhibit a more Oxidative Phosphorylation phenotype (in red) compared to cells in basal layer (in yellow) with a more Glycolytic phenotype”

Here the FLIM is used to confirm spectral variations. The interpretation of the FLIM data in terms of metabolic fingerprint is not so straightforward to the non-experts. I suggest they add at least a reference or explain why the lifetime data (e.g. longer lifetime in the apical layer) confirm spectral variations

We have updated both figure 4 and its caption to simplify interpretation of the metabolic fingerprint. The figure:

Caption has been updated for panels d and e and now reads:

“(d) Average spectra for the cells in dashed boxes (1 and 2 in panel c) show a blue spectral shift in the direction of the apical layer. (e) Fluorescence Lifetime Image Microscopy (FLIM) of the sample, acquired using a frequency domain detector validates the interpretation from panel b, Gradient Descent Map, where cells in the apical layer exhibit a more Oxidative Phosphorylation phenotype (longer lifetime in red) compared to cells in the basal layer (shorter lifetime in yellow) with a more Glycolytic phenotype. The selections correspond to areas selected in phasor FLIM analysis (e, top left inset, red and yellow selections) based on the relative phasor coordinates of NAD⁺/NADH lifetimes. “

In an effort to further clarify the FLIM procedure, we have edited the corresponding Main text and Methods section.

Main Text now reads:

“Changes in autofluorescence inside live samples are associated to variations in the ratio of NAD⁺/NADH, which in turn is related to the ratio of free to protein bound NADH³⁹. Despite very similar fluorescence emission spectra, these two forms of NADH are characterized by different decay times (0.4ns free and 1.0-3.4 ns bound)^{35,40-44}. FLIM provides a sensitive measurement for the redox states of NADH and glycolytic/oxidative phosphorylation. Metabolic imaging by FLIM is well established and has been applied for characterizing disease progression in multiple animal models, in single cells, and in human as well as to distinguish stem cell differentiation and embryo development⁴¹⁻⁴⁸.

Previous work has shown that both Hyperspectral Imaging and FLIM correlate with metabolic changes in cells from retinal organoids⁴⁹. Here, the dashed squares highlight cells with distinct spectral representation through SEER, a difference which the FLIM image (Figure 4d, Supplementary Figure 14) confirms.”

The methods section now reads:

“FLIM analysis of intrinsic fluorophores was performed as previously described and reported in detail^{23,35,43}. Phasor coordinates (g,s) were obtained through Fourier transformations. Cluster identification was utilized to associate specific regions in the phasor to pixels in the FLIM dataset according to published protocols³⁵ “

Where [35] is the Gratton’s group protocol for “Fit-free analysis of fluorescence lifetime imaging data using the phasor approach” (Ranjit et al, Nature Protocols 13, 1979–2004 (2018)).

Supplementary Figure 14 has been updated:

Supplementary Figure 14 caption has been updated and now reads:

“(c) The line joining free and bound NADH in the phasor plot is known as the “metabolic trajectory”, and a shift in the free NADH direction is representative of a more reducing condition and a glycolytic metabolism, while a shift towards more bound NADH is indicative of more oxidizing conditions and more oxidative phosphorylation, as described in previous studies^{2–5}.

The extremes of the metabolic trajectory are the lifetimes for NADH free and bound. The parameters for lifetime (τ phase and modulation) are in line with those reported in literature (0.4ns free and 1.0-3.4 ns bound)^{3–8}.

”

Where:

[4] is Sharick et al, “Protein-bound NAD(P)H Lifetime is Sensitive to Multiple Fates of Glucose Carbon” Scientific Reports 8, 5456 (2018)

[2] is Browne et al, “Structural and functional characterization of human stem-cell derived retinal organoids by live imaging”. Invest Ophthalmol Vis Sci. 2017; 58:3311–3318

[6] is Lakowicz, J. R., et al. Fluorescence lifetime imaging of free and protein-bound NADH. Proc Natl Acad Sci USA 89, 1271–1275 (1992)

[8] is Melissa C. Skala, et al, “ In vivo multiphoton microscopy of NADH and FAD redox states, fluorescence lifetimes, and cellular morphology in precancerous epithelia Proc Natl Acad Sci USA 2007, 104 (49) 19494-19499

[7] is Stringari et al, “ Multicolor two-photon imaging of endogenous fluorophores in living tissues by wavelength mixing”, Scientific Reports volume 7, Article number: 3792 (2017)

[3] is Stringari et al, “Phasor approach to fluorescence lifetime microscopy distinguishes different metabolic states of germ cells in a live tissue”, PNAS August 16, 2011 108 (33) 13582-13587

[5] Ranjit et al, Fit-free analysis of fluorescence lifetime imaging data using the phasor approach, Nature Protocols 13, 1979-2004 (2018)

5) Figure 5 : zebrafish label + autofluorescence

a) here we have a fluorophore in the 570nm range + broad autofluorescence. It would be nice to see the average spectrum of the sample

b) Same test as 1a,b also for this sample

The answers to 5a and 5b are reported in the figure below, Supplementary Figure 16. Other visualization approaches do not show any appreciable improvement.

“Supplementary Figure 16. Visualization comparison for single fluorescent label with other RGB standard visualizations in presence of autofluorescence. Visualization of *Tg(fli1:mKO2)* (pan-endothelial fluorescent protein label) zebrafish with intrinsic signal arising from the yolk and xanthophores (pigment cells) (Figure 5) is here shown with different standard approaches. Details for these visualization are reported in the Methods section. (a) SEER RGB mask for a single z-plane, obtained using gradient angular map in scaled mode, this mask shows the colors associated by SEER to each pixel, without considering intensity. (b) SEER maximum intensity projection (MIP) for the entire volume (c) TrueColor 32 channels volume MIP (d) Peak wavelength volume MIP. (e) Gaussian Default Kernel with RGB centered respectively at 650nm, 510nm and 470nm. (f) Gaussian Kernel at 10% threshold, RGB values centered at 686nm, 588nm and 499nm. (g) Gaussian Kernel at 20% threshold, RGB values centered at 668nm, 579nm and 499nm. (h) Gaussian kernel at 30% threshold, RGB values centered at 641nm, 570nm and 499nm. (i) wavelength-to-RGB color representation for Peak Wavelength mask in panel d. A representation of the RGB visualization parameters is reported in (j) Average spectrum (blue plot) for the entire dataset with boundaries used for TrueColor 32ch MIP in panel c. (k) kernel used for panel e, average spectrum of the dataset (yellow plot), (l) kernel used for panel f, average spectrum of the dataset (yellow plot), (m) kernel used for panel g, average spectrum of the dataset (yellow plot), (n) kernel used for panel h, average spectrum of the dataset (yellow plot).”

6) Fig. 6: zebrafish with Cerulean -GFP- citrine (emission 470 to 550) + pigment autofluorescence

a) show spectrum

b) Same test described in point 1a,b also for this sample

The answers to 6a and 6b are reported in the figure below, Supplementary Figure 22. Other visualization approaches do not show any appreciable improvement.

“**Supplementary Figure 17. Visualization comparison for triple label fluorescence with other RGB standard approaches.** Visualization of *Tg(kdrl:eGFP); Gt(desmin-Citrine); Tg(ubiq:H2B-Cerulean)* labelling respectively vasculature, muscle, and nuclei (Figure 6) is shown here with different standard approaches. Details for these visualization are reported in the Methods section. The same slice (here $z=3$) is shown as a maximum intensity projection (MIP) using: (a) SEER gradient descent map in max morph mode, (b) SEER MIP angular map mass morph mode, (c) TrueColor 32 channels, (d) Peak wavelength, (e) Gaussian Default Kernel with RGB centered respectively at 650nm, 510nm and 470nm. (f) Gaussian Kernel at 10% threshold, RGB values centered at 597nm, 526nm and 463nm (g) Gaussian Kernel at 20% threshold, RGB values centered at 579nm, 517 and 463nm (h) Gaussian kernel at 30% threshold, RGB values centered at 561nm, 526nm and 490nm. A representation of the RGB visualization parameters is reported in (i) wavelength-to-RGB color representation for Peak Wavelength mask in panel d, (j) Average spectrum (blue plot) for the entire dataset with boundaries used for TrueColor 32ch MIP in panel c. (k) kernel used for panel e, average spectrum of the dataset (yellow plot), (l) kernel used for panel f, average spectrum of the dataset (yellow plot), (m) kernel used for panel g, average spectrum of the dataset (yellow plot), (n) kernel used for panel h, average spectrum of the dataset (yellow plot).”

7) Fig.7: “The results (Figure 7) highlight the difficulty of visualizing these datasets with standard approaches as well as how the compressive maps simplify the distinction of both spectrally close and more separated recombinations”

a) This is true for some cells. For others, for instance those in the upper left corner close to the arrow, the TrueColor map (a) gives a better separation than the the SSER map in c or d. Why?

The reason for this difference is the intensity of the spectra and the rendering/visualization modality we used. Particularly, Figure 7 panels (c) and (d) are SEER RGB masks, the maps of RGB values that SEER associates in the image. These RGB masks do not take into account the intensities in the real data; instead they highlight difference in spectra. The (a) and (b) panels are Alpha Color renderings, which take into account the intensities in the data and associate a “brightness” to the color in pixels.

In this case, the three spectra are very similar but differ in intensity. The result is that SEER shows very similar spectra in the RGB Mask (c and d, figure 7 main manuscript) (which does not account for intensity) but shows different “brightness” of the same color in the Alpha color renderings (b, figure 7 main manuscript).

In the figure below we show a zoom in of the area the reviewer refers to (from figure 7 main paper). The major difference between the TrueColor and SEER appeared in the cells highlighted in (c). The spectra are reported in (a) absolute values and (b) normalized. The (d) SEER alpha color rendering shows different “colors” for the 3 selections as it accounts for intensities. The (e) SEER rgb mask shows very similar colors (slightly darker for 1 and 2 compared to 3) as the choice of map, in this case, was the gradient descent, where phase is used as a discriminator. Intensity changes will mainly shift modulation with minimal phase change.

The other cells in the upper area of the image (panel c, d, e below) appear to be well separated in color.

However, we understand the confusion and took the following steps to improve this point in the manuscript.

We edited figure 7 to specify the RGB Mask:

The caption reads:

“Figure 7. Visualization of combinatorial expression on Zebrafish³¹ samples. Maximum Intensity Projection renderings of *Tg(ubi:Zebrafish)* muscle acquired live in multi-spectral

confocal mode with 458nm excitation. (a) The elicited signal (e.g. white arrows) is difficult to interpret in the TrueColor Image display (Zen Software, Zeiss, Germany). (b) Discerning spectral differences is increasingly simpler with Gradient Descent map scaled to intensities, while compromising on the brightness of the image. (c) Gradient Descent and (d) Gradient Ascent **RGB Masks** in scale mode **show the color values assigned to each pixel and** greatly improve the visual separation of recombined CFP, YFP and RFP labels.”

b) Also here I would test points 1a,b as an alternative to the reported TrueColor map

The caption reads:

“**Supplementary Figure 19. Visualization comparison for combinatorial expression with other RGB standard approaches.** Visualization of *Tg(ubi:Zebrawow)* muscle (Figure 7) with different standard approaches. Details for these visualization are reported in the Methods section. The same slice is shown as an RGB mask which represents the color associated to each pixel, independent from the intensity, or as a maximum intensity projection (MIP) using: (a) SEER gradient descent map mask in scaled mode, (b) Average spectrum (blue plot) for the entire dataset with boundaries used for TrueColor 32ch MIP in panel c. (c) TrueColor 32 channels, (d) Peak wavelength mask, (e) Gaussian Default Kernel with RGB centered respectively at 650nm, 510nm and 470nm. (f) Gaussian Kernel at 10% threshold, RGB values centered at 659nm, 561nm and 463nm. (g) Gaussian Kernel at 20% threshold, RGB values centered at 641nm, 552nm and 463nm. (h) Gaussian kernel at 30% threshold, RGB values centered at 632nm, 552nm and 472nm. (i) wavelength-to-RGB color representation for Peak Wavelength mask in panel d. A representation of the RGB visualization parameters is reported in (j) kernel used for panel e, average spectrum of the dataset (yellow plot), (k) kernel used for panel f, average spectrum of the dataset (yellow plot), (l) kernel used for panel g, average spectrum of the dataset (yellow plot), (m) kernel used for panel h, average spectrum of the dataset (yellow plot). “

8) What happens if the fluorophores in the specimen have emission peaks distant apart more than half the wavelength range (more than 150nm in this case)?

Can we still use the 2nd harmonic – phasor to distinguish them?

The authors do not discuss at all why they used only the 2nd harmonic

We agree with the reviewer that a more proper description of the harmonic choice is necessary for a wider adoption of the method.

To answer the question: if emission peaks are 150nm apart or more (example CFP EM=475nm, mKate2 EM= 634) 2nd harmonic will not perform well. In such a scenario, the 1st harmonic should be utilized. The reason we utilized the 2nd harmonic in this work is dual:

1. If the two fluorophores are 150 nm apart (blue-red) and so, well separated, it is generally easy to visually tell them apart. The necessity of multispectral to tell the two fluorophores apart may be less evident. The complexity of visualization is more common at shorter peak-to-peak distances.
2. In most scenarios we encounter it is preferable to utilize a single wavelength excitation with a single band dichroic to avoid dips in the emission spectra and optimize fluorescence signal collection. For example, the standard dichroics in Zeiss microscopes have a set of 2 band dichroic (458-514) and one of 3 band (458-514-594). These dichroics will have ~15-20 nm dips in the transmittance spectrum, causing the emission fluorescent spectra to be considerably hampered. We are attaching a transmittance spectrum for such a dichroic.

With a single wavelength excitation, multiple fluorescent proteins can be excited, as we report in our figures here and previous work (example CFP, citrine, GFP, autofluorescence).

We are in the process of preparing a separate protocol paper which will provide the scientific community with a more proper step-by-step description of these details and of the multispectral live imaging knowledge we have acquired in the past years.

We edited the paragraph at page 10 to include:

“different levels of spectral overlap in the simulations and for different harmonics. The supplement presents results for SHTC with very similar spectra (Supplementary Figure 3), using second harmonic in the transform (Methods), and for an image with frequently encountered level of overlap (Supplementary Figure 8) using first harmonic. In both scenarios, SEER improves visualization of multispectral datasets (Supplementary Figure 6,7,9,10) compared to standard approaches (Supplementary Figure 3,8). A detailed description of the choice of harmonic for visualization is presented in Supplementary Note 1. Implementation of 1x to 5x spectral denoising filters²⁹ further enhances visualization (Supplementary Figure 11,12).”

We edited the Methods section, adding :

“The effects of different harmonic numbers on the SEER visualization are reported in (Supplementary Note 1)”

We edited the legend of Supplementary Figure 6 to specify the harmonic used:

“Here second harmonic is utilized for the calculations. “

Likewise we edited the legend of Supplementary Figure 7:

Here second harmonic is utilized for SEER.

We updated Supplementary Figure 9 and 10 to use the first harmonic:

“Supplementary Figure 9.[...] Here first harmonic is utilized for SEER [...]”

“Supplementary Figure 10. [...] Here first harmonic is utilized for SEER [...]”

We prepared a Supplementary Note that describes the process of choice for harmonics:

“ Supplementary Note 1 Choice of harmonic for visualization

The distribution of spectral wavelengths on the phasor plot is highly dependent on the harmonic number used. Typically, the first and second harmonics are utilized for obtaining the hyperspectral phasor values due to visualization limitations imposed by branching within the Riemann surfaces in complex space¹.

The first harmonic results in a spectral distribution which approximately covers $3/2\pi$ radians for spectrum within the visible range (400 nm – 700 nm), within the universal circle, along a counterclockwise path. As a result spectra separated by any peak-to-peak distance will appear in different positions on the phasor plot. However, the first harmonic provides a less efficient use of the phasor space, leaving $1/2\pi$ radians non utilized and leading to a lower dynamic range of separation as can be seen in Supplementary Figures 9 and 10.

The second harmonic approximately spans over $(3/2 + 2)\pi$ radians on the phasor for spectrum within the visible range (400 nm – 700 nm), distributing spectra in a more expansive fashion within the universal circle, simplifying the distinction of spectra which may be closely overlapping and providing a higher dynamic range of separation as demonstrated in Supplementary Figures 6 and 7. The downside of this harmonic is the presence of an overlap region from orange to deep red fluorescence. Within this region, spectra separated by 140 nm (in our system with 32 bands, 410.5 nm to 694.9 nm with 8.9 nm bandwidth), may end up

overlapping within the phasor plot. In this scenario, it would not be possible to differentiate those well-separated spectra using the second harmonic, requiring the use of the first. Thanks to SEER, the choice of which harmonic to use for visualization can be quickly verified and changed within the HySP software (<http://bioimaging.usc.edu/software.html>).

In the common scenario of imaging with a single laser line, the range of the majority of the signal emitted from multiple common fluorophores is likely to be much smaller than 150 nm due to the Stokes' shift usually in the 20-25 nm range. Excitation spectra of fluorescent proteins separated by 140nm is generally not well overlapping, requiring utilization of a second excitation wavelength to obtain the signal.

The SEER method presented here has utilized the second harmonic in order to maximize the dynamic range of the phasor space and separate closely overlapping spectra. However, SEER can work with the first harmonic seamlessly, maintaining swift visualization of multiple fluorophores that may be far in peak spectral wavelength.”

9) abstract: “reference maps that quantitatively enhance different spectral properties”
As a mentioned before, the ‘quantitative’ aspect could be better exploited. For instance the phase information could be translated into nanometers. In the SEER maps it would be nice to associate what I see in a given color with some spectral information.

We have added new colorbars to each of the original maps in order to provide reference markers for “standard” spectrum corresponding to wavelengths within the 400 nm to 700 nm range. While the exact association of a single wavelength to a color is not exact, this colorbar is indicative of the wavelength represented by the SEER maps. Each color bar was created by using simulated spectra formed from equally spaced gaussian curves whose peaks correspond to wavelengths ranging from 400 nm to 700 nm. Phasor values were calculated from these simulated spectra according to the selected harmonic and converted to their corresponding RGB values, following the same algorithms utilized to generate the SEER reference maps.

Figure 2 was updated:

The caption has been updated :

“[...] Colorbars for b,c,d,e represent the main wavelength associated to one color in nanometers.
 [...] Colorbar represents the normalized relative gradient of counts.”

Figure 4 was updated to include the colorbars:

Figure caption has been updated to include the colorbar:

“[...] Colorbar represents the main wavelength associated to one color in nanometers.
 [...] Colorbar represents the normalized relative gradient of counts.”

Figure 5 was updated to include the colorbars:

Caption reads:

“ [...] Colorbar represents the main wavelength associated to one color in nanometers.”

Figure 6 has been updated to include colorbars:

Caption has been updated:

“ [...] Colorbars represent the main wavelength associated to one color in nanometers.”

Figure 7 has been updated to include colorbars:

Caption has been updated:

“ [...] Colorbars represent the main wavelength associated to one color in nanometers.”

Minor comments

1) Introduction:

“Phasor analysis is generally done by exploring the 2d-histogram of spectral fingerprints by means of geometrical selectors”

This is just the ‘exploring’ feature of Phasor analysis. I agree with the authors that with phasors can be done more than that.

There are also many examples of direct quantitative calculations starting from g and s values. For instance, the works of Scipioni et al (Biophys J 2006, Communications Biol 2008) are not using geometrical selectors but calculate maps directly from the g and s values

We thank the reviewer for directing our attention to these references. We have included the relevant references and supporting text within the main manuscript in order to provide a better explanation of the previous work done in determining direct quantitative calculations from the phasor values:

The text now reads:

“Prior works directly utilize phase and modulation for quantifying, categorizing, and representing features within Fluorescence Lifetime and Image Correlation Spectroscopy data^{31–33}. Our method differs from previous implementations^{22,23,25,27,28,30}, as it focuses instead on providing a quantitatively constructed, holistic pre-processing visualization of large hyperspectral data.”

2) Standard Reference maps:

It should be noted that what the authors call 'Angular' and 'Radial' maps are very common maps in the field of Frequency domain FLIM, as they represent the 'Phase' and 'Modulation' images. They are normally visible in the softwares for Frequency domain FLIM (for instance ISS Vista).

Besides FLIM, examples of phase/modulation images can be found at:

Scipioni et al, Communications Biol 2018 (show phase images)

Sarmento et al, Nat Comm 2018 (show modulation and phase images)

The authors should highlight more what is absolutely new in this paper, i.e. the more complex maps (Gradient, Morphing etc) generated in the phasor plot

We thank the reviewer for pointing this out. We have included a few sentences clarifying that the Angular and Radial maps have previously been introduced.

The main text now reads:

"It is important to note that the idea of Angular and Radial maps have been previously utilized in a variety of applications and approaches^{32,33} and are usually introduced as "Phase" and "Modulation", respectively. Here, we have recreated and provided these maps for our hyperspectral fluorescence data as simpler alternatives to our more adaptable maps."

3) Fig.7b : MIP or section?

Figures 7b and 7d are Imaris, alpha blended, maximum intensity projections.

We have updated the caption of Figure 7 to provide the relevant information:

"Maximum Intensity Projection renderings of *Tg(ubi:Zebrafow)* muscle acquired live in multi-spectral confocal mode with 458nm excitation."

4) Methods: The value of the deviation parameter sigma is not reported. How is it set?

The deviation parameter sigma was experimentally chosen after running several iterations of the gaussian kernel method to generate our initial TrueColor maps.

Reviewer #2 (Remarks to the Author):

The authors report an innovative visualization method that builds on the spectral phasor approach. They designed a set of reference maps/modes to visualize simultaneously multiple spectral components characterizing the analyzed samples. These maps allow

to highlight different spectral properties of the original acquired hyperspectral images. Therefore, by exploiting their reference maps, they can obtain a colored image in which each color is related to a different characteristic of the acquired spectral images.

In the first part of the manuscript the authors provide a description of the spectral phasor approach and the proposed method. Reference color maps and modes are described and tested under different conditions explaining the spectral characteristic that can be highlighted in the acquired spectral images.

The authors applied their method to multiple biological problems. At first they visualized the autofluorescence signal primed by two photon excitation in a mouse tracheal explant, showing the different results that can be obtained by exploiting the Gradient Descent morphed map or the tensor map. They concluded that the first map is able to visualize the metabolic activity, while the second can increase the contrast of cell boundaries.

In a second application, the authors exploit the angular map to discriminate different autofluorescence signals from the mKO2 protein fluorescence in zebrafish samples. In a third example, the authors also show how their reference maps can be exploited to visualize different spectral properties in a triple labeled zebrafish embryo. They report that their method is useful in discerning the difference between multiple fluorophores and autofluorescence both in single plane and in volumetric visualizations. Finally, the authors exploit their method to highlight spectral differences due to the recombination of CFP, YFP and RFP labels in a Zebrafish sample.

The manuscript shows an interesting method to visualize separate fluorescence spectra characterized by different properties. The authors report essentially a discussion of their method together with its validation by means of simulations and in standard biological samples. The potentially more interesting biological application in Zebrafish samples is not explored in details.

The phasor approach has been exploited for multiple quantitative applications in spectral, lifetime, magnetic resonance imaging and second harmonic generation signal analysis. Although a change of color maps can be useful to visualize some details in the samples, I think that the major drawback of the manuscript is the lacking of quantitative evidence, which could improve the impact of the work.

I suggest to the authors to take into account and discuss some major points:

- Is their method robust with respect to typical problems of in-vivo microscopy, such as samples movement in the xy or xyz planes? It would be interesting a discussion of this problem. How can their method take into account sample movements? Does it produce an effect on the phasor plot that can be corrected?

We thank the reviewer for this question. The sample movement in the xy or xyz will affect the acquisition more than the analysis. In our setup (LSM780 Zeiss Quasar) spectra are collected at the same time, not sequentially, hence for each pixel dwell time (typically 1.5-12.6 μ s) the signal of 32 wavelengths bins is collected at the same time. This parallel acquisition approach is more robust, on a pixel-to-pixel basis, than a sequential scanning approach, where sample movements can considerably affect the spectra acquired. For the algorithm to be affected by xy or xyz planes movement, the spectra have to be distorted.

We agree with the reviewer; it would be interesting to see if sample movements produce an effect on the phasor plot that can be corrected. While this is not necessarily within the focus of the paper we went ahead and tested this interesting idea.

We performed multispectral fluorescence volumetric imaging on a zebrafish embryo at 21 hours post fertilization utilizing our commercial setup (LSM780 Zeiss Quasar).

Zebrafish sample was *Gt(cltca-citrine)ct116a; Tg(lifeact:mRuby); Tg(ubiq:membTdTomato)* with triple fluorescence. We acquired:

1. a z-stack with 178 slices (shape 512x512x178x32, [x,y,z,lambda]) as reference
2. The same z-stack was acquired a second time. During this acquisition we performed two “artificial” movements with the stage controller simulating a fish twitch, moving a distance of ~20-30% of the frame size.

In short, we do not see appreciable changes on the phasor due to movement.

The figure below summarizes our test. **(a)** shows a 3D SEER alpha color rendering of the z-stack without and **(b)** with movement, after visualization with SEER as RGB color stack (gradient descent, morph mode center of mass). The arrows show the point where the “twitches” happened. The unnatural sectioning of the sample is evident, making the notocord (generally round) appear as two adjoined semi-circles (left arrow) and shifting the entire sample (right arrow). **(c)** Single plane SEER visualization of the plane containing the position where the scanner was imaging while shifting the stage for the left arrow in panel b (white box). The top portion in the white box shows evident artifacts of information along multiple pixels, which differs from **(d)** the original shape in the same z plane in the volume acquired without perturbation. **(e)** Frame corresponding to the large shift in right arrow panel b. The unnatural shift is visible in the center portion of the frame, where the typical shape of muscle is distorted in an elongated fashion. **(f)** the same frame in the unperturbed sample. **(g)** The phasor of the entire static volume

(panel a) is reported for reference, given the plot integrates all pixels in the volume it is somewhat expected to not see big distortions here. **(h)** Phasor for the whole volume with two artificial twitches (panel b). **(i)** Phasor of the section with movement in panel c, and corresponding **(j)** phasor of the same z-section without movement (from panel d). The main visible difference is highlighted in the red circle in both panels i and j, **(k)** phasor of the “twitch” in panel e and corresponding **(l)** reference for unperturbed slice in panel f. **(m)** From the selection in red on the phasor plot in panel i, the corresponding pixels are highlighted in red on top of an average grayscale image for the moved image and **(n)** the reference (from red phasor selection in panel j). The pixels correspond to cells migrating between one acquisition and the other and are not related to the artificial movement perturbation we introduced in the system.

- Which is the effect of the photobleaching in their visualization? If they acquire images as a function of time in presence of this effect, is it possible that the reconstructed image highlight this effect and not other spectral differences (for example related to combinatorial expression

of different proteins) among the fluorophores? Is there a color map which can take into account this effect among those they designed? A discussion of this problem could be helpful for in-vivo time-lapse imaging.

We thank the reviewer for this question. It is a really good point which can showcase the efficacy of the “modes” implemented in SEER. The “modes” we presented are the answer to the reviewer’s question. In a bleaching scenario, intensities will shift toward the background. The center of mass mode, for example, will dynamically adapt to the dataset during the change of intensities.

To better illustrate this process we performed a bleaching experiment on a zebrafish embryo (details in caption below) and visualized the timelapse using SEER. We have assembled the results in a figure and a Supplementary Video 3 which we believe will answer this question.

We edited the main text to include:

“The adaptivity of the SEER modes can prove advantageous for visually correcting the effect of photobleaching in samples, by changing the apex of the map dynamically with the change of intensities. (Supplementary Figure 24). “

We added the figure as Supplementary Figure 24:

“**Supplementary Figure 24. Visualization of photobleaching with SEER.** Photo-bleaching experiments were performed on a 24 hpf zebrafish embryo *Gt(cltca-citrine); Tg(fli1:mKO2); Tg(ubiq:membTdTomato)*, labeling clathrin, pan-endothelial and membrane respectively. The experiments were performed utilizing the “bleaching” modality in the Zeiss Zen 780 inverted

confocal, where single z positions were acquired in lambda mode. Frames are acquired every 13.7 sec, with 5 intermediate bleaching frames (not acquired) at high laser power until image intensity reached 90% bleaching. The SEER RGB mask represents the values of colors associated to each pixel, independent from the intensity values. The map used here is Radial map in Center of Mass mode. In this modality the map will adjust its position on the shifting center of mass of the phasor clusters, visually compensating for the decrease in intensity. **(a)** In the initial frame the cltca-citrine is associated to a magenta color, membrane to cerulean, pan-endothelial is not in frame, and background to yellow. **(b)** Frame 10 shows consistent colors with the initial bleaching; the colors are maintained **(c)** at frame 40 and **(d)** frame 70 where most of the signal has bleached and most colors have switched to yellow (here, background). **(e)** Final frame shows the 90% bleached sample. The Alpha Color rendering adds the information of intensity to the image visualization. Here we show for comparison **(f)** frame 1, **(g)** frame 10, **(h)** frame 40, and **(i)** frame 70. Scale bar 10 μ m. **(j)** Average total intensity plot as a function of frame, calculated from the sum of 32 channels, shows evident bleaching in the sample. Further visualization is provided in Supplementary Video 3”

In Supplementary Table 1 the authors report a power higher than that used in their previous work (Supplementary Table 3, "Hyperspectral phasor analysis enables multiplexed 5D in vivo imaging" Nature Methods 2017). Is the photobleaching more relevant in the applications shown in this manuscript?

Photobleaching is not more relevant in this application. We have added an experiment where the photobleaching is characterized in a live sample [Supplementary Figure 24, answer to the question above].

Regarding the power levels comparison with our previous work (Cutrale, Trivedi et al, Nature Methods 2017), the aim of this manuscript is to present a preprocessing visualization tool, whereas the previous work was focused on the volumetric time-lapse imaging of multiple labels, making bleaching and denoising important factors. The only zebrafish line in common between the two works is *Tg(kdrl:eGFP); Gt(desmin-Citrine); Tg(ubiq:H2B-Cerulean)* (Fig. 6 in this manuscript) and the power levels we utilized for this line in this work (Supplementary Table 1, 1% @458, this manuscript) are in line with our previous work (3.0-5.0% @458, Supplementary Table 3, Cutrale, Trivedi, Nat. Meth 2017). The autofluorescence images were acquired on mouse samples, which were not described in our previous work, and utilized 1.8% power at 740nm 2photon which is low in our experience (in other experiments we utilize 1.8-2.5% 740nm for cell culture autofluorescence, 1.5-4.5% 740nm for zebrafish autofluorescence using Zeiss LSM780 inverted/Coherent).

The laser power values for Zebrafish samples are high (20%@458nm figure 1, 47.7%@458nm figure 7) for 3 reasons:

- I. the laser utilized on the instrument during those experiments had lower power than the current one and was later changed;
- II. the 458nm here is used to excite 3 fluorescent proteins at once: CFP, GFP, RFP. The three proteins have different excitation efficiencies at 458nm, 93%, 62% and 13% respectively, as well as different expression levels: RFP high (expressed during development), GFP and CFP low (expressed after 4OHT stimulation).
- III. The pixel dwell time was set to the fastest time (1.58 μ s)

- The number of acquired photons decreases rapidly while going deeper into the sample and therefore the fluorescence spectra (especially in the blue region of the visible spectrum) are characterized by lower signal to noise ratio (S/N). Is it possible that the color maps highlight this effect instead of other spectral difference related to the considered biological problems? Is it possible that their method assigns the same color to pixels with low S/N in a superficial plane (due for example to a low protein expression) and to pixels in a deeper plane? In this case the spectra would be depicted with the same color but the low S/N of each spectrum is related to two different effects. Could this eventual problem give rise to wrong tissue/cell segmentation and classification of the associated properties? Is it possible to correct this possible effect due to in-depth imaging?

Maybe they could show the phasor plots and maps for different z-planes and discuss this eventual effect for different color maps/modes.

Is it possible that their color maps introduce some errors in cells segmentation and tracking along the z direction?

It is clear that we have not provided a clear enough explanation of our method. In answering this question, we would first like to reiterate that the SEER maps are focused on enhanced representation, not analysis, and hence used as a pre-visualization step to aid in further post acquisition processing steps. The SEER maps allow users to determine phasor plot regions of interest. To accomplish this purpose the maps associate multiple regions within the biological sample to similar spectra, which are organized by the bins of the phasor histogram.

The SEER maps are constructed directly from the frequency domain values which are generated using the phasor method applied to hyperspectral data, and RGB colors are used to directly represent these values. As such, the quality of color separation has a maximum resolution limited by the spectral separation provided by the phasor method

itself. Therefore, as long as the phasor method can differentiate between spectra which have a higher amount of fluorescent signal vs noise (high signal to noise) and spectra with a higher combination of the noise versus the signal (low signal to noise), the SEER maps will assign different colors to these spectra.

In the scenario where spectra deriving from two different effects are exactly the same, for example a case where low protein expression on an outer layer and attenuation of a high level expression at a deeper level, the phasor method and the SEER maps, in their current implementation, will not be able to differentiate between the two effects. The separation of these two effects is a different and complex problem which depends on the optical microscopy components, the sample, the labels, the multispectral imaging approach and more factors in the experimental design, and we believe falls outside of the scope of this paper and constitutes a project on its own.

However, we have included a figure denoting the fluorescent signal at different z depths for a single fluorescent protein zebrafish here for the reviewer and added a Supplementary Note 2 to the supplementary files which describes our finding. The data presented in the following figure was acquired from a 24hpf Gt(desmin-Citrine) zebrafish embryo in multispectral mode (32 bands, 410.5 nm to 694.9 nm with 8.9 nm bandwidth) with a 488nm laser at 1% power. The dataset is a tiled z-stack with 60 slices, shape (1536x512x60x32) [x,y,z,lambda], and with $2.0\mu\text{m}$ steps. This figure demonstrates that in our samples, as long as there is sufficient signal, the spectral fingerprint on the phasor plot remains consistent even at deeper positions.

In this figure, (a, c, e) denote manually morphed Gradient Descent SEER map rendered z slices at increasing depths (0 um, 60 um, 120 um from the first imaging slice) with (b, d, f) as the phasor plot calculated from the entire z-stack. For each slice, the average phasor coordinate values are calculated for small blocks of the z-slice with (a) red, (c) orange, (e) brown boxes corresponding to the (b) red, (d) orange, (f) brown dots on the phasor plot. In (g), we show that the average phasor coordinates from (a,b), (c,d), (e,f) are within one standard deviation of each other.

We thought this was an important point that readers need to consider, so we edited the last paragraph of the results, which now reads:

“Some limitations that should be considered are that SEER pseudo-color representation sacrifices the “true color” of the image, creating inconsistencies with the human eyes’ expectation of the original image and does not distinguish identical signals arising from different biophysical events (Supplementary Note 2).”

The supplementary Note 2:

“Supplementary Note 2

Color visualization limitations for SEER

The SEER maps are built based on the frequency domain values generated by applying the phasor method to hyperspectral and multispectral fluorescent data. RGB colors are used to directly represent these values. As such, the quality of color separation has a maximum resolution limited by the spectral separation provided by the phasor method itself. Therefore, as long as the phasor method can differentiate between spectra which have a higher amount of fluorescent signal vs noise (high signal to noise) and spectra with a higher combination of the noise versus the signal (low signal to noise), the SEER maps will assign different colors to these spectra. In the scenario where spectra derived from two different effects are exactly the same, for example, a case where low protein expression is on an outer layer and a high level expression is attenuated at a deeper level, the phasor method and the SEER maps, in their current implementation, will not be able to differentiate between the two effects. The separation of these two effects is a different and complex problem which depends on the optical microscopy components, the sample, the labels, the multispectral imaging approach and more factors in the experimental design, and we believe this separation falls outside of the scope of this paper and constitutes a project on its own.

”

In my opinion the major concern about the manuscript is that the authors provide only a visualization comparison among different color maps/modes. The authors should discuss in more detail which parameters can be extracted from the images obtained by applying the different color maps and show some quantitative results. Quantitative information allowing to reach a deeper understanding of the biological problems with respect to other unmixing/color maps methods (for example Ref. 26, 29, or hue vs saturation plot in Ref.31) could greatly enhance the impact of their method.

We thank the reviewer for the question. We would like to reiterate that this work is focused on the pre-processing enhanced representation, rather than analysis, of the multidimensional hyperspectral dataset that can be useful in guiding the experimentalists during acquisition and analysis of the data. In the context of this work, the main goal of SEER is to informatively visualize the spectral information contained in the data on a standard RGB monitor rather than to be an unmixing method.

However, we thank the reviewer for the suggestion of improving the quantitative aspect of SEER. We agree it will improve the manuscript.

To address these points we have performed the following steps:

1. Added a comparison image with our previous work (Cutrale, Trivedi, Nature Methods 2017).
2. Added an “in image” information providing a quantitative value to the colormaps.
3. Expanded our simulations, adding a quantitative evaluation of the timing and accuracy of SEER compared to a number of rapid color visualization strategies and Independent Component Analysis.

1. Comparison with our previous work

We would like to express that the SEER presented here and HySP (our previous work) are different implementation of the phasor advantages.

SEER maps are focused on enhanced representation, not analysis, and hence used as a pre-visualization step to aid in further post-acquisition processing steps. They require minimal amount of user interaction (the choice of map) and output color images (RGB, shape [x,y,3]). The process takes seconds for large (Gigabytes) datasets, as we report in our Supplementary Figure 1 (listed below). The question SEER answers is the visualization of multidimensional spectral data (x,y,z,t, lambda).

The work we reported previously (Cutrale, Trivedi 2017) is an analysis method for denoising and unmixing spectral signals in longitudinal low signal to noise microscopy datasets. The analysis is assisted, requires user interaction with ROIs and outputs separate unmixed channels-volumes (monochrome, shape [x,y,z,t]).

The process takes minutes for large (Gigabytes) datasets, as we reported in our previous work.

The question HySP answers is the separation of components in any multidimensional spectral data (x,y,z,t, lambda).

We updated the section introducing SEER in the main manuscript, which now reads:

“Spectrally Encoded Enhanced Representations (SEER):

SEER rapidly produces 3 channel color images (Supplementary Figure 1) that approximate the visualization resulting from a more complete spectral unmixing analysis (Supplementary Figure 2).”

We added a supplementary figure with the comparison showing that the visualization obtained with SEER well approximates that obtained from the complete hyperspectral phasor analysis.

The figure:

“Supplementary Figure 2. Comparison of SEER with visualized HySP [1] results.

Here we show a zebrafish embryo *Tg(kdrl:eGFP); Gt(desmin-Citrine); Tg(ubiq:H2B-Cerulean)* labelling respectively vasculature, muscle, and nuclei. Live imaging with a multi-spectral confocal microscope (32-channels) using 458nm excitation. Single plane slices of the tiled volume are rendered with SEER maps (3 channel, RGB) and compared to rendering of the same dataset analyzed with HySP (here 5 channels) [1]. **(a)** Rendering of a 5 channel HySP analyzed dataset, the dashed box is expanded in the zoomed-in portion of panel a with its **(b)** line profile to the right along the solid line for all 5 separate channels, eGFP, Citrine, Cerulean, Pigments and autofluorescence at 458nm. **(c)** Visualization of the 5 channel dataset as a blended RGB, similarly to how it appears on a screen. The **(d)** morphed mode center of mass visualization shows patterns in accordance with HySP with a differently color coded **(e)** line profile along the solid line in panel d, which shows intensities in the 3 R,G,B channels of the image. The profiles of the single R,G,B do not match the unmixed HySP profiles in panel b. However, **(f)** color visualization of the same line plot (as R,G,B vectors), shows patterns in accordance to the on-screen visualization of HySP unmixed data. Similarly, **(g)** morphed mode max visualization shows an image in accordance to the rendered HySP analyzed data in panel a with its **(h)** line profile along the solid line of the zoomed-in portion of panel g being comparable to both the HySP 5 separate channels and the R,G,B profiles of the different morphed center of mass map in panel e. **(i)** The color on-display visualization of the RGB intensities in g reveals different color features as those of the HySP unmixed channels (panel b). “

2. Added an “in image” colorbar providing quantitative values to the colormaps. While the exact association of a single wavelength to a color is not exact, this colorbar is indicative of the wavelength represented by the SEER maps. (See response to reviewer #1 question 9)
3. Quantitative evaluation of the timing and accuracy of SEER compared to a number of rapid color visualization strategies and Independent Component Analysis.

In the light of a more quantitative comparison with other methods, we have added a deeper description of the effects at different SNR and at different spectral overlaps, providing an accuracy value in comparison to other RGB visualization approaches.

A new section has been added in the main text Methods part describing the parameter of accuracy we utilized. The text reads:

“**Accuracy calculation:** We utilize the Simulated Hyperspectral Test Chart to produce different levels of spectral overlap and signal to noise ratio (SNR). We utilize multiple RGB visualization approaches for producing compressed RGB images (Supplementary Fig 15, Supplementary Fig 16). Each panel of the simulation is constructed by three different spectra, organized as three concentric squares Q_1, Q_2, Q_3 (Supplementary Fig 1). Hence, the maximal contrast visualization is expected to have three well separated colors. For quantifying this difference, we consider each (R,G,B) vector, with colors normalized [0,1], in each pixel as a set of Euclidean coordinates (x,y,z) and for each pixel calculate the Euclidean distance:

$$l_{12} = \sqrt{\sum_{i=R}^B (p_{Q1} - p_{Q2})_i^2} \quad (\text{eq. 3})$$

where l_{12} is the color distance between square Q_1 and Q_2 , p_{Q1} and p_{Q2} are the (R,G,B) vectors in the pixels considered, i is the color coordinate R, G or B. The color distances Q_1Q_3, l_{13} , and Q_2Q_3 are calculated similarly. The accuracy (Supplementary Fig 17) is calculated as:

$$\text{acc} = \frac{(l_{12} + l_{13} + l_{23})}{3\sqrt{2}} \quad (\text{eq. 4})$$

where the denominator is the largest color distance $l_{\text{red-green}} + l_{\text{red-blue}} + l_{\text{green-blue}}$ “

We have added a supplementary figure showing the image result of SEER at different harmonics with respect to other methods under different values of spectral overlap and SNR. Supplementary figure and caption now read:

“Supplementary Figure 20. RGB Visualization with multiple modalities under different spectral overlap and SNR conditions. In this simulation, the first panel (top-left) of the Simulated Hyperspectral Test Chart (SHTC, Supplementary Figure 3) is reduced in intensity by a factor of $(.5*10)^1 - (.5*10)^4$ (panel 1-5 respectively) in the presence of a constant background. Background with average intensity 5 was generated in Matlab, poissonian noise was added using the `poissrnd()` function obtaining 5 different levels of SNR. **(a,b,c,d,e)** Peak-to-peak distance for the spectra in the middle and outer concentric squares in the SHTC is shifted by units of 8.9nm with respect to the peak of the average spectrum in the intermediate square, which is kept constant in this simulation (similarly to Supplementary Figure 3) starting from distance 0 (a) to 35.6nm (e). For each level of spectral overlap **(a-e)**, seven different RGB visualization modalities are presented here for comparison at five different level of SNR. In order from the top row, SEER at harmonic 2 (SEER h=2) and harmonic 1 (SEER h=1), peak wavelength selected (Peak Wav.), Gaussian kernel set at 30% of the spectrum (Gauss r=.3), set at 20% (Gauss r=.2) and at 10% (Gauss r=.1), finally Gaussian kernel set at 650nm, 510nm, 470nm for RGB respectively (Gauss. Def.). **(f)** the wavelength-to-RGB conversion map used for the peak wavelength visualization. **(g)** center wavelength for the R=579nm, G=534nm, B=499nm channels of Gauss r=.3. Average spectrum (yellow) **(h)** center wavelength for the R=597nm, G=543nm, B=490nm channels of Gauss r=.2. Average spectrum (yellow). **(i)** center wavelength for the R=614nm, G=543nm, B=481nm channels of Gauss r=.1. Average spectrum (yellow). **(j)** center wavelength for the R=650, G=510nm, B=470nm channels of Gauss. Def. Average spectrum (yellow). The maps utilized here for SEER were gradient descent in scale mode (a, b, c, d), and center of mass mode (e). Visualization with SEER shows a reasonably constant contrast and color for the different spectra in the simulation at different SNR.

”

We then provided a visual representation of the spectra in the above simulation:

“Supplementary Figure 21. Spectra of extreme conditions in SNR-Overlap simulation. The extremes of the simulation utilized in Supplementary Figure 20 are reported here as spectra for comparison. For high signal-to-noise ratio (a) average spectrum for spectra with peak-maxima distance set to zero and (b) example single spectra from each concentric square region of the simulation (digital levels, DL). (c) Average and (d) single spectra at high SNR for simulation with spectra separated with a peak-to-peak distance of 35.6nm. (e) Reference Simulated Hyperspectral Test Chart with color coded concentric squares. The low SNR simulation spectra are reported here for a peak distance of zero as (f) average and (g) single and for a peak distance of 35.6nm as (h) average and (i) single.”

We quantified the accuracy of SEER, as defined in the Methods section, for the multiple overlaps and SNR, comparing with other visualization methods:

“Supplementary Figure 22. Accuracy of SEER under different spectral overlap and SNR conditions. Accuracy was calculated for different signal-to-noise ratios and spectral maxima separation (a) aligned, (b) 8.9nm, (c) 17.8nm, (d) 26.7nm, (e) 35.6nm, starting from the visualizations in Supplementary Figure 20 and corresponding spectra in Supplementary Figure 21. The accuracy is calculated here as the sum of the Euclidean distance of the RGB vectors between pairs of the concentric squares of the simulation, in ratio to the largest color separation (red to green, red to blue, blue to green). A thorough description of accuracy calculation is reported in the Methods section. Each value in the plots represents the average distance of 200^2 pixels; error bars are the standard deviation of normalized accuracy value across all pixels. The average accuracies over multiple SNR conditions for each spectral maxima separation: (a) with highly overlapping spectra SEER provides on average 38.0% for harmonic 1, 50.6% for harmonic 2, while the best performing other comparison here is Gaussian $r=.3$ with an average 26.7%. (b) With a 8.9nm peak-to-peak separation SEER h=1 averages 57.0%, SEER h=2 49.6%, other best performing here is Peak Wavelength with 22.2%; (c) with 17.8nm separation SEER h=1 averages 57.2%, SEER h=2 $60.0 \pm 2.3\%$, other best Gauss $r=.3$ with 26.2% (d) with 26.7nm separation SEER h=1 averages 59.9%, SEER h=2 60.4%, other best Gauss $r=.3$ with 32.1%; (e) with well separated spectra 35.6nm apart, SEER h=1 averages 66.3%, SEER h=2 66.7%, with other best Gauss $r=.3$ scoring 43.5% in average. “

In order to perform a timing comparison with ICA, we utilized Python package Scikit-Learn function, `sklearn.decomposition.FastICA`, with 3 independent components (ICs) (source: <https://scikit-learn.org/stable/modules/generated/sklearn.decomposition.FastICA.html>). We made a comparison in computing time of SEER with ICA and added a new supplementary figure:

Figure caption reads:

“Supplementary Figure 1. Computational time comparison of SEER and ICA for different file sizes. (a) HySP and ICA run times (plot in log scale) were measured on a HP workstation with two 12 core CPUs, 128 GB RAM, and 1TB SSD. SEER run times were measured within a modified version of the software. ICA run times were measured using a custom script and the FastICA submodule of the python module, scikit-learn. Timers using the `perf_counter` function within the python module, `time`, were placed around specific functions corresponding to the calculations required for the creation of SEER maps in HySP and extracting individual component outputs from the custom ICA script. Data size varies from 0.02-10.97GB, with constant number of bands (32 bands, 410.5 nm to 694.9 nm with 8.9 nm bandwidth) corresponding to a range of $2.86 \cdot 10^5$ - $1.83 \cdot 10^8$ spectra. ICA testing was limited to 10.97GB maximum as for higher values the RAM requirements exceeded the 128GB available on our workstation. **(b)** For the custom ICA script, timers were placed to measure the time to reshape the hyperspectral data for ICA input, to run the ICA algorithm, and to convert values of the ICA

components into image intensity values, reaching minutes of computation at just 1.1GB (plot in log scale) . (c) For HySP, timers were placed to measure the generation of the phasor values from hyperspectral data, including initial calculations of the real and imaginary components (g and s) and creation of the phasor plot histogram. A timer was also placed around all preparatory functions required for on-the-fly creation of SEER maps. The more memory-efficient phasor process allowed us to compute datasets of size 0.02-43.9GB, corresponding to a range of $2.86 \cdot 10^5$ - $7.34 \cdot 10^8$ spectra (plot in log scale) .”

The main text was updated, now reads:

“Our tests show SEER can process a 3.7 GB dataset with $1.26 \cdot 10^8$ spectra in 6.6 seconds and a 43.88 GB dataset with $1.47 \cdot 10^9$ spectra in 87.3 seconds. Comparing with the python module, scikit-learn’s implementation of fast Independent Component Analysis, SEER provides up to a 67 fold speed increase (Supplementary Figure 1) and lower virtual memory usage.”

Moreover, a comparison of the images and the information that can be retrieved by means of the color maps/modes shown in this manuscript and the unmixing method of their previous work (Ref. 29) should be interesting.

We thank the reviewer for the question. We believe point 1 in the previous answer covers this point. We updated the section introducing SEER in the main manuscript, which now reads:

“Spectrally Encoded Enhanced Representations (SEER):

SEER rapidly produces 3 channel color images (Supplementary Figure 1) that approximate the visualization resulting from a more complete spectral unmixing analysis (Supplementary Figure 2).”

We added a supplementary figure with the comparison showing that the visualization obtained with SEER well approximates that obtained from the complete hyperspectral phasor analysis.

Minor point:

-In Scale and Morph modes, the apex of the map shifts and changes the reference center of the color palette depending on the dataset. Are therefore these modes useful only when applied to static images?

We thank the reviewer for raising this point. Since one of the main focuses in our lab is live imaging, we designed the adaptivity of scale and morph mode with dynamics in mind. The example application (suggested by this reviewer above) is an excellent example where the visualization of bleaching can be partially compensated using a morphed map.

These modes will also work with static images. The reviewer raises a good point which we addressed in the software: we added a function to maintain the same “scaled” and “morphed” mode across multiple samples, to aid in the visual comparison.

We edited the main text in the Modes (Scale and Morph) paragraph, which now reads:

“The boundaries of the Scaled mode can be set to a constant value across different samples to facilitate comparison. “

Reviewer #3 (Remarks to the Author):

This article describes an enhancement to the visualisation of a hyperspectral image processing data produced by a technique called SEER, which spectrally decomposes fluorescence hyperspectral images into spectral components to facilitate visualisation of multiple fluorophores in a single HSI image. The main contribution of this paper seems to be in the mapping of SEER data onto a standard color representation with enhanced contrast between fluorophores. The primary question is whether this is sufficient in itself, if done well, to justify publication in NatComm. The ancillary question is whether the work and the assessment of the results is sufficient. I believe that the manuscript as it stands is marginal on both counts, but could be sufficiently improved, particularly in terms of clarity and rigour to justify publication.

SEER may be considered as an alternative to established spectral decomposition techniques, such as PCA, (or better, ICA), but has the advantage of simplicity and computational speed – this is the major claimed advantage. While fast processing should certainly be true, speed is not quantified absolutely or relatively – and this is an important omission for would-be users.

An increase in processing speed compared to a traditional technique (such as ICA), which may already be implemented quite easily and computed quickly (<1s) using standard software such as ENVI, might not be that important. Conversely a modest increase in speed of a slow process (minutes) could be important. Absolute processing time and improvement in processing time are both important and neither are quantified. Nevertheless the main narrative of this paper seems to be visualisation of SEER data rather than SEER itself.

We thank the reviewer for the constructive criticism. We have worked on improving the clarity of the manuscript to address the concerns of this reviewer, particularly:

1. Clarifications on the method
2. Processing speed measurements/ENVI testing
3. Comparison with ICA

1. Clarifications on the method

It is clear that we have not provided a clear enough explanation of our method. In answering this point, we would first like to reiterate that the SEER maps are focused on enhanced representation, not analysis, and hence used as a pre-visualization step to aid in further post acquisition processing steps. SEER is an acronym for Spectrally Encoded Enhanced Representation, an acronym we chose to highlight the “visualization” aspect of this work and avoid confusion with Hyperspectral Data Analysis or with Unmixing.

In the fluorescence microscopy field spectral datasets are large, easily in the tens of GigaBytes, with spectra numbering in the billions (1.85×10^9 for our Figure 5 for example) with a lower number of spectral bands compared to remote sensing. Given the intrinsic biological variability and the usually low SNR, it is common to acquire a number of samples (in our experiments ~10) for each condition and each control, positive and negative. Analysis of multiple large multispectral datasets is a major bottleneck, as a common preprocessing step is to visualize data before performing a full-fledged analysis in order to optimize computational time.

We have made the following changes in the text to stress the pre-processing component of this work:

The title has been updated; it now reads:

“Pre-processing visualization of hyperspectral fluorescent data with Spectrally Encoded Enhanced Representations (SEER)”

We have added a column in the Supplementary Table 1 with the file size [GigaBytes].

The column now reads:

Size
[GB]
1.93
0.06
3.44
0.45
0.13
6.48

We added a section with data availability which contains the data utilized in the figures:

“Data Availability

All the relevant data are available from the corresponding author upon reasonable request. Datasets for figures 1, 2-3, 4, 5, 6, 7 are available for download at <http://bioimaging.usc.edu/software.html#sampledatasets> in the samples section. “

2. Processing speed measurements/ENVI testing

We thank the reviewer for raising this very good point on timing and processing speed; this has helped us improve the manuscript and demonstrate the power of the technique.

We first tested ENVI. We are aware of the existence of the software, however such software is not designed for the microscopy fluorescence field and does not appear to perform a direct “color” visualization of hyperspectral data in a timely manner. While we do not own a license, we managed to obtain a demo from the manufacturer.

The reviewer is correct, ICA on ENVI can be computed in <1s, however, from our testing, the fast computing time is the case only if a small dataset of shape

100x100x32 is utilized when performing the calculation on a typical high performance laptop (6-core 2.2GHz Intel Core i7, 32GB RAM 2400MHz DDR4, Flash drive). This dataset size is generally much smaller (~2000-10000 times) than the standard microscopy images as we report in Supplementary Table 1.

ENVI software seems to be optimized for a different type of data and analysis and does not perform well, timewise, with our microscopy data. Considering that the purpose of SEER is the rapid visualization of large data from microscopy images, usually 4D (x,y,z,lambda) or 5D (x,y,z,lambda,time), we could not load volumes (x,y,z,lambda) in ENVI in a simple way, and so it was necessary to perform the analysis on a single microscopy slice with size 768x512x32 (x,y,lambda) 16bit, by utilizing ENVI's SPEAR Independent Component analysis wizard with 3 independent components.

The timing (from video frames) was 7 seconds for computing statistics, 13 seconds for calculating independent components, 2 seconds for 2D coherence and sorting bands, totaling 22 seconds for one z plane.

Here is a video screen recording with the analysis and timing

<https://youtu.be/rAX1Veob3PA>

In addition, it is not clear how SPEAR Independent Component Analysis works, since the code is closed-source.

In order to perform a timing comparison with ICA, we utilized Python package Scikit-Learn function, `sklearn.decomposition.FastICA`, with 3 independent components (ICs) (source [https://scikit-](https://scikit-learn.org/stable/modules/generated/sklearn.decomposition.FastICA.html)

[learn.org/stable/modules/generated/sklearn.decomposition.FastICA.html](https://scikit-learn.org/stable/modules/generated/sklearn.decomposition.FastICA.html)).

We made a comparison in computing time of SEER with ICA and added a new supplementary figure:

Figure caption reads:

“**Supplementary Figure 1. Computational time comparison of SEER and ICA for different file sizes.** (a) HySP and ICA run times (plot in log scale) were measured on a HP workstation with two 12 core CPUs, 128 GB RAM, and 1TB SSD. SEER run times were measured within a modified version of the software. ICA run times were measured using a custom script and the FastICA submodule of the python module, scikit-learn. Timers using the `perf_counter` function within the python module, `time`, were placed around specific functions corresponding to the calculations required for the creation of SEER maps in HySP and extracting individual component outputs from the custom ICA script. Data size varies from 0.02-10.97GB, with constant number of bands (32 bands, 410.5 nm to 694.9 nm with 8.9 nm bandwidth) corresponding to a range of $2.86 \cdot 10^5$ - $1.83 \cdot 10^8$ spectra. ICA testing was limited to 10.97GB maximum as for higher values the RAM requirements exceeded the 128GB available on our workstation. (b) For the custom ICA script, timers were placed to measure the time to reshape the hyperspectral data for ICA input, to run the ICA algorithm, and to convert values of the ICA components into image intensity values, reaching minutes of computation at just 1.1GB (plot in log scale). (c) For HySP, timers were placed to measure the generation of the phasor values from hyperspectral data, including initial calculations of the real and imaginary components (`g` and `s`) and creation of the phasor plot histogram. A timer was also placed around all preparatory functions required for on-the-fly creation of SEER maps. The more memory-efficient phasor process allowed us to compute datasets of size

0.02-43.9GB, corresponding to a range of $2.86 \cdot 10^5$ - $7.34 \cdot 10^8$ spectra (plot in log scale)
.”

The main text was updated and now reads:

“Our tests show SEER can process a 3.7 GB dataset with $1.26 \cdot 10^8$ spectra in 6.6 seconds and 43.88 GB with $1.47 \cdot 10^9$ spectra in 87.3 seconds. Comparing with the python module, scikit-learn’s implementation of fast Independent Component Analysis, SEER provides up to a 67 fold speed increase (Supplementary Figure 1) and lower virtual memory usage.”

3. Comparison with ICA

We have performed a comparison with ICA, utilizing a more thorough simulation. We have added a more detailed description of the effects at different SNR and at different spectral overlaps, providing an accuracy value in comparison to other RGB visualization approaches.

A new section has been added in the main text Methods part describing the parameter of accuracy we utilized. The text reads:

“Accuracy calculation: We utilize the Simulated Hyperspectral Test Chart to produce different levels of spectral overlap and signal to noise ratio (SNR). We utilize multiple RGB visualization approaches for producing compressed RGB images (Supplementary Fig 15, Supplementary Fig 16). Each panel of the simulation is constructed by three different spectra, organized as three concentric squares Q_1 , Q_2 , Q_3 (Supplementary Fig 1). Hence, the maximal contrast visualization is expected to have three well separated colors. For quantifying this difference, we consider each (R,G,B) vector, with colors normalized $[0,1]$, in each pixel as a set of Euclidean coordinates (x,y,z) and for each pixel calculate the Euclidean distance:

$$l_{12} = \sqrt{\sum_{i=R}^B (p_{Q1} - p_{Q2})_i^2} \quad (\text{eq. 3})$$

where l_{12} is the color distance between square Q_1 and Q_2 , p_{Q1} and p_{Q2} are the (R,G,B) vectors in the pixels considered, i is the color coordinate R, G or B. The color distances Q_1Q_3 , l_{13} , and Q_2Q_3 are calculated similarly. The accuracy (Supplementary Fig 17) is calculated as:

$$acc = \frac{(l_{12} + l_{13} + l_{23})}{3\sqrt{2}} \quad (\text{eq. 4})$$

where the denominator is the largest color distance $l_{\text{red-green}} + l_{\text{red-blue}} + l_{\text{green-blue}}$

We added a new Supplementary Figure:

Figure caption reads:

“Supplementary Figure 23. Comparison of SEER and ICA spectral image visualization (RGB) under different spectral overlap and SNR conditions. The same simulation used in Supplementary Figure 20, which changes parameters for the Simulated Hyperspectral Test Chart obtaining different values of peak-to-peak spectral overlap and signal to noise, is used here to compute the accuracy for Independent Component Analysis using Python package Scikit-Learn function, `sklearn.decomposition.FastICA`, with 3 independent components (ICs) without optimization for this specific dataset. (a,b,c,d,e). The three ICs are utilized as R, G, B channels for creating a color image for each simulation parameter (ICA = 3 line) and are shown here next to SEER harmonic 1 and 2 (SEER h=1 and SEER h=2 respectively). Error bars are the standard deviation. (f,g,h,i,j) The parameters of accuracy described in the methods section are applied here to the SEER and ICA results. Each value in the plots represents the average distance of 200^2 pixels and error bars are the standard deviation of normalized accuracy value across all pixels. Accuracy of ICA, as calculated here, for multiple overlap values at high SNR (over 30) is in average 48.0% comparable to SEER, 57.9% h=1 and 57.6% h=2, but is reduced at low SNR (below 10) where it averages 21.0%, with SEER at 50.6% and 57.7% respectively for the first and second harmonic. (f) average accuracy ICA 20.2%, SEER 38.0% for harmonic 1, 50.6% for harmonic 2; (g) average accuracy ICA 36.1%, SEER h=1 57.0%, SEER h=2 49.6%; (h) average accuracy ICA 25.3%, SEER h=1

57.2%, SEER h=2 60.0%; (i) average accuracy ICA 35.9%, SEER h=1 59.9%, SEER h=2 60.4%; (j) average accuracy ICA 32.6%, SEER h=1 66.3%, SEER h=2 66.7% “

The main text was updated to reference these supplementary figures:

“A simulation comparison with other common visualization approaches such as Gaussian kernel and peak wavelength selection (Supplementary Figure 20, Methods) shows an increased accuracy (Methods) for SEER to associate distinct colors to closely overlapping spectra under different noise conditions (Supplementary Figure 21). The accuracy improvement was 1.4-2.6 fold for highly overlapping spectra, 0nm-8.9nm spectra maxima distance, and 1.5-2.3 fold for overlapping spectra with maxima separated by 17.8nm-35.6nm (Supplementary Figure 22, 23, Methods).”

The fundamental principles of SEER are described in ref 26 by Fereidouni, Bader and Gerritson of Utrecht (Optics Express, 2012) and its application to in vivo imaging of zebrafish was described in ref 27 (Nature Methods by the authors of this manuscript) in 2017. The first four pages of the manuscript describes the principles of SEER covering ground previously described in references 26 and 27 without clearly highlighting the new contribution compared to ref 27. There is additional new material as described below but the novelty of this new material is not clear.

We thank the reviewer for this question. We have made edits in the main text to separate our previous work and Dr. Fereidouni's from the current manuscript. We have also improved the description of the novelty of this work compared to our previous work.

The abstract now reads:

“Here we present Spectrally Encoded Enhanced Representations (SEER), a novel approach **for simultaneous color visualization of multiple** spectral components of hyperspectral fluorescence images. [...] We present multiple relevant biological fluorescent samples and highlight how our family of SEER maps enhance specific and subtle spectral differences, providing a fast, intuitive and quantitative way to interpret hyperspectral images during collection, **pre-processing** and analysis.”

The main text introduction (page 1) now reads:

“To optimize image acquisition, it is advantageous to perform an informed visualization of the spectral data as it is acquired, especially during lengthy time-lapse recordings and prior to performing analysis. Such preprocessing visualization allows scientists to evaluate image collection parameters while in the experimental pipeline as well as to properly choose the most appropriate processing method. However, the challenge is to rapidly **visualize subtle spectral**

differences with a set of three colors, compatible with displays and human eyes, while minimizing loss of information.”

In the paragraph about “Dimensional reduction strategies” at the end of page 1, we added ICA and reiterated the challenge of visualizing large spectral data prior to a full-fledged lengthy analysis.

“One strategy is to construct fixed spectral envelopes from the first three components produced by principal component analysis (PCA) or independent component analysis (ICA), thus converting a hyperspectral image display to a three-band visualization¹⁰⁻¹⁴.

[...]

The challenge is to develop tools that allow for the efficient visualization of multi-dimensional datasets without the need for computationally demanding dimensionality reduction, such as ICA, prior to analysis. “

We then edited the last paragraph on page 2 where we introduce the phasors and separate this work from the previous work. First we reiterate the novelty of our previous work, which is not only the “application to in vivo imaging of zebrafish” from the work of Gerritson of Utrecht [26] and that of Gratton of UC Irvine [23, 27, 28, 31, 35, 70], but also an implementation of rapid denoising for multispectral volumetric fluorescence microscopy data that enables reduced photo-bleaching and -toxicity. The paragraph reads:

“Exploiting the advantages of the phasor approach, Hyper-Spectral Phasors (HySP) has enabled analysis of 5D hyperspectral time-lapse data semi-automatically as similarly colored regions cluster on the phasor plot. These clusters have been characterized and exploited for simplifying interpretation and spatially lossless denoising of data, improving both collection and analysis in low-signal conditions²⁹. Phasor analysis generally explores the 2d-histogram of spectral fingerprints by means of geometrical selectors^{22,23,25,27,28,30}, which is an effective strategy but requires user involvement.“

In the same paragraph (still part of the introduction, page 2) we introduce the limitations of the prior works :

“While capable of imaging multiple labels and separating different spectral contributions as clusters, this approach is inherently limited in the number of labels that can be analyzed and displayed simultaneously. Prior works directly utilize phase and modulation for quantifying, categorizing, and representing features within Fluorescence Lifetime and Image Correlation Spectroscopy data³¹⁻³³ Our method differs from previous implementations^{22,23,25,27,28,30}, as it focuses instead on providing a quantitatively constructed, holistic pre-processing visualization of large hyperspectral data. ”

SEER (Spectrally Encoded Enhanced Representations) is introduced in the following paragraph (page 4) and in the section titled “Spectrally Encoded Enhanced Representations (SEER)”

The manuscript goes on to describe the main contribution: maps of the SEER data onto RGB colour spaces including mechanisms for spectral contrast enhancement. The techniques seem to be largely heuristic and assessed qualitatively using striking fHSI images. The manuscript demonstrates the application of these techniques for various samples such as 2p fluorescence of mouse trachea and multiply labelled zebrafish. There are several claims of enhanced performance, and also claims of confirmation by FLIM images, but given the lack of rigour in the comparison and uncontrolled nature of the biological samples this seems to be a rather weak validation and the confirmation by FLIM is not clear to this reviewer.

We thank the reviewer for this point . We would like to reiterate that SEER is an acronym for Spectrally Encoded Enhanced Representation and is meant as a rapid algorithm for visualization of Hyperspectral data through a dimensionality reduction from n channels to 3 (R,G,B).

We have worked on improving the clarity of the manuscript to address the concerns of this reviewer, particularly:

1. Rigour in the comparison
 2. Unclear confirmation by FLIM for autofluorescence samples, weak validation, uncontrolled nature of samples
-
1. Improvements on rigour of comparison.

To address the concerns on comparison of SEER with other methods, combined with the uncontrolled nature of biological samples, we have created a more thorough simulation and performed a more quantitative comparison with other methods. We have added a more detailed description of the effects at different SNR and at different spectral overlap, providing an accuracy value in comparison to other RGB visualization approaches.

A new section has been added in the main text Methods part describing the parameter of accuracy we utilized. The text reads:

“**Accuracy calculation:** We utilize the Simulated Hyperspectral Test Chart to produce different levels of spectral overlap and signal to noise ratio (SNR). We utilize multiple RGB visualization approaches for producing compressed RGB images (Supplementary Fig 15, Supplementary Fig 16). Each panel of the simulation is constructed by three different spectra, organized as three concentric squares Q_1, Q_2, Q_3 (Supplementary Fig 1). Hence, the maximal contrast visualization is expected to have three well separated colors. For quantifying this difference, we consider each (R,G,B) vector, with colors normalized [0,1], in each pixel as a set of Euclidean coordinates (x,y,z) and for each pixel calculate the Euclidean distance:

$$l_{12} = \sqrt{\sum_{i=R}^B (p_{Q1} - p_{Q2})_i^2} \quad (\text{eq. 3})$$

where l_{12} is the color distance between square Q_1 and Q_2 , p_{Q1} and p_{Q2} are the (R,G,B) vectors in the pixels considered, i is the color coordinate R, G or B. The color distances Q_1Q_3, l_{13} , and Q_2Q_3 are calculated similarly. The accuracy (Supplementary Fig 17) is calculated as:

$$acc = \frac{(l_{12} + l_{13} + l_{23})}{3\sqrt{2}} \quad (\text{eq. 4})$$

where the denominator is the largest color distance $l_{\text{red-green}} + l_{\text{red-blue}} + l_{\text{green-blue}}$ “

We have added a supplementary figure showing the image result of SEER at different harmonics with respect to other methods under different values of spectral overlap and SNR. Supplementary figure and caption now read:

“Supplementary Figure 20. RGB Visualization with multiple modalities under different spectral overlap and SNR conditions. In this simulation, the first panel (top-left) of the Simulated Hyperspectral Test Chart (SHTC, Supplementary Figure 3) is reduced in intensity by a factor of $(.5*10)^1 - (.5*10)^4$ (panel 1-5 respectively) in the presence of a constant background. Background with average intensity 5 was generated in Matlab, poissonian noise was added using the `poissrnd()` function obtaining 5 different levels of SNR. **(a,b,c,d,e)** Peak-to-peak distance for the spectra in the middle and outer concentric squares in the SHTC is shifted by units of 8.9nm with respect to the peak of the average spectrum in the intermediate square, which is kept constant in this simulation (similarly to Supplementary Figure 3) starting from distance 0 (a) to 35.6nm (e). For each level of spectral overlap **(a-e)**, seven different RGB visualization modalities are presented here for comparison at five different level of SNR. In order from the top row, SEER at harmonic 2 (SEER $h=2$) and harmonic 1 (SEER $h=1$), peak wavelength selected (Peak Wav.), Gaussian kernel set at 30% of the spectrum (Gauss $r=.3$), set at 20% (Gauss $r=.2$) and at 10% (Gauss $r=.1$), finally Gaussian kernel set at 650nm, 510nm, 470nm for RGB respectively (Gauss. Def.). **(f)** the wavelength-to-RGB conversion map used for the peak wavelength visualization. **(g)** center wavelength for the R=579nm, G=534nm, B=499nm channels of Gauss $r=.3$. Average spectrum (yellow) **(h)** center wavelength for the R=597nm, G=543nm, B=490nm channels of Gauss $r=.2$. Average spectrum (yellow). **(i)** center wavelength for the R=614nm, G=543nm, B=481nm channels of Gauss $r=.1$. Average spectrum (yellow). **(j)** center wavelength for the R=650, G=510nm, B=470nm channels of Gauss. Def. Average spectrum (yellow). The maps utilized here for SEER were gradient descent in scale mode (a, b, c, d), and center of mass mode (e). Visualization with SEER shows a reasonably constant contrast and color for the different spectra in the simulation at different SNR.

We then provided a visual representation of the spectra in the above simulation:

“Supplementary Figure 21. Spectra of extreme conditions in SNR-Overlap simulation. The extremes of the simulation utilized in Supplementary Figure 20 are reported here as spectra for comparison. For high Signal to Noise Ratio (a) average spectrum for spectra with peak-maxima distance set to zero and (b) example single spectra from each concentric square region of the simulation (digital levels, DL). (c) Average and (d) single spectra at high SNR for simulation with spectra separated with a peak-to-peak distance of 35.6nm. (e) Reference Simulated Hyperspectral Test Chart with color coded concentric squares. The low SNR simulation spectra are reported here for a peak distance of zero as (f) average and (g) single and for a peak distance of 35.6nm as (h) average and (i) single.”

We quantified the accuracy of SEER, as defined in the Methods section, for the multiple overlaps and SNR, comparing with other visualization methods:

“Supplementary Figure 22. Accuracy of SEER under different spectral overlap and SNR conditions. Accuracy was calculated for different Signal to noise ratios and spectral maxima separation (a) aligned, (b) 8.9nm, (c) 17.8nm, (d) 26.7nm, (e) 35.6nm, starting from the visualizations in Supplementary Figure 20 and corresponding spectra in Supplementary Figure 21. The accuracy is calculated here as the sum of the Euclidean distance of the RGB vectors between pairs of the concentric squares of the simulation, in ratio to the largest color separation (red to green, red to blue, blue to green). A thorough description of accuracy calculation is reported in the Methods section. Each value in the plots represents the average distance of 200^2 pixels; error bars are the standard deviation of normalized accuracy value across all pixels. The average accuracies over multiple SNR conditions for each spectral maxima separation: (a) with highly overlapping spectra SEER provides on average 38.0% for harmonic 1, 50.6% for harmonic 2, while the best performing other comparison here is Gaussian $r=.3$ with an average 26.7%. (b) With a 8.9nm peak-to-peak separation SEER $h=1$ averages 57.0%, SEER $h=2$ 49.6%, other best performing here is Peak Wavelength with 22.2%; (c) with 17.8nm separation SEER $h=1$ averages 57.2%, SEER $h=2$ $60.0 \pm 2.3\%$, other best Gauss $r=.3$ with 26.2% (d) with 26.7nm separation SEER $h=1$ averages 59.9%, SEER $h=2$ 60.4%, other best Gauss $r=.3$ with 32.1%; (e) with well separated spectra 35.6nm apart, SEER $h=1$ averages 66.3%, SEER $h=2$ 66.7%, with other best Gauss $r=.3$ scoring 43.5% in average. “

2. Clarification of FLIM for autofluorescence samples and SEER visualization

The claim we are making is that SEER can visualize differences in the spectra across pixels. In this work we are not claiming to correlate metabolic changes seen by FLIM with spectra, as this was extensively shown in previous works from Melissa Skala's, Enrico Gratton's and Emmanuel Beaurepaire's groups just to cite some (more citations below).

Our aim is to rapidly preprocess the data for the purpose of visualization and exploration of the data, prior to performing a full scale analysis.

It is not clear if the reviewer comment on lack of rigour was referred to the acquisition of data. For clarification on this point, the images were acquired sequentially within seconds, minimizing potential changes in autofluorescence. In previous work, we demonstrated that FLIM and spectra correlate, utilizing the same acquisition setup (Browne et al, "Structural and functional characterization of human stem-cell derived retinal organoids by live imaging". *Invest Ophthalmol Vis Sci.* 2017; 58:3311–3318).

Regarding the concern on weak validation of autofluorescence with FLIM:

- FLIM provides sensitive measurements of the free and protein-bound NAD(P)H ratio and of the redox states (NADH/NAD⁺) of cells, this is known since 1992 (Lakowicz, J. R., et al. Fluorescence lifetime imaging of free and protein-bound NADH. *Proc Natl Acad Sci USA* 89, 1271–1275 (1992)).
- FLIM can be used to distinguish glycolytic and oxidative phosphorylation metabolic states. This has been demonstrated in different systems from multiple groups. The number of works on this topic is too large to list all of them, but here are some publications that describe in depth the relationship between FLIM, autofluorescence and metabolism:
 - a. Stringari C, et al, "Phasor approach to fluorescence lifetime microscopy distinguishes different metabolic states of germ cells in a live tissue", *Proc Natl Acad Sci U S A.* 2011 Aug 16; 108(33):13582-7)
 - b. Sharick et al, "Protein-bound NAD(P)H Lifetime is Sensitive to Multiple Fates of Glucose Carbon" *Scientific Reports* 8, 5456 (2018)
 - c. Browne et al, "Structural and functional characterization of human stem-cell derived retinal organoids by live imaging". *Invest Ophthalmol Vis Sci.* 2017; 58:3311–3318

- d. Melissa C. Skala, et al, “ In vivo multiphoton microscopy of NADH and FAD redox states, fluorescence lifetimes, and cellular morphology in precancerous epithelia Proc Natl Acad Sci USA 2007, 104 (49) 19494-19499
 - e. Sharik et al, “Protein-bound NAD(P)H Lifetime is Sensitive to Multiple Fates of Glucose Carbon”, Scientific Reports volume 8, Article number: 5456 (2018)
 - f. Stringari et al, “ Multicolor two-photon imaging of endogenous fluorophores in living tissues by wavelength mixing”, Scientific Reports volume 7, Article number: 3792 (2017)
 - g. Stringari et al, “Phasor approach to fluorescence lifetime microscopy distinguishes different metabolic states of germ cells in a live tissue”, PNAS August 16, 2011 108 (33) 13582-13587
- FLIM is commonly considered a reference for high precision measurements of NADH. Photonics West SPIE has an entire section named “Metabolism/NADH/FAD/Tryptophan” where numerous groups utilize FLIM for imaging autofluorescence.
 - FLIM-Phasor now has a protocol for performing analysis:
 - h. Ranjit et al, Fit-free analysis of fluorescence lifetime imaging data using the phasor approach, Nature Protocols 13, 1979-2004 (2018)

In an effort to further clarify the FLIM procedure, we have edited the corresponding Main text and Methods section.

Main Text now reads:

“Changes in autofluorescence inside live samples are associated to variations in the ratio of NAD⁺/NADH, which in turn is related to the ratio of free to protein bound NADH³⁹. Despite very similar fluorescence emission spectra, these two forms of NADH are characterized by different decay times (0.4ns free and 1.0-3.4 ns bound)^{35,40-44}. FLIM provides a sensitive measurement for the redox states of NADH and glycolytic/oxidative phosphorylation. Metabolic imaging by FLIM is well established and has been applied for characterizing disease progression in multiple animal models, in single cells and in human as well as to distinguish stem cells differentiation and embryo development⁴¹⁻⁴⁸. Previous work showed Hyperspectral Imaging and FLIM correlate with metabolic changes in cells from retinal organoids⁴⁹. Here, the dashed squares highlight cells with

distinct spectral representation through SEER, a difference which the FLIM image (Figure 4d, Supplementary Figure 14) confirms.”

The methods section now reads:

“ FLIM analysis of intrinsic fluorophores was performed as previously described and reported in detail^{23,35,43}. Phasor coordinates (g,s) were obtained through Fourier transformations. Cluster identification was utilized to associate specific regions in the phasor to pixels in the FLIM dataset according to published protocols³⁵”

Where [35] is the Gratton’s group protocol for “Fit-free analysis of fluorescence lifetime imaging data using the phasor approach” (Fit-free analysis of fluorescence lifetime imaging data using the phasor approach, Ranjit et al, Nature Protocols 13, 1979–2004 (2018)).

Supplementary Figure 14 has been updated:

Supplementary Figure 14 caption has been updated and now reads:

“(c) The line joining free and bound NADH in the phasor plot is known as “metabolic trajectory”, and a shift in the free NADH direction is representative of a more reducing condition and a glycolytic metabolism, while a shift towards more bound NADH is indicative of more

oxidizing conditions and more oxidative phosphorylation, as it has been described in previous studies²⁻⁵.

The extremes of the metabolic trajectory are the lifetimes for NADH free and bound. The parameters for lifetime (τ phase and modulation) are in line with those reported in literature (0.4ns free and 1.0-3.4 ns bound)³⁻⁸.

Where:

[4] is Sharick et al, "Protein-bound NAD(P)H Lifetime is Sensitive to Multiple Fates of Glucose Carbon" Scientific Reports 8, 5456 (2018)

[2] is Browne et al, "Structural and functional characterization of human stem-cell derived retinal organoids by live imaging". Invest Ophthalmol Vis Sci. 2017; 58:3311-3318

[6] is Lakowicz, J. R., et al. Fluorescence lifetime imaging of free and protein-bound NADH. Proc Natl Acad Sci USA 89, 1271-1275 (1992)

[8] is Melissa C. Skala, et al, " In vivo multiphoton microscopy of NADH and FAD redox states, fluorescence lifetimes, and cellular morphology in precancerous epithelia Proc Natl Acad Sci USA 2007, 104 (49) 19494-19499

[7] is Stringari et al, " Multicolor two-photon imaging of endogenous fluorophores in living tissues by wavelength mixing", Scientific Reports volume 7, Article number: 3792 (2017)

[3] is Stringari et al, "Phasor approach to fluorescence lifetime microscopy distinguishes different metabolic states of germ cells in a live tissue", PNAS August 16, 2011 108 (33) 13582-13587

[5] Ranjit et al, Fit-free analysis of fluorescence lifetime imaging data using the phasor approach, Nature Protocols 13, 1979-2004 (2018)

Regarding the uncontrolled nature of our samples:

We understand the concerns of the reviewer. To address this concern, we have added a set of simulations with quantitative comparisons (described in point 1 of this answer).

However, to further clarify, our experimental conditions were optimized on a Zeiss 780 inverted two-photon microscope with ISS FLIMBox frequency domain lifetime system, with particular care to control temperature of instrument and sample mount, timing of experiments, laser power etc. This type of experiment is considered standard in the metabolic imaging field.

In mammals, the tracheal epithelium has a very low turnover (<https://www.ncbi.nlm.nih.gov/pubmed/7005143?dopt=Abstract>). Laboratory animals are expected to display an even lower turnover because of several

factors: air in the animal facility is double filtered and the bedding is changed every week to remove contaminants and particles; temperature and light/dark cycles are finely maintained within a tight range. In mice, it is estimated that every 3 weeks, only 8 percent of the tracheal epithelial cells is replaced (<http://www.pnas.org/content/106/31/12771>). Because of all these factors, the tracheal epithelium at homeostasis was the ideal tissue to image with FLIM. Moreover, to minimize interference of external factors on the data collection and to reach experimental consistency, the tracheal tissue was collected as soon as the animal was euthanized and immediately arranged onto glass slides for imaging. FLIM data nicely fitted our hypothesis and recapitulates findings of previous published work

(<https://www.sciencedirect.com/science/article/pii/S193459091100230X>, <http://dmm.biologists.org/content/3/9-10/545>, <http://dmm.biologists.org/content/3/9-10/545>).

Because of its specific function and anatomical features, the tracheal epithelium is specifically designed to be resilient against small changes of environmental cues for enough time to provide reliable and consistent measurements.

We have edited the main text in the autofluorescence section to clarify this aspect. The text now reads:

“SEER is applied here for visualizing multispectral autofluorescent data of an explant of freshly isolated trachea from a wildtype C57Bl mouse. The tracheal epithelium is characterized by a very low cellular turnover, and therefore the overall metabolic activity is attributable to the cellular function of the specific cell type. Club and ciliated cells are localized in the apical side of the epithelium and are the most metabolically active as they secrete cytokines and chemically and physically remove inhaled toxins and particles from the tracheal lumen. Contrarily, basal cells which represent the adult stem cells in the upper airways, are quiescent and metabolically inactive[37,38]. Because of this dichotomy in activity, the tracheal epithelium at homeostasis constituted the ideal cellular system for testing SEER and validating FLIM imaging. The slight bend on the trachea caused by the cartilage rings allowed us to visualize the mesenchymal collagen layer, the basal and apical epithelial cells and tracheal lumen in a single focal plane. “

We have updated both figure 4 and its caption to simplify interpretation of the metabolic fingerprint. The figure shows a blue spectral shift toward the apical layer:

Caption has been updated for d and e panels, now reads:

“(d) Average spectra for the cells in dashed boxes (1 and 2 in panel c) show a blue spectral shift in the direction of the apical layer. (e) Fluorescence Lifetime Image Microscopy (FLIM) of the sample, acquired using a frequency domain detector validates the interpretation from panel b, Gradient Descent Map, where cells in apical layer exhibit a more Oxidative Phosphorylation phenotype (longer lifetime in red) compared to cells in basal layer (shorter lifetime in yellow) with a more Glycolytic phenotype. The selections correspond to areas selected in phasor FLIM analysis (e, top left inset, red and yellow selections) based on the relative phasor coordinates of NAD⁺/NADH lifetimes. “

There are various comparisons with TrueColor, but the reason for this is not clear since the main contribution of the paper seems to be about enhanced visualisation of SEER images rather the enhancement provided by SEER compared to traditional techniques (which has been reported previously by the authors).

We thank the reviewer for this question. The reviewer is correct, the exact point of the paper is “enhanced visualisation of SEER images”. The work we present here, SEER (Spectrally Encode Enhanced Representations), is largely different from our previous work, HySP (HyperSpectral Phasors).

SEER maps are focused on enhanced representation, not analysis, and hence used as a pre-processing step to aid in further post-acquisition processing steps. They require minimal amount of user interaction (the choice of map) and output color images (RGB, shape [x,y,3]). The process takes seconds for large (Gigabytes) datasets, as we report in our Supplementary Figure 1.

The question SEER answers is the visualization of multidimensional spectral data (x,y,z,t, lambda).

The work we reported previously (Cutrale, Trivedi 2017) is an analysis method for denoising and unmixing spectral signals in longitudinal low signal to noise microscopy

datasets. The analysis is assisted, requires user interaction with ROIs and outputs separate unmixed channels-volumes (monochrome, shape [x,y,z,t]).

The process takes minutes for large (Gigabytes) datasets, as we reported in our previous work.

The question HySP answers is the separation of components in any multidimensional spectral data (x,y,z,t, lambda).

Overall, the main new contribution claimed by this paper seems to be about the mapping of the SEER data for visualisation – although I found that the narrative lacked clarity overall.

The manuscript presents some striking results and there are indications and claims of enhanced ability to discriminate and identify overlapping fluorophores that may be true, however the analysis presented did not demonstrate emphatically and quantitatively that the functionality of the data was improved in a way that could not be achieved by various other contrast-enhancing color mappings of SEER data, or indeed of ICA data, and this may discourage readers from investing in the implementation of these techniques.

We would like to reiterate that SEER is the method we presented here for visualizing the hyperspectral fluorescent data. We believe the points of quantifying and demonstrating the improvements of SEER compared to ICA or other dimensionality reduction (gaussian kernels, peak wavelengths) strategies have been explained in the above figures (Supplementary Figures 15-20).

The main text edits we described above for this reviewer (previous questions) are also focused on addressing this question and improving the quantitative aspect of this work.

My main concern is that enhanced contrast, as apparent in many images, does not equate to enhanced functionality or (accuracy of) discrimination of fluorophores. For example, a simple increase in contrast in some spectral space while also increasing scatter within a cluster, or amplifying noise, might be visually appealing without giving more accurate information or functionality. This paper might achieve a real enhancement that is more than aesthetic, but this is not clearly demonstrated. A quantitative analysis of detectivity of features, or of spectral clustering and classification accuracy would be more convincing to would-be adopters.

Regarding the “increasing scatter within a cluster, or amplifying noise”: an in-depth description of the denoising approach for the phasor method with details on the

poissonian noise and scatter shape was presented in our previous work (Cutrale, Trivedi, Nature Methods 2017) and was the key to accessing longitudinal imaging of multispectral fluorescent samples with reduced laser power and photo-bleaching. In Figure 1 of our previous work (<https://www.nature.com/articles/nmeth.4134>) we show that the phasor denoising approach reduces the scatter within a cluster and reduces the poissonian noise. This was very clearly quantified in our previous work Supplementary Figures 1-5 and extensively described in 5 Supplementary Notes 1-5.

We understand this aspect was not clear, so we edited the main text paragraph page 2 where we introduce the phasor concept to include the denoising component we inherit by utilizing the Phasor platform previously published.

The paragraph reads:

“Exploiting the advantages of the phasor approach, Hyper-Spectral Phasors (HySP) has enabled analysis of 5D hyperspectral time-lapse data semi-automatically as similarly colored regions cluster on the phasor plot. These clusters have been characterized and exploited for simplifying interpretation and spatially lossless **denoising of data**, improving both collection and analysis in low-signal conditions²⁹. Phasor analysis generally explores the 2d-histogram of spectral fingerprints by means of geometrical selectors^{22,23,25,27,28,30}, which is an effective strategy but requires user involvement. “

Regarding the quantitative analysis and classification accuracy points: we would like to again thank this reviewer for her/his suggestions. We believe we have considerably expanded the quantitative aspect of SEER in comparison to other dimensionality reduction techniques utilized in the fluorescence microscopy field.

Particularly, we quantified the processing speeds for SEER, ICA (Supplementary Figure 1), calculated accuracy on simulations for SEER and multiple modalities of RGB visualization (Supplementary Figure 20-22) as well as ICA (Supplementary Figure 23), and compared SEER to the on-screen visualization of a fully analyzed dataset with our previous work (Supplementary Figure 2).

Reviewers' comments:

Reviewer #1 (Remarks to the Author):

The authors have replied to all my major comments and amended to the minor points. In the revised version, Shi, Koo and coworkers have substantially improved the manuscript. They have now included new data in Supplementary Figures, which provide a fair comparison with different 'standard' visualization modes and help the reader to judge the efficacy of the method. They have also added several experimental details that were missing in the previous version. They have also made substantial effort in improving the 'quantitative' aspect of the visualization (although this is a method for "pre-processing visualization", as correctly stated now in the title) and to characterize some technical aspects of the software (i.e. processing time and accuracy compared to other methods), also in response to the comments of the other referees. I strongly recommend acceptance by Nature Communications without further revision.

I only suggest to check again the manuscript for minor errors/typos, like:

- In Supplem. Fig 13 caption: "A representation of the RGB visualization parameters is reported in [...] (j) kernel"
- In Supplem. Fig 19: image (c) is 'flipped'
- In Supp Note 1: "Excitation spectra of fluorescent proteins separated by 140nm is generally not well overlapping"

Reviewer #2 (Remarks to the Author):

I would like to thank the authors for answering all the questions I posed. They properly address the photobleaching and sample movement problems, but I think it will not be straightforward for the readers to use their method for in-vivo microscopy experiments in organs/tissues less transparent than the zebrafish samples, since many effects will probably combine to produce images difficult to be "pre-analyzed" with their visualization method. In this light it would have been more interesting to see how from the colors it is possible to extract quantitative parameters to describe a biological issue.

Therefore, my major concern is again about the quantitative results that can be achieved after applying such a pre-visualization tool. I think that though the method can be useful for the readers, in the end it is not clear how to transform a qualitative information, obtained by visualization, in a quantitative analysis in which new and relevant biological information can be extracted from the acquired images.

In my opinion there is an ambiguity between the rebuttal letter and the manuscript: in the rebuttal the authors claim that the method is useful in the pre-visualization ("the SEER maps are focused on enhanced representation, not analysis, and hence used as a pre-visualization step to aid in further post acquisition processing steps") but in the manuscript they write that the method is quantitative ("We present multiple biological fluorescent samples and highlight SEER's enhancement of specific and subtle spectral differences, providing a fast, intuitive and quantitative way to interpret hyperspectral images during collection, pre-processing and analysis."); "Our method differs from previous implementations, as it focuses instead on providing a quantitatively constructed, holistic pre-processing visualization of large hyperspectral data", "The results of SEER show an enhanced visualization of spectral properties, representing distinct fluorophores with distinguishable pseudo-colors and quantitatively highlighting differences between intrinsic signals during live-imaging"). Moreover, a section is entitled "Quantitative differences can be visualized in combinatorial approaches", but in my opinion they do not show any quantitative data: I expected to find in this section some information about how to obtain numbers from colors.

It is clear that, also thanks to many simulations, the method allows the readers to visualize more accurately some information concerning the sample, but it is not clear, at least in my opinion, how to exploit such visual information in the analysis part, to reach some new biological findings that could not be obtained with other more standard color scales. I think a more quantitative approach is necessary to publish the manuscript in the Nature Communications journal.

Reviewer #3 (Remarks to the Author):

Second review of Visualization of hyperspectral fluorescent data with Spectrally Encoded Enhanced Representations (SEER)

Authors: Wen Shi et al

Overall, this reviewer appreciates the considerable effort that the authors have applied to addressing the comments of all reviewers.

As stated in my previous review, this article describes an enhancement to the visualisation of hyperspectral image processing data produced by a technique called SEER, which spectrally decomposes fluorescence hyperspectral images into spectral components to facilitate visualisation of multiple fluorophores in a single HSI image. The main contribution of this paper seems to be in the mapping of SEER processed data onto a standard color representation with enhanced contrast between fluorophores. Such a capability is already possible using eg TrueColor, PCA and ICA, as the manuscript states. The authors state that their aim is visualisation rather than analysis. I understand therefore that the key niche that SEER could demonstrate is a necessary and sufficient performance for visualisation as described by the authors (therefore equal performance to, a gold standard analysis technique such as ICA, is not essential, it just needs to be good enough for eg rapid and/or real-time appraisal), but with a computational and implementational speed that is much faster than eg ICA.

On the first issue, the authors claim a significantly enhanced accuracy compared to ICA. This is somewhat surprising (and I believe is not necessary) since PCA can be considered to produce a mathematically optimal compression into three basis sets and ICA and PCA tend to produce similar basis sets: ICA aims to reconstruct actual spectra whereas PCA reconstructs orthogonal spectra that normally resemble raw spectra. This merits some discussion and explanation of why a decomposition into Fourier components, (which are not in general an accurate match to actual spectra), produces a higher accuracy than a decomposition into an optimised basis set. The authors use the term accuracy, but it is not clear how the metric is related to 'accuracy'. Indeed the metric appears to be a measure of color contrast rather than accuracy? Also the results of the ICA comparison are not as I would expect: (1) my understanding is that higher values of 'accuracy' would correspond to greater differences in color contrast, but that is not apparent from the curves: for example for the two left graphs of supplementary Fig 23, the highest 'accuracy' occurs at highest SNR according to the graphs – as is expected – but the ICA images actually seem to have the lowest contrast for the highest SNR. (2) More importantly, the results for ICA are not what I expect from ICA – I would expect that each of the different spectra to approximate pure R, G or B as for SEER – however they all look greyish suggesting little discrimination. This suggests that ICA may be being achieved. Also the hue varies systematically with SNR, which is also unexpected. Overall, I believe that either these results suggest an incorrect implementation of ICA or they require careful explanation – at least to convince this reviewer of their validity.

On the issue of speed, it is unfortunate that the authors seem to find some additional challenges in using ENVI for microscopy data – this is not a problem that we have

experienced and indeed ENVI is also recommended for HSI microscopy by the distributors. Nevertheless the comparison of SEER speed with *fastICA* should be sufficient, but it should be made more clear in the text that in both cases the algorithms are implemented in Python (an interpreted language and hence potentially one to two orders of magnitude slower than a compiled implementation in eg ENVI) and hence that the speed comparison is valid.

Overall, I think that SEER is an interesting approach to visualisation, but the comparison with a gold standard of ICA does not appear to me to be valid.

Minor issues

In the caption to Supp Fig 20 – the values ' $(.5*10)^1 - (.5*10)^4$ ' seem to be incorrectly written.

Point by point response

- Reviewer's points
- "manuscript text"
- Modifications in the manuscript highlighted.

Reviewers' comments:

Reviewer #1 (Remarks to the Author):

The authors have replied to all my major comments and amended to the minor points. In the revised version, Shi, Koo and coworkers have substantially improved the manuscript. They have now included new data in Supplementary Figures, which provide a fair comparison with different 'standard' visualization modes and help the reader to judge the efficacy of the method. They have also added several experimental details that were missing in the previous version. They have also made substantial effort in improving the 'quantitative' aspect of the visualization (although this is a method for "pre-processing visualization", as correctly stated now in the title) and to characterize some technical aspects of the software (i.e. processing time and accuracy compared to other methods), also in response to the comments of the other referees. I strongly recommend acceptance by Nature Communications without further revision.

I only suggest to check again the manuscript for minor errors/typos, like:

- In Supplem. Fig 13 caption: "A representation of the RGB visualization parameters is reported in [...] (j) kernel"
- In Supplem. Fig 19: image (c) is 'flipped'
- In Supp Note 1: "Excitation spectra of fluorescent proteins separated by 140nm is generally not well overlapping"

We thank the reviewer for her/his contributions. We have performed the corrections listed above and highlighted them in the text.

Reviewer #2 (Remarks to the Author):

I would like to thank the authors for answering all the questions I posed. They properly address the photobleaching and sample movement problems, but I think it will not be straightforward for the readers to use their method for in-vivo microscopy experiments in organs/tissues less transparent than the zebrafish samples, since many effects will probably combine to produce images difficult to be "pre-analyzed" with their visualization method. In this light it would have been more interesting to see how from the colors it is possible to extract quantitative parameters to describe a biological issue.

We thank the reviewer for his/her contributions and for helping us further improve the manuscript. In this part we would like to address these two comments from the reviewer since they lead to his/her main concern in the subsequent section:

1. "it will not be straightforward for the readers to use their method for in-vivo microscopy experiments in organs/tissues less transparent than the zebrafish samples, since many effects will probably combine to produce images difficult to be "pre-analyzed" with their visualization method"

2. “it would have been more interesting to see how from the colors it is possible to extract quantitative parameters to describe a biological issue”

Response:

1. There appears to be a misunderstanding between the purpose of SEER (pre-processing/visualization) and the problems related to in-vivo optical microscopy (acquisition of the data). The method we present here is aimed at visualizing data during acquisition to optimize the experimental pipeline. The method is not aimed at solving problems related to the optics and microscopy used in the acquisition. This point was stated very extensively in our previous answers to this reviewer (Page 43 Response_to_Referees_Letters_20181211.pdf):

“We would like to reiterate that this work is focused on the pre-processing enhanced representation, rather than analysis, of the multidimensional hyperspectral dataset that can be useful in guiding the experimentalists during acquisition and analysis of the data.”

as well as in the abstract of the manuscript,

“Understanding and visualizing these large multi-dimensional datasets during acquisition and pre-processing can be challenging. [...] Here we present Spectrally Encoded Enhanced Representations (SEER), an approach for improved and computationally efficient simultaneous color visualization of multiple spectral components of hyperspectral fluorescence images. “

in the introduction:

“To optimize experimental time, it is advantageous to perform an informed visualization of the spectral data during acquisition, especially for lengthy time-lapse recordings, and prior to performing analysis.”

in the discussion:

“Some limitations that should be considered are that SEER pseudo-color representation sacrifices the “true color” of the image, creating inconsistencies with the human eyes’ expectation of the original image and does not distinguish identical signals arising from different biophysical events (Supplementary Note 2).”

in the conclusion:

“SEER has the potential of optimizing the experimental pipeline, from data collection during acquisition to data analysis”.

The reviewer’s statement “many effects will probably combine to produce images difficult to be "pre-analyzed" with their visualization method” was answered in the previous round of review by editing the discussion section as above quoted and by adding an entire new Supplementary Note 2 specifically discussing the color visualization limitations for SEER. Particularly the Supplementary Note 2 states:

“ [...] In the scenario where spectra derived from two different effects are exactly the same, for example, a case where low protein expression is on an outer layer and a high level expression is attenuated at a deeper level, the phasor method and the SEER maps, in their current

implementation, will not be able to differentiate between the two effects. The separation of these two effects is a different and complex problem which depends on the optical microscopy components, the sample, the labels, the multispectral imaging approach and more factors in the experimental design, and we believe this separation falls outside of the scope of this paper and constitutes a project on its own.”

The statement “many effects will probably combine to produce images difficult to be "pre-analyzed" with their visualization method “ suggests that the images acquired through optical microscopy will return spectral data that would be too noisy either from scattering or autofluorescence. It is true that if the spectral data returned is so noisy that the signal is buried under the noise, then our method would not work. We agree with the reviewer that this was not explicitly stated in the manuscript. We have further edited Supplementary Note 2 which now reads:

“If signal is indistinguishable from noise, SEER maps will assign the same color to these spectra.”

However, in this scenario of signal-equal-to-noise, it is not possible to extract this information with any current method, making the experiment unsuccessful in the first place. We believe it is beyond the scope of this manuscript to solve problems related to optical opacity of samples or related to incorrect acquisition.

If the statement by the reviewer, instead, means that the data has signal which can be acquired, but those “many effects” will specifically prevent our phasor method and SEER from working, then the demonstration that that is not the case has been extensively shown in our previous rebuttal (with bleaching and z-stack demonstrations) and the original HySP paper.

In regard to imaging of samples “less transparent than zebrafish”, we would like to remind the reviewer that figure 3 in the main text is a mouse sample, freshly excised. However, to address the concerns of the reviewer, we have acquired another set of volumes in freshly excised tracheal tissue of unlabeled wild type C57Bl mouse and rendered them. As the reviewer can see, even under the complex imaging conditions of weak autofluorescent signals, volumes can be acquired in samples “less transparent than zebrafish” without any evident effect combining “to produce images difficult to be "pre-analyzed" with their visualization method ”.

From our observation (here and in our previous answers to this reviewer), signal fades into background before any appreciable effect arise that might complicate analysis with SEER. To address this concern, we have added a Supplementary Figure 26:

“**Supplementary Figure 26. Autofluorescence visualization in volumetric data of unlabeled freshly isolated mouse tracheal explant.** A tiled z-stack (x,y,z) imaged with multi-spectral two-photon microscopy (740nm excitation, 32 wavelength bins, 8.9nm bandwidth, 410-695 nm detection) is here visualized as a single (x,y) z-slice SEER RGB Gradient Descent Max Morph mask at (a) 43μm, (b) 59μm, (c) 65μm depth. Color differences between basal and apical layer cells are maintained at different depths, with consistent hue for each of the cell layers. Colorbar represents the main wavelength associated to one color in nanometers. Volume renderings presented as SEER Alpha Color renderings for (d) top-down (x,y) view, (e) Lateral (y,z) view and (f) zoomed-in lateral (y,z) view show the shape and the 98μm thickness of the unlabeled tissue sample.”

We reference the figure in the section titled “Color maps enhance different spectral gradients in biological samples” in the paragraph about Figure 4. The edit reads:

“The visualization improvement is maintained against different implementations of RGB visualization (Supplementary Figure 13) and at different depths in volumetric datasets (Supplementary Figure 26).”

In this image we show a z-stack in mouse trachea (non-cleared, freshly excised tissue) imaged with multispectral two-photon microscopy at 740nm to excite autofluorescence. a,b and c show single slices (x,y) at different depths. The SEER RGB masks show information consistent with Figure 4 in the manuscript, with cells in basal and apical layer exhibiting different spectral fluorescent signals, here reflected with different hue. The apical and basal layer colors remain respectively the same throughout the z-slices. Likewise, the basal layer also maintains the same color in this representation. d,e and f show SEER Alpha Color renderings of the volume, to further illustrate the 3D aspect of this dataset and how SEER is not affected by this non-cleared mouse sample. The sample fades to background (black) after ~100um. When only background (black) is present, SEER (and arguably any analysis) will not show any difference in the data, because there is no difference to show.

2. We agree it would be interesting to have a different and more analysis-centric method, however we believe we have already established the need for our method (navigating hyperspectral data) and explained how it can be used in a pipeline toward more quantitative results (example: utilizing standard HySP or unmixing to separate contributions of different fluorophores in multiple channels, then applying other standard methods to determine interactions between them). The previous response to this reviewer's same question (Page 43 Response_to_Referees_Letters_20181211.pdf) extensively explains these points. In short, the focus of SEER is currently to visualize data better to aid in experimental setting and guide the choice of (a separate) analysis method. Extracting quantitative parameters to describe a biological issue from colors, while an exciting idea, is a complex new project, which is beyond the scope of this manuscript, but can likely be implemented starting from SEER results.

Therefore, my major concern is again about the quantitative results that can be achieved after applying such a pre-visualization tool. I think that though the method can be useful for the readers, in the end it is not clear how to transform a qualitative information, obtained by visualization, in a quantitative analysis in which new and relevant biological information can be extracted from the acquired images.

We understand the concern of this reviewer. The position of SEER in the experimental pipeline is ideally during acquisition and before analysis. The purpose is to visualize data better to aid the experimentalist in the choice of experimental settings and type of analysis.

The **problem**: spectral data is large. Visualizing data informatively during acquisition is important for improving experimental output. This is a dimensionality reduction problem, combined with low signal-to-noise fluorescent data.

The **hypothesis**: an improved view of the data can consequently improve the whole experiment starting from a better microscopy acquisition and a more educated choice analysis (which will depend on the user experiment).

The **current solutions**: PCA/ICA are computationally expensive and time consuming, especially for large data. Rapid RGB visualizations (example Figure 5a) do not discriminate subtle differences in spectra.

Our **solution**: visualization of large data through the Phasor approach combined with spatially lossless denoising of data.

While we believe firmly there is no "one step solution" for every experiment, we believe that one common problem is visualizing the data in a more informative way. As such we focus on solving the visualization problem.

The **problem** is explained in the introduction paragraph:

"The drawback of acquiring this vast multidimensional spectral information is an increase in complexity and computational time for the analysis, showing meaningful results only after lengthy calculations. To optimize experimental time, it is advantageous

to perform an informed visualization of the spectral data during acquisition, especially for lengthy time-lapse recordings, and prior to performing analysis. Such preprocessing visualization allows scientists to evaluate image collection parameters within the experimental pipeline as well as to choose the most appropriate processing method. “

For better reiterating the problem, in the Spectrally Encoded Enhanced Representations (SEER) paragraph:

“Discriminating the very similar spectra within the original acquisition space is challenging using standard multispectral dataset visualization approaches”

The **hypothesis** is explained in the introduction paragraph:

“The results of SEER show an enhanced visualization of spectral properties, representing distinct fluorophores with distinguishable pseudo-colors and quantitatively highlighting differences between intrinsic signals during live-imaging. SEER has the potential of optimizing the experimental pipeline, from data collection during acquisition to data analysis, greatly improving image quality and data size. “

The **current solutions** are explained in the introduction paragraph:

“A drawback to approaches such as Singular Value Decomposition to compute PCA bases and coefficients or generating the best fusion weights is that it can take numerous iterations for convergence²¹. Considering that fHSI datasets easily exceed GigaBytes range and many cross the TeraBytes threshold⁶, such calculations will be both computationally and time demanding. Furthermore, most visualization approaches have focused more on interpreting spectra as RGB colors and not on exploiting the full characterization that can be extracted from the spectral data. “

Our solution is also explained in the introduction paragraph:

“In this work, we build maps based on Phasors (Phase Vectors). The Phasor approach has multiple advantages deriving from its properties²²⁻²⁵ [...] Exploiting the advantages of the phasor approach, [...] clusters have been characterized and exploited for simplifying interpretation and spatially lossless denoising of data, improving both collection and analysis in low-signal conditions²⁹. [...] The solution we propose extracts from both the whole phasor and image to reconstruct a “one shot” view of the data and its intrinsic spectral information. Spectrally Encoded Enhanced Representations (SEER) is a dimensionality reduction-based approach, achieved by utilizing phasors, and automatically creating spectrally representative color maps. “

The examples in figure 5 shows a very clear example of the problem: during acquisition Figure 5a is what a current experimentalist sees. Setting of acquisition parameters on the instrument and the choice of analysis are often challenging for live samples. Some examples of how this improved visualization can improve the experimental pipeline for a scientist:

- a. identifying the independent spectral components for linear unmixing or HySP analysis:

For fluorescent samples, spectra can have subtle variations on a sample-to-sample basis which depend on multiple factors (pH, expression level, sensitivity of the instrument, excitation wavelength). A “canned” spectrum acquired from literature or from a different instrument can yield sub-par results. SEER visualization can allow for identification of pixels containing good spectra based on uniformity of color and anatomical position

- b. identifying areas of interest for the acquisition, as often the developmental biology experimentalist needs to “compromise” resolution for speed.

Scientific discovery is often aided by the ability to distinguish new events within noisy or spectrally similar areas. In many occasions the microscopist needs to compromise on the total area to image because of the level of expression or the size of the area with respect to the resolution desired. This more sensitive visualization is more robust to noisy environments and allows faster scanning, consequently allowing larger areas for acquisition and/or higher resolution.

- c. understanding the patterns and location of labels, with respect to intrinsic, often undesired, signals

Autofluorescence is present in the majority of biological samples. Being able to distinguish the location of a signal of interest from the surrounding spectrally similar pixels can aid experiments where fluorescent labels emit in the autofluorescence range or where multiple labels spectrally overlap.

- d. understanding the different intrinsic signals and their relationship with metabolism

As we show in figure 4 and in the text in the manuscript, there is information in intrinsic signals. This information relates to the metabolism of tissues. While standard visualization approaches show a uniform color (see below) or require different imaging strategies (FLIM), SEER can capture these subtle differences using spectra. Better understanding intrinsic signals and metabolism of the sample can improve the understanding of developmental processes.

- e. there are color-based segmentation algorithms available through Fiji (Fiji is Just ImageJ) (Plugins -> Segmentation -> Color Clustering)

While it is beyond the scope of this manuscript, there is an extensive library of color-based algorithms that perform analysis of data. A large portion of these algorithms are aimed for brightfield pathology slides images but with SEER, in novel projects, these algorithms could be applied to perform completely new experiments or to reprocess observations on already spectrally acquired data.

In our experience as an imaging center which services one of the largest universities in Southern California, these problems are extremely common for microscopy users.

In my opinion there is an ambiguity between the rebuttal letter and the manuscript: in the rebuttal the authors claim that the method is useful in the pre-visualization ("the SEER maps are focused on enhanced representation, not analysis, and hence used as a pre-visualization step to aid in further post acquisition processing steps") but in the manuscript they write that the method is quantitative ("We present multiple biological fluorescent samples and highlight SEER's enhancement of specific and subtle spectral differences, providing a fast, intuitive and quantitative way to interpret hyperspectral images during collection, pre-processing and analysis."; "Our method differs from previous implementations, as it focuses instead on providing a quantitatively constructed, holistic pre-processing visualization of large

hyperspectral data", "The results of SEER show an enhanced visualization of spectral properties, representing distinct fluorophores with distinguishable pseudo-colors and quantitatively highlighting differences between intrinsic signals during live-imaging").

We believe the confusion arises from our choice of words. In saying that SEER is “quantitatively constructed” our intention was to explain to the reader that the maps developed for SEER are not random and are not subjective. These maps have been mathematically constructed using our knowledge of how the phasor transform method works in order to provide an easy way to parse the admittedly difficult to navigate phasor plot for multiple spectral identities in the data.

SEER performs a variety of numerical calculations (phasor, denoising), analyzing the spectral similarities/differences, with the purpose of visualizing the multidimensional data better.

To avoid further misunderstanding we have made the following changes:

In the abstract:

*“We present multiple biological fluorescent samples and highlight SEER’s enhancement of specific and subtle spectral differences, providing a fast, intuitive and **mathematical** way to interpret hyperspectral images during collection, pre-processing and analysis.”*

In the introduction:

*“Our method differs from previous implementations^{22,23,25,27,28,30}, as it focuses instead on providing a **mathematically** constructed, holistic pre-processing visualization of large hyperspectral data.”*

*“The results of SEER show an enhanced visualization of spectral properties, representing distinct fluorophores with distinguishable pseudo-colors and **mathematically highlighted** differences between intrinsic signals during live-imaging.”*

In the Standard Reference Maps:

*“The ability of the Standard Reference Maps to capture either different ratios of labels or changes in a label, such as a calcium indicator, offers the dual advantage of providing a rapid, **mathematically improved** overview and simplifying the comparison between multiple samples. “*

We also added a clarification in the Discussion section that describes the limitations of SEER:

“SEER is currently intended as a pre-processing visualization tool and, currently, is not utilized for quantitative analysis. Combining SEER with novel color-image compatible segmentation algorithms^{58,59} might expand the quantitative capabilities of the method. “

Moreover, a section is entitled "Quantitative differences can be visualized in combinatorial approaches", but in my opinion they do not show any quantitative data: I expected to find in this section some information about how to obtain numbers from colors.

We agree with the reviewer, the combinatorial approaches section title does not exactly match the paragraphs contained in that section. We changed the title to better match the content:

*“**Spectral** differences can be visualized in combinatorial approaches“*

It is clear that, also thanks to many simulations, the method allows the readers to visualize more accurately some information concerning the sample, but it is not clear, at least in my opinion, how to exploit such visual information in the analysis part, to reach some new biological findings that could not be obtained with other more standard color scales.

This point is partially answered 2 sections above. The hypothesis that guided our development of SEER is that an improved view of the data can consequently improve the whole experiment starting from a better microscopy acquisition and a more educated choice analysis (which will depend on the user experiment).

This visualization requires improved performance at low SNR and reduced computational time.

In an experimental setting for microscopy imaging which is dependent on time (e.g. developmental biology, biological processes that happen at specific time points), often the nature of the sample does not allow sufficient time to run lengthy analysis for visualizing the data currently being acquired.

Similarly, after acquisition, before analysis, an experimentalist could run long visualization calculations, but considering the generally large number of samples required to build statistics for biological data, the cumulative added analysis time can considerably extend the experimental pipeline.

Regarding the use of “more standard color scales”, as the reviewer writes, “it is clear that the method allows the readers to visualize more accurately some information concerning the sample”.

Some examples of how this improved visualization can improve the experimental pipeline for a scientist:

- f. identifying the independent spectral components for linear unmixing or HySP analysis:

For fluorescent samples, spectra can have subtle variations on a sample-to-sample basis which depend on multiple factors (pH, expression level, sensitivity of the instrument, excitation wavelength). A “canned” spectrum acquired from literature or from a different instrument can yield sub-par results. SEER visualization can allow for identification of pixels containing good spectra based on uniformity of color and anatomical position

- g. identifying areas of interest for the acquisition, as often the developmental biology experimentalist needs to “compromise” resolution for speed.

Scientific discovery is often aided by the ability to distinguish new events within noisy or spectrally similar areas. In many occasions the microscopist needs to compromise on the total area to image because of the level of expression or the size of the area with respect to the resolution desired. This more sensitive visualization is more robust to noisy environments and allows faster scanning, consequently allowing larger areas for acquisition and/or higher resolution.

- h. understanding the patterns and location of labels, with respect to intrinsic, often undesired, signals

Autofluorescence is present in majority of biological samples. Being able to distinguish the location of signal of interest from the surrounding spectrally similar pixels can aid experiments where fluorescent labels emit in the autofluorescence range or where multiple labels spectrally overlap.

- i. understanding the different intrinsic signals and their relationship with metabolism

As we show in figure 4 and in the text in the manuscript, there is information in intrinsic signals. This information relates to metabolism of tissues. While standard visualization approaches show a uniform color (see below) or require different imaging strategies (FLIM), SEER can capture these subtle differences using spectra. Better understanding intrinsic signals and metabolism of the sample can improve the understanding of developmental processes.

- j. there are color-based segmentation algorithms available through Fiji (Fiji is Just ImageJ) (Plugins -> Segmentation -> Color Clustering)

While it is beyond the scope of this manuscript, there is an extensive library of color-based algorithms that perform analysis of data. A large portion of these algorithms were aimed for brightfield pathology slides images but with SEER, in novel projects, these algorithms could be applied to perform completely new experiments or to reprocess observations on already spectrally acquired data.

These problems are extremely common for microscopy users.

The application in figure 4 (which is an uncleared mouse tissue), on autofluorescence, is one clear example of an application of this visualization that can lead to improved biological finding, otherwise impossible with “standard color scales”. We believe no useful information can be seen in this “standard color scale”, despite the anatomical knowledge of the basal and apical layer:

We have described this point in the figure caption:

“These intrinsic molecules have been used as reporters for metabolic activity in tissues by measuring their fluorescence lifetime, instead of wavelength, due to their closely overlapping emission spectra. This overlap increases the difficulty in distinguishing spectral changes when utilizing a (a) TrueColor image display (Zen Software, Zeiss, Germany) “

The SEER visualization, instead, shows differences between apical and basal layer (in this sample):

We explained this also in the caption:

“(b) The gradient descent morphed map shows differences between apical and basal layers, suggesting different metabolic activities of cells based on the distance from the tracheal airway. Cells on the apical and basal layer (dashed boxes) are rendered with distinct color groups. Colorbar represents the main wavelength associated to one color in nanometers.”

These differences were validated in two ways:

1. using a different method (not a different map), Fluorescence Lifetime Imaging Microscopy (FLIM) (Figure 4e)
2. showing the actual spectra (Figure 4d)

While FLIM does certainly provide great information, FLIM systems are not as diffused as, for example the spectral detectors commercially available in Zeiss confocal/2p systems or the lambda mode available in Leica confocal/2p systems. The fact that differences highlighted by the colors in the SEER maps correlate with the information seen in FLIM opens a new set of potential experiments that could not be performed using the “standard color scale”.

We want to reiterate the fact that we are presenting a pre-processing visualization method, not a quantitative analysis tool, nor a biological problem. This figure shows potential biological findings that cannot be obtained with “standard color scales.”; one potential project, which is beyond the scope of this paper, could evaluate changes of metabolism in tissue under different perturbations from chemicals.

I think a more quantitative approach is necessary to publish the manuscript in the Nature Communications journal.

We respectfully disagree. We believe we have extensively established the need for our method (navigating hyperspectral data) and explained how it can be used in a pipeline toward more quantitative results (example: utilizing standard HySP or unmixing to separate contributions of different fluorophores in multiple channels, then applying other standard methods to determine interactions between them).

The previous response to this reviewer's same question (Page 43 Response_to_Referees_Letters_20181211.pdf) extensively explains these same points.

In short, the focus of SEER is to visualize data better in order to aid in determining experimental parameters and guide the choice of (a separate) analysis method.

Extracting quantitative parameters to describe a biological issue from colors, while an exciting idea, is a complex new project, which is beyond the scope of this manuscript, but can likely be implemented starting from SEER results.

Reviewer #3 (Remarks to the Author):

Second review of Visualization of hyperspectral fluorescent data with Spectrally Encoded Enhanced Representations (SEER)

Authors: Wen Shi et al

Overall, this reviewer appreciates the considerable effort that the authors have applied to addressing the comments of all reviewers.

As stated in my previous review, this article describes an enhancement to the visualisation of hyperspectral image processing data produced by a technique called SEER, which spectrally decomposes fluorescence hyperspectral images into spectral components to facilitate visualisation of multiple fluorophores in a single HSI image. The main contribution of this paper seems to be in the mapping of SEER processed data onto a standard color representation with enhanced contrast between fluorophores. Such a capability is already possible using eg TrueColor, PCA and ICA, as the manuscript states.

The authors state that their aim is visualisation rather than analysis. I understand therefore that the key niche that SEER could demonstrate is a necessary and sufficient performance for visualisation as described by the authors (therefore equal performance to, a gold standard analysis technique such as ICA, is not essential, it just needs to be good enough for eg rapid and/or real-time appraisal), but with a computational and implementational speed that is much faster than eg ICA.

On the first issue, the authors claim a significantly enhanced accuracy compared to ICA. This is somewhat surprising (and I believe is not necessary) since PCA can be considered to produce a mathematically optimal compression into three basis sets and ICA and PCA tend to produce similar basis sets: ICA aims to reconstruct actual spectra whereas PCA reconstructs orthogonal spectra that normally resemble raw spectra. This merits some discussion and explanation of why a decomposition into Fourier components, (which are not in general an accurate match to actual spectra), produces a higher accuracy than a decomposition into an optimised basis set.

We thank the reviewer for the further questions. We agree with the reviewer that ICA can be considered to produce mathematically optimal compression into three basis set for some specific data.

However, this is true for estimating a noise free model, not for noisy data [citation: Aapo Hyvärinen. Fast ICA for noisy data using Gaussian moments. In Circuits and Systems, 1999. ISCAS'99. Proceedings of the 1999 IEEE International Symposium on, volume 5, pages 57–61.].

The short answer is that PCA/ICA are known to underperform in low SNR noisy data. The data we analyze here is low signal, photon starved fluorescent data. SEER performs compression (with averaging of similar spectra) and spatially lossless denoising.

We would like to reiterate the problem here under analysis:

1. We are working with generally low signal to noise fluorescent data.

This is explained in the discussion section:

“SEER overcomes the challenges in visualization and analysis deriving from low signal-to-noise images, such as intrinsic signal autofluorescence imaging”

In the Introduction:

“These clusters have been characterized and exploited for simplifying interpretation and spatially lossless denoising of data, improving both collection and analysis in low-signal conditions²⁹. ”

We also edited the Modes (Scale and Morph) section to clarify this point:

“The contribution of these **generally low signal-to-noise** intrinsic signals to the overall fluorescence is generally called autofluorescence⁷. Hyperspectral imaging and HySP²⁹ can be employed to diminish the contribution of autofluorescence to the image. The improved sensitivity of the phasor, however, enables autofluorescence to become a signal of interest and allows for exploration [..]”.

The simulations comparing ICA with SEER (requested by this reviewer) had different signal to noise conditions as stated in the discussion:

“shows an increased accuracy (Methods) for SEER to associate distinct colors to closely overlapping spectra under different noise conditions (Supplementary Figure 21). “

The supplementary Figure 21 added in the previous rebuttal (also reported below) shows example spectra from the simulation:

“**Supplementary Figure 21. Spectra of extreme conditions in SNR-Overlap simulation.** The extremes of the simulation utilized in Supplementary Figure 20 are reported here as spectra for comparison. For high signal-to-noise ratio (a) average spectrum for spectra with peak-maxima distance set to zero and (b) example single spectra from each concentric square region of the simulation (digital levels, DL). (c) Average and (d) single spectra at high SNR for simulation with spectra separated with a peak-to-peak distance of 35.6nm. (e) Reference Simulated Hyperspectral Test Chart with color coded concentric squares. The low SNR simulation spectra are reported here for a peak distance of zero as (f) average and (g) single and for a peak distance of 35.6nm as (h) average and (i) single.”

We believe we have clearly stated the conditions of low SNR and the different SNR settings for simulation.

2. **The solution we propose, SEER, accomplishes visualization of large data through the Phasor approach combined with spatially lossless spectral denoising of data.**

This is explained in the introduction paragraph:

“In this work, we build maps based on Phasors (Phase Vectors). The Phasor approach has multiple advantages deriving from its properties^{22–25} [...] Exploiting the advantages of the phasor approach, [...] clusters have been characterized and exploited for simplifying interpretation and spatially lossless denoising of data, improving both collection and analysis in low-signal conditions²⁹. [...]”

The solution we propose extracts from both the whole phasor and image to reconstruct a “one shot” view of the data and its intrinsic spectral information. Spectrally Encoded Enhanced Representations (SEER) is a dimensionality reduction-based approach, achieved by utilizing phasors, and automatically creating spectrally representative color maps. “

The spatially lossless denoising of spectral data was explained to this reviewer in the previous response (Response_to_referees_Letters_20181211.pdf, page 70-71), from which we quote:

“An in-depth description of the denoising approach for the phasor method with details on the poissonian noise and scatter shape was presented in our previous work (Cutrale, Trivedi, Nature Methods 2017) and was the key to accessing longitudinal imaging of multispectral fluorescent

samples with reduced laser power and photo-bleaching.

In Figure 1 of our previous work (<https://www.nature.com/articles/nmeth.4134>) we show that the phasor denoising approach reduces the scatter within a cluster and reduces the poissonian noise. This was very clearly quantified in our previous work Supplementary Figures 1-5 and extensively described in Supplementary Notes 1-5.” [of our previous Nature Methods work].

The denoising effect was also described in the text of the Modes (Scale and Morph) section (page 10):

“Implementation of 1x to 5x spectral denoising filters²⁹ further enhances visualization (Supplementary Figure 11,12)”

And in the Supplementary Figures 11 and 12 (here we show Supplementary Figure 11):

“**Supplementary Figure 11. Spectral denoising effect on Angular and Radial maps visualization of standard overlapping spectra (Simulated Hyperspectral Test Chart II, Supplementary Figure 8).** Phasor spectral denoising affects the quality of data along the spectral dimension, without changing intensities. Here second harmonic is utilized for calculations. Noisy data appears as a spread cluster on the phasor, here shown overlaid with the (a) Angular map and (b) Radial map, with the overlaid visualization exhibiting salt and pepper noise. (c, d) When denoising is applied on the phasor, the cluster spread is reduced, providing greater smoothing and less noise in the simulated chart. (e, f) Increasing the number of denoising filters results in a clearer distinction between the three spectrally different areas in each block of the simulation. (a, c, e) In Max Morph Mode, each denoising filter introduces a shift of the apex of the map, changing the reference center of the color palette (b, d, f) In Scale Mode, the less scattered phasor cluster makes maximum use of the reference maps, enhancing the contrast (d, f) of the rendered SHTC.”

We believe the denoising aspect is underlined quite extensively in the manuscript.

3. PCA/ICA are known to underperform in low SNR conditions.

3a. From this work of Mike Davies (University of London) “ Identifiability Issues in Noisy ICA (IEEE Signal Proc. Lett., vol. 11, 5, 2004)”

we quote:

“The majority of work to date on independent component analysis (ICA) has assumed that the independent sources are observed via some unknown mixing process in the absence of noise. “
“The aim of this letter is to point out a weakness in generalizing ICA to include noise”

3b. From S. Ikeda “Independent component analysis for noisy data — MEG data analysis”
(Neural Networks Volume 13, Issue 10, December 2000, Pages 1063-1074):
we quote:

“when we apply ICA to real-world problems, the situation is different from the above ideal case. In many cases, we cannot ignore noise [...] [ICA] is not enough to describe the problem. It is pointed out that, especially when the number of the sources is small, one cannot have a good solution generally”

3c. From Jaakko Sarela “Overlearning in Marginal Distribution-Based ICA: Analysis and Solutions” (Journal of Machine Learning Research 4 (2003) 1447-1469):

we quote:

“The present paper is written as a word of caution, with users of independent component analysis (ICA) in mind, to overlearning phenomena that are often observed. [...] These algorithms [ICA] can be seen to maximise the negentropy of the source estimates. The first kind of overlearning results in the generation of spike-like signals, if there are not enough samples in the data or there is a considerable amount of noise present.”

3d. Overcoming ICA problems with noise are still under investigation and appear to require ad-hoc solution. For example in recent pre-print from Niklas Pfister at ETH “Robustifying Independent Component Analysis by Adjusting for Group-Wise Stationary Noise” [<https://arxiv.org/pdf/1806.01094.pdf>]

we quote:

“We show that our general noise model allows to perform ICA in settings where other noisy ICA procedures fail.”

3e. Fereidouni’s work (J. Biophotonics 7, No. 8, 589–596 (2014)) also discusses the advantages of Phasor compared to PCA in the case of fluorescence data:

“Conventional spectral analyses methods like linear unmixing and principal component analyses methods have been used to analyze conventional spectral imaging data from fluorescently labeled specimens. These techniques are, however, less suitable for analyzing AF [=Autofluorescence] and SHG [=Second Harmonic Generation] images. In the case of autofluorescence, the number of components is large and often the spectral differences are only minor.”

4. Noise in fluorescence data is not only additive and is poissonian in nature.

From the article of Aapo Hyvärinen (the main and only author of the reference paper on FastICA Neural Networks, 10(3):626-634, 1999.) “Fast ICA for noisy data using gaussian moments”

(Circuits and Systems, 1999. ISCAS '99. Proceedings of the 1999 IEEE International Symposium on Volume: 5):

“We may express the model as

$$x = As + n \quad (1)$$

where $x = (x_1; x_2; \dots; x_m)$ is the vector of observed random variables, $s = (s_1; s_2; \dots; s_n)$ is the vector of the latent variables called the independent components, and A is an unknown constant matrix, called the mixing matrix. The vector n is noise, and is often omitted; most research has concentrated on the problem of estimating the noise-free model [1, 2, 3, 5, 15, 9, 17, 22]. For simplicity, we make in this paper some assumptions that are not strictly necessary: 1) the dimension of s equals the dimension of x , i.e. $n = m$, 2) the noise n is gaussian and 3) the noise covariance matrix is known.”

As you can see from the above equation, in ICA the noise is often omitted and is considered only additive and Gaussian.

Fluorescence noise cannot be omitted, is not only additive, and is poissonian in nature.

In general, measured fluorescent signal S [photon/pixel] will depend on Quantum Efficiency QE , background I_b , noise factor F_n , readout noise N_r and gain G of the system

The noise in particular will be proportional to:

$$Noise \propto \sqrt{QE * (S + I_b) * F_n^2 + \left(\frac{N_r}{G}\right)^2}$$

Where the ideal fluorescent signal ($QE*S$) is embedded in the noise. The only additive component is the background noise. The remaining parts of the noise are poissonian and depend on the noise itself (Hamamatsu Photonics K.K. Photomultiplier Technical Handbook. (1994) Hamamatsu Photonics K.K.).

It is true that for large numbers the poissonian distribution approaches normal distribution, becoming closer to the gaussian noise described in the ICA modeling. However, this is not the general case in fluorescence.

In our instrument (Zeiss LSM 780) at 800 gain, the conversion factor photon to Digital Levels:

$$\frac{single_{photon}}{Digital_{Levels}} (@800\ gain) = 201$$

to give an idea of the measurements, for autofluorescence the high DL intensities peak around 4000 DL, hence ~20 photons or less.

For fluorescent data, generally a good signal averages around 200 photons. Hence the gaussian distribution approximation of a poissonian is generally not achieved.

This photon-starved signal has an energy distribution which is the fluorescence emission spectrum of the protein/dye. As a consequence, photon starved spectra appear different from the expected.

A purely mathematical approach (ICA) appears to not properly account for these factors and performs a dimensionality reduction with decreased quality as compared to the phasor approach.

5. Phasor is not simply “*a decomposition into Fourier components*”

Phasor has a number of advantages which make it attractive for multispectral fluorescence imaging. It is more than a simple decomposition in Fourier components and has been demonstrated to perform well especially in noisy fluorescent conditions.

This is anticipated in the introduction:

“Exploiting the mathematical properties of the phasor method, we transform the wavelength space into information-rich color maps for RGB display visualization. We present multiple biological fluorescent samples and highlight SEER’s enhancement of specific and subtle spectral differences,”

The first paragraph on page 3 in the introduction explains the advantages of the phasor, which is not just a decomposition into Fourier components:

“In this work, we build maps based on Phasors (*Phase Vectors*). The Phasor approach has multiple advantages deriving from its properties²²⁻²⁵. After transforming the spectrum at each pixel into its Fourier components, the resulting complex value is represented as a 2-dimensional histogram where the axes represent the real and imaginary components. Such histogram has the advantage of providing a representative display of the statistics and distributions of pixels in the image from a spectral perspective, simplifying identification of independent fluorophores. Pixels in the image with similar spectra generate a cluster on the phasor plot. While this representation is cumulative across the entire image, each single point on the phasor plot is easily remapped to the original fluorescent image²⁶⁻²⁹. “

As is mentioned here, references 22-25 guide the reader through a large variety of literature (limited by the number of citations in the journal) with regards to the properties of the phasor.

The components are binned in a 2d histogram. Bins in the histogram contain spectra with similar shape. This binning partially reduces the noisy aspect of the data, as smaller differences, indistinguishable due to uncertainties in microscope measurements, are associated in the same bin. This 2d-histogram also provides a graphical interpretation of all the spectra in the dataset, at a glance.

This graphical interpretation has helped us characterize and further reduce the noise from a spectral dimension approach, without decreasing the spatial information. This aspect is described in the text, right below the above quote:

“These clusters have been characterized and exploited for simplifying interpretation and spatially lossless denoising of data, improving both collection and analysis in low-signal conditions²⁹. “

Reference 29 is our previous work in Nature Methods and we believe it does not need to be demonstrated again in this work. This aspect of denoising was already underlined in our previous correspondence (Response_to_referees_Letters_20181211.pdf, page 70-71), from which we quote:

“An in-depth description of the denoising approach for the phasor method with details on the poissonian noise and scatter shape was presented in our previous work (Cutrale, Trivedi, Nature Methods 2017) and was the key to accessing longitudinal imaging of multispectral fluorescent samples with reduced laser power and photo-bleaching. In Figure 1 of our previous work (<https://www.nature.com/articles/nmeth.4134>), we show that the phasor denoising approach reduces the scatter within a cluster and reduces the poissonian noise. This was very clearly quantified in our previous work Supplementary Figures 1-5 and extensively described in Supplementary Notes 1-5.”

We understand the reviewer might not have access to this article, so we report some of the Supplementary Figures of (Cutrale, Trivedi, Nature Methods 2017) here below:

Supplementary Figure 2

Errors on hyper-spectral phasor plot.

(a) *scatter error* scales inversely as the square root of the total *digital counts*. Scatter error also depends on the Poissonian noise in the recording. R-squared statistical method is used to confirm linearity with the reciprocal of square root of counts. The slope is a function of the detector gain used in acquisition showing the counts-to-scatter error dynamic range is inversely proportional to the gain. Lower gains produce smaller scatter error at lower intensity values. The legend is applicable to all parts of the figure. (b) Denoising in the phasor space reduces the scatter error without affecting the location of expected values ($z_e(n)$) on the phasor plot. (c) Denoised scatter error linearly depends on the scatter error without filtering, irrespective of the acquisition parameters. The slope is determined by the filter size (3x3 here). (d) Denoising does not affect normalized *shifted-mean* errors since the locations of $z_e(n)$'s on the phasor plot remain unaltered due to filtering (d). In this case one filtering was applied.

Supplementary Figure 3

Sensitivity of phasor point.

(a,b,c) $|z(n)|$ remains nearly constant for different imaging parameters. Legend applies to (a,b,c,d,e). (d) Total digital counts as a function of laser power. (e) Proportionality constant in eq. 2 depends on the gain. (f) Relative magnitudes of residuals ($R(n)$) on phasor plots at harmonics 1 to 16, shows that harmonics $n=1$ and 2 are sufficient for unique representation of spectral signals.

Reprinted by permission from Springer Nature Customer Service Centre GmbH: Springer Nature, Cutrale, F., Trivedi, V., Trinh, L. et al. Hyperspectral phasor analysis enables multiplexed 5D in vivo imaging. *Nat Methods* **14**, 149–152 (2017) doi:10.1038/nmeth.4134

Supplementary Figure 4

Effect of phasor space denoising on scatter error and shifted-mean error.

(a) Scatter Error as a function of digital counts for different numbers of denoising filters with 3 by 3 mask. Data origin is fluorescein dataset acquired at gain 800 (Supp. Table 2). (b) Scatter Error as a function of number of denoising filters with 3 by 3 mask for different laser powers. (c) Shifted-Mean Error as a function of digital counts for different number of denoising filters with 3 by 3 mask. Data origin is fluorescein dataset acquired at gain 800. (d) Shifted-Mean Error as a function of number of filters with 3 by 3 mask for different laser powers. (e) Relative change of Scatter Error as a function of number of denoising filters applied for different mask sizes. (f) Relative change of Shifted-Mean Error as a function of number of filters applied for different mask sizes.

[REDACTED]

The authors use the term accuracy, but it is not clear how the metric is related to ‘accuracy’.
Indeed the metric appears to be a measure of color contrast rather than accuracy?

We apologize for any confusion the reviewer may have had about our use of the term accuracy.
We have changed the title of the Accuracy calculation paragraph in our Methods section to be :

“Spectral separation accuracy calculation”

We also improved the detail in that paragraph, which was outlining our equations for determining the accuracy. In summary,

1. we simulate three concentric squares, as shown in Supplementary Figure 3. Each concentric square has a unique spectrum.
2. we run ICA, obtain 3 independent components. Normalize those components to 1, use them as R, G, B colors.
3. Notice: the perfect result should show three perfectly separate squares. Which would result in three separate colors red (1,0,0), green (0,1,0) and blue (0,0,1).
4. we consider each square color as a vector and calculate the distance between square 1, 2 and 3.
5. we defined accuracy as the percent ratio of the contrast determined by each implementation (ICA or SEER) to the maximum of a perfect separation of the three squares (perfect R, G and B).

While we believe this is a correct usage of the term accuracy, we also understand that this may cause some confusion, and so, we have further defined the term accuracy by replacing it with “accuracy of spectral separation”. We also edited equation 4 which defines *Sp.sep.acc* (Spectral separation accuracy) to improve clarity and added one more equation (eq. 5) to better explain our definition.

The paragraph now reads:

Spectral separation accuracy calculation

We utilize the Simulated Hyperspectral Test Chart to produce different levels of spectral overlap and signal to noise ratio (SNR). We utilize multiple RGB visualization approaches for producing compressed RGB images (Supplementary Figure 20, Supplementary Figure 21). Each panel of the simulation is constructed by three different spectra, organized as three concentric squares Q_1 , Q_2 , Q_3 (Supplementary Figure 3). The maximal contrast visualization is expected to have three well separated colors, in this case red, green and blue. For quantifying this difference, we consider each (R,G,B) vector, with colors normalized [0,1], in each pixel as a set of Euclidean coordinates (x,y,z) and for each pixel calculate the Euclidean distance:

$$l_{12} = \sqrt{\sum_{i=R}^B (p_{Q1} - p_{Q2})_i^2} \quad (\text{eq. 3})$$

where l_{12} is the color distance between square Q_1 and Q_2 . p_{Q1} and p_{Q2} are the (R,G,B) vectors in the pixels considered, i is the color coordinate R, G or B. The color distances l_{13} and l_{23} between squares Q_1Q_3 and Q_2Q_3 respectively, are calculated similarly. The accuracy of spectral separation (Supplementary Figure 22) is calculated as:

$$Sp. sep. acc = \frac{(l_{12} + l_{13} + l_{23})}{l_{red-green} + l_{red-blue} + l_{green-blue}} \quad (\text{eq. 4})$$

where the denominator is the maximum color distance which can be achieved in this simulation, where Q_1 , Q_2 and Q_3 are respectively pure red, green and blue, therefore:

$$l_{red-green} + l_{red-blue} + l_{green-blue} = 3\sqrt{2} \quad (\text{eq. 5})$$

“

We changed every reference to “accuracy” in the text to be “spectral separation accuracy”

Also the results of the ICA comparison are not as I would expect: (1) my understanding is that higher values of ‘accuracy’ would correspond to greater differences in color contrast, but that is not apparent from the curves: for example for the two left graphs of supplementary Fig 23, the highest ‘accuracy’ occurs at highest SNR according to the graphs – as is expected – but the ICA images actually seem to have the lowest contrast for the highest SNR.

- (1) As explained extensively in our reply above, the ability of ICA to provide the most optimal basis relies on the assumption that the signal to noise ratio is very high and that the noise is additive. As stated in the caption and methods, the simulation images used in our analysis have been generated with real noise acquired from data captured by a confocal microscope.

Both the definition of spectral separation accuracy and how it is calculated, are shown in an improved section “Spectral separation accuracy calculation” in the Methods. From the definition we used (**equation 4**, Methods), we utilize the normalized values of the independent components as a measure for how good the separation between the 3 concentric squares is.

Regarding images at highest SNR in Supp. Fig. 23 panel *a*, to avoid any further discussion on ICA implementation (which is not the focus of this paper), we have utilized ENVI to perform the ICA calculations, as suggested by this reviewer.

ICA Calculations:

The files utilized performing calculations are multi-layer tiffs containing 100x100x32 (x,y,channel) datasets and can be downloaded from our website (<http://bioimaging.usc.edu/software.html>):

- [LINK] Multi-Layer Tiff files for simulation/ICA

The files were converted to ENVI using the Spectral library for python (<http://www.spectralpython.net/>) function `envi.save_image`.

- [LINK] all python scripts for converting Tiff-ENVI/ENVI-Tiff

The results were .hdr ENVI files which for simplicity we also link here:

- [LINK] ENVI files for simulation/ICA

We loaded the files in ENVI and utilized Forward ICA Rotation New Statistics:

We increased the number of iterations to allow more chances for ICA to converge, changed contrast function to Kurtosis (as it gave better results, see image below) and changed the number of output IC bands to 3 (as we know there are only 3 bands in this simulation). We then Run the ICA and ENVI visualizes a result. We show here the highest SNR in Supp. Fig. 23 panel *a*.

For each simulation (overlap/SNR) we saved the ENVI results. All ICA ENVI files are available here:

- [LINK] ENVI files results (/ICA/ folder)

We then wrote another python script to open the ENVI ICA files and convert them to normalized tiff images. We read the ENVI files using SpectralPython function `envi.open()` .

The ICA results contain negative values, so for each independent component, we subtracted the minimum value of the ICA result if this was negative. We made sure the data type was float64 to avoid memory overflow for the high SNR simulations, which was the reason for the confusing previous Supp. Fig. 23.

This is the portion of the code that performs this value shift and the normalization to 2^{16} .

```
====Python Code====
if ICA_results.min()<0:

    for chan_v in range(0,3):

        ICA_results[:, :, chan_v] = ICA_results[:, :, chan_v] - ICA_results[:, :, chan_v].min()

=====
```

These results for both ICA-Kurtosis and ICA-LogCosh were then written as uint16 in Tiff files. These Tiff files can be downloaded here:

- [LINK] ENVI files results (/ICA_Tif/ folder)

Python scripts are in the same link above.

We checked if there was any way to improve this ICA performance and ran a comparison of scikit.learn FastICA function with different parameters with the results of ENVI. We utilized LogCosh as function and improved the tolerance in hope that results would improve. The settings we utilized can be seen in the Python files linked above and in the following line:

====Python Code====

```
ica = FastICA(n_components=nc, algorithm='deflation', whiten=True, fun='logcosh',
             fun_args=None, max_iter=5000, tol=0.001, w_init=None, random_state=None)
```

=====

The figure below shows a summary of the performance of Envi ICA (LogCosh and Kurtosis) and 6 different ICA fine tunings in python scikit-learn:

The same simulation used in Supplementary Figure 20, which changes parameters for the Simulated Hyperspectral Test Chart obtaining different values of peak-to-peak spectral overlap and signal to noise, is used here to compute the spectral separation accuracy for Independent Component Analysis using ENVI (ICA-Kurtosis and ICA-LogCosh) and Scikit-learn's FastICA function (LC-tol= 10^{-3} to LC-tol= 10^{-8}), with 3 independent components (ICs) with optimization for this specific dataset. (a,b,c,d,e). The three ICs are utilized as R, G, B channels for creating a color image for each simulation parameter (ICA = 3 line) and are shown here next to SEER harmonic 1 and 2 (SEER h=1 and SEER h=2 respectively). Error bars are the standard deviation. (f,g,h,i,j) The parameters of spectral separation accuracy described in the methods section are applied here to the SEER and ICA results. From this calculation the values of spectral separation accuracy for ICA are all within range of each other and in accordance.

In an effort to avoid any further misunderstanding and maximize transparency and repeatability we will list a step-by-step instruction on how to obtain the same results using SEER. The

software has been available to the reviewer and the entire community from the first submission on our website (<http://bioimaging.usc.edu/software.html#HySP>) .

For simplicity:

Step 1: Download and install HySP (latest version is always on our webpage)

MacOS version

Win64 version).

MacOS: open .dmg and drag-drop the app to the application folder. (.app should be in Applications)

Win64: double click .msi installer and follow instructions (executable should be in Program Files/HySP)

Step 2: Download simulation datasets [LINK] and extract zip file. All other datasets in this paper are available on our webpage (<http://bioimaging.usc.edu/software.html#sampledatasets>)

Step 3: Open the HySP application and click on “Show Image”.

Navigate to the folder with the datasets and select one of the tiff files (these are multilayer tiffs with 32 channels). A prompt should appear:

this is the importer for tiff files and should list a good guess of the data shape. In this case it's a single multilayer tiff file with 32 channels inside so it is correct. Click OK and a new Image window showing the average grayscale image should appear.

Step 4: Calculate Phasor. Click on the “Calculate Spectral Phasor” button which is now active in the main initial window. A new window with the Phasor plot will appear.

Select the multiplier next to “Filter Data” and set it to 5x. Click on “Filter Data”.

Step 5: Visualize SEER. Return to the Image window and on the bottom left click on the dropdown menu labeled “Grayscale” and the different SEER options of Gradient ascent, descent, radius, angle, and contour will appear. Select “Gradient descent (hsv)”.

The “Mode” dropdown menu is right below the SEER map menu. The initial value is set as “Original Maps”. Click on it and select “Scaled”. The image will update.

Selecting Morphed Maps in the “Mode” dropdown menu will activate the different Morph modes in the drop down right below.

Selecting Mass Morph will yield the result below:

We have updated the supplementary figure 23 which now looks like this:

Figure caption reads:

“Supplementary Figure 23. Comparison of SEER and ICA spectral image visualization (RGB) under different spectral overlap and SNR conditions. The same simulation used in Supplementary Figure 20, which changes parameters for the Simulated Hyperspectral Test Chart obtaining different values of peak-to-peak spectral overlap and signal to noise, is used here to compute the **spectral separation** accuracy (**Methods**) for Independent Component Analysis using **ENVI**, with 3 independent components (ICs) **with** optimization for this specific dataset. **(a,b,c,d,e)**. The three ICs are utilized as R, G, B channels for creating a color image for each simulation parameter (ICA = 3 line) and are shown here next to SEER harmonic 1 and 2 (SEER h=1 and SEER h=2 respectively). Error bars are the standard deviation. **(f,g,h,i,j)** The parameters of **spectral separation** accuracy described in the methods section are applied here to the SEER and ICA results. Each value in the plots represents the average distance of 200^2 pixels and error bars are the standard deviation of normalized **spectral separation** accuracy value across all pixels. Overall **spectral separation** accuracy of ICA, as calculated here, averages at 24.8% for Kurtosis function (ICA-K), 24.7% for LogCosh (ICA-L), while SEER’s 55.7% h=1 and 57.5% h=2. For the different levels of overlaps the average spectral separation accuracy are **(f)** ICA-K 15.8%, ICA-L 15.9%, SEER 38.0% for harmonic 1, 50.6% for harmonic 2; **(g)** ICA-K 29.4%, ICA-L 29.2%, SEER h=1 57.0%, SEER h=2 49.6%; **(h)** ICA-K 20.1%, ICA-L 19.2%, SEER h=1 57.2%, SEER h=2 60.0%; **(i)** ICA-K 30.3%, ICA-L 30.1%, SEER h=1 59.9%, SEER h=2 60.4%; **(j)** ICA-K 28.6%, ICA-L 28.8%, SEER h=1 66.3%, SEER h=2 66.7%;

“

(2) More importantly, the results for ICA are not what I expect from ICA – I would expect that each of the different spectra to approximate pure R, G or B as for SEER – however they all look greyish suggesting little discrimination. This suggests that ICA may be being achieved. Also the hue varies systematically with SNR, which is also unexpected. Overall, I believe that either these results suggest an incorrect implementation of ICA or they require careful explanation – at least to convince this reviewer of their validity.

- (2) The reviewer is correct on both points:
- (a) the ICA R,G or B values look grays because ICA returns values with little discrimination,
 - (b) Using two different approaches for ICA we obtain almost identical performance. We believe ICA is here achieved (as we demonstrated above using multiple ICA implementations). The lower performance of ICA in this case is in line with low signal to noise data and, as we explain above, relates to a different underlying model of ICA which does not properly apply in these low intensity, poissonian noised fluorescence signal.

In consideration of all these components of noise, modeling, SNR and the comments of the reviewer her/himself, we believe that the ICA implementation is now correct.

To better explain to this reviewer the process which leads to an overall expected low performance of ICA in noisy environment, we have restated our previous answers (above) to this reviewer:

- a. We are working with generally low Signal to noise fluorescent data. The SNR conditions used in this simulation (and generally encountered in this data) were added as a figure in the previous review with **“Supplementary Figure 21. Spectra of extreme conditions in SNR-Overlap simulation”**

- b. The solution we propose, SEER, accomplishes visualization of large data through the Phasor approach combined with spatially lossless spectral denoising of data.
- c. shown that PCA/ICA are known to underperform in low SNR conditions, quoting a number of articles which mention words of caution on using ICA in noisy environments.
- d. described in depth the ICA basic model for noise which is physically different from the noise in fluorescence data.
- e. Shown that the solution we propose, SEER, accomplishes visualization of large data through Phasor approach combined with spatially lossless spectral denoising of data (with references of where all these points are explained in the manuscript) and that Phasor is not simply “a decomposition into Fourier components”.

We believe the answer to this question is described in detail in our previous answers to this reviewer.

On the issue of speed, it is unfortunate that the authors seem to find some additional challenges in using ENVI for microscopy data – this is not a problem that we have experienced and indeed ENVI is also recommended for HSI microscopy by the distributors.

This is quite surprising. We certainly believe the reviewer in having had no problems with speed in their implementation of fluorescence microscopy spectral imaging with ENVI, but, as we showed in the video recording in our first rebuttal this was not the case for us.

Regarding ENVI being recommended for HSI fluorescence microscopy by the distributors, in our experience, this is not the case. The reasons for this are multiple:

1. Our past and present expertise in multispectral fluorescence imaging. The spectral fluorescence confocal imager (US Patents 6,403,332 and 20060238756A1) were developed by Scott E. Fraser (co-author) and are currently licensed to Zeiss and commercialized in all confocal units from the LSM510, 710, 880.

[REDACTED]

US Patent pending 62/593,079 and 7636101-18-0051 are both deriving from our work and currently under licensing discussion.

2.

[REDACTED]

The major reason being it is designed for satellite imaging, not fluorescence.

3. Widespread online microscopy resources like: Zeiss Campus (<http://zeiss-campus.magnet.fsu.edu/articles/spectralimaging/index.html>), Nikon MicroscopyU (<https://www.microscopyu.com/techniques/confocal/spectral-imaging-and-linear-unmixing>) do not mention ENVI to the best our knowledge.

4. There are four publications in total that we could find out of the several hundred on multispectral fluorescence, which utilize ENVI in fluorescence imaging. Three publications belong to the same group in Alabama, authored by Silas J. Leavesley . [1,2,3]

[1] Peter F. Favreau, Clarissa Hernandez, Ashley Stringfellow Lindsey, Diego F. Alvarez, Thomas C. Rich, Prashant Prabhat, Silas J. Leavesley, "Thin-film tunable filters for hyperspectral fluorescence microscopy," *J. Biomed. Opt.* 19(1) 011017 (26 September 2013).

[2] Silas J. Leavesley, Mikayla Walters, Carmen Lopez, Thomas Baker, Peter F. Favreau, Thomas C. Rich, Paul F. Rider, Carole W. Boudreaux, "Hyperspectral imaging fluorescence excitation scanning for colon cancer detection," *J. Biomed. Opt.* 21(10) 104003 (28 October 2016).

[3] Leavesley, S. J., Annamdevula, N., Boni, J., Stocker, S., Grant, K., Troyanovsky, B., Rich, T. C., ... Alvarez, D. F. (2011). Hyperspectral imaging microscopy for identification and quantitative analysis of fluorescently-labeled cells in highly autofluorescent tissue. *Journal of biophotonics*, 5(1), 67-84.

[4] Harris, A. T. (2006), Spectral mapping tools from the earth sciences applied to spectral microscopy data. *Cytometry*, 69A: 872-879. doi:[10.1002/cyto.a.20309](https://doi.org/10.1002/cyto.a.20309)

Given these motivations, we are quite confused about this comment from the reviewer.

Nevertheless, the comparison of SEER speed with fastICA should be sufficient, but it should be made more clear in the text that in both cases the algorithms are implemented in Python (an interpreted language and hence potentially one to two orders of magnitude slower than a compiled implementation in eg ENVI) and hence that the speed comparison is valid.

We agree with the reviewer. Implementation using a compiled language would speed up the computation 100x to 400x. We have edited the text to include this important point:

“Comparing with the python module, scikit-learn’s implementation of fast Independent Component Analysis, SEER provides up to a 67-fold speed increase (Supplementary Figure 1) and lower virtual memory usage. These results were obtained using Python, an interpreted language. Implementation of SEER with a compiled language could potentially increase speed by one order of magnitude. “

Overall, I think that SEER is an interesting approach to visualisation, but the comparison with a gold standard of ICA does not appear to me to be valid.

We believe we have fully addressed the concerns of this reviewer in the above questions with numerous demonstrations, descriptions, and literature quotations. We have also edited the text to reflect the changes suggested and to further improve the clarity of this work.

Minor issues

In the caption to Supp Fig 20 – the values ‘ $(.5*10)^1 - (.5*10)^4$ ’ seem to be incorrectly written.

We have corrected the caption, which now reads:

“
is reduced in intensity by a factor of $5*10^0 - 5*10^4$ (panel 1-5 respectively)
”

Similarly, we have corrected Supplementary Figure 5:

“is reduced in intensity by a factor of $10^1 - 10^4$ (panel 1-4 respectively) ”

Reviewers' comments:

Reviewer #2 (Remarks to the Author):

I would like to thank the authors for addressing all the points I raised. I think that now the purpose of the SEER approach is better highlighted and I suggest the paper for publication.

Reviewer #3 (Remarks to the Author):

Third review of Visualization of hyperspectral fluorescent data with Spectrally Encoded Enhanced Representations (SEER)

Authors: Wen Shi et al

The authors have provided a rather long defence. Although the answers are long and detailed and argue that ICA may not be optimal for low-SNR images, there is no response to my comment

"...This merits some discussion and explanation of why a decomposition into Fourier components, (which are not in general an accurate match to actual spectra), produces a higher accuracy than a decomposition into an optimised basis set."

That is: because ICA may not be optimal, as argued by the authors (and accepted by this reviewer), it does not mean that SEER is better. Sceptical readers (as all readers should be) would be justified in seeking the comfort of an explanation of why such a simple representation as the phasor of a single Fourier component would work so much better than ICA in general. This manuscript still does not provide that comfort.

The authors mount a prolonged argument against my description of "a decomposition into Fourier components" in reference to the phasor representation used in SEER. I'm not sure why: the output of the Fourier-transform used in SEER is an array of complex frequencies, of which each component, including the subset use by SEER can be depicted as a phasor. It is strange to offer refs 22-25 to describe the properties of a phasor when it can be found in a text book on elementary maths – indeed the authors state in their Nature Methods paper "SP uses the Fourier transform to depict the spectrum of every pixel in an image as a point on the phasor plane (Fig. 1a), providing a density plot of the ensemble of pixels." The rationale for the lengthy rebuttal on this simple point is not clear to me. It is a natural question for a reviewer and reader to ask why such a simple representation should work better than more complex analyses.

The authors suggest that I am somehow confused by the term 'accuracy' and suggest that 'spectral separation accuracy' is a more suitable phrase. Accuracy refers to closeness to the correct value – I believe that in these data produced in the paper, there is no ground truth from which the correct value is known and so accuracy is not known. I refer the authors to my previous comment, that the measure seems to refer instead to contrast.

In response to my comment:

"Also the results of the ICA comparison are not as I would expect: (1) my understanding is that higher values of 'accuracy' would correspond to greater differences in color contrast, but that is not apparent from the curves: for example for the two left graphs of supplementary Fig 23, the highest 'accuracy' occurs at highest SNR according to the graphs – as is expected – but the ICA images actually seem to have the lowest contrast for the highest SNR"

the rebuttal seemed to provide a long explanation of how ICA was calculated but I could not find a straightforward explanation of why there is lowest contrast at high SNR – contrary to expectations and the comments by the authors that ICA is suboptimal only for low SNR.

On my comment

".(2) More importantly, the results for ICA are not what I expect from ICA – I would expect that

each of the different spectra to approximate pure R, G or B as for SEER – however they all look greyish suggesting little discrimination. This suggests that ICA may [not*] be being achieved. Also the hue varies systematically with SNR, which is also unexpected. Overall, I believe that either these results suggest an incorrect implementation of ICA or they require careful explanation – at least to convince this reviewer of their validity.”

the main argument used in the rebuttal for ICA being achieved is that two implementations gave similar responses. Another explanation is that for some reason the ICA is incorrectly implemented in both cases. I could not see an explanation for each of the factors above that one would expect to see in a correct implementation – that would be more compelling than a description of the mechanics of implementation.

*My apologies I accidentally omitted the all-important “not” in my original 2nd review

I find the long discussion about the usability of ENVI for microscopy data to be irrelevant to the acceptability of the paper for publication.

Reviewers' comments:

Reviewer #2 (Remarks to the Author):

I would like to thank the authors for addressing all the points I raised. I think that now the purpose of the SEER approach is better highlighted and I suggest the paper for publication.

Reviewer #3 (Remarks to the Author):

Third review of Visualization of hyperspectral fluorescent data with Spectrally Encoded Enhanced Representations (SEER)

Authors: Wen Shi et al

The authors have provided a rather long defence. Although the answers are long and detailed and argue that ICA may not be optimal for low-SNR images, there is no response to my comment

“...This merits some discussion and explanation of why a decomposition into Fourier components, (which are not in general an accurate match to actual spectra), produces a higher accuracy than a decomposition into an optimised basis set.”

That is: because ICA may not be optimal, as argued by the authors (and accepted by this reviewer), it does not mean that SEER is better. Sceptical readers (as all readers should be) would be justified in seeking the comfort of an explanation of why such a simple representation as the phasor of a single Fourier component would work so much better than ICA in general. This manuscript still does not provide that comfort.

We thank the reviewer for the comment. We would like to further clarify and answer the points in the above paragraph as follows.

Regarding *“there is no response to my comment*

“...This merits some discussion and explanation of why a decomposition into Fourier components, (which are not in general an accurate match to actual spectra), produces a higher accuracy than a decomposition into an optimised basis set.” [reviewer #3 above]

Our previous rebuttal presents a thorough discussion to this question:

- (page 12-22 Second_Response_to_Referees_Letters_2019-03-30),

with edits in the text, references of relevant parts of the manuscript answering this point and supporting data from other publications, namely:

1. We are working with generally low signal to noise fluorescence data.
2. The solution we propose, SEER, accomplishes visualization of large data through the Phasor approach combined with spatially lossless spectral denoising of data.
3. PCA/ICA are known to underperform in low SNR conditions.
4. Noise in fluorescence data is not only additive but also poissonian in nature.
5. Phasor is not simply “*a decomposition into Fourier components*”

Particularly, the last point (5) describes how Phasor introduces denoising in the data, with reference to all the paragraphs in the manuscript describing the denoising aspect and improved performance at lower SNR, also topic of our previous work [ref 29].

On the “Sceptical readers (as all readers should be) would be justified in seeking the comfort of an explanation of why such a simple representation as the phasor of a single Fourier component would work so much better than ICA in general. This manuscript still does not provide that comfort.” [reviewer #3 above]

As described in point 5, SEER, the visualization approach presented in this manuscript, is not simply “a decomposition into Fourier components”. SEER consists in complex maps applied on a denoised Phasor plot, which is a histogram representation of the Fourier components.

SEER has 3 processing steps consequent to the calculation of the “Fourier components”, namely:

1. Histogram representation (the phasor plot): which provides insight on the spectral population distribution and improvement in the signal through summation of spectra in the histogram bins. This property of the phasor plot was explained in the original work of the Gratton group (ref 23) and in that of the Gerritsen group (ref 26).
2. Denoising: in our previous work we presented a spatially lossless approach for denoising spectral fluorescent data which decrease spectral error up to ~40% at low signal to noise (ref 29, supplementary figure 4, also reported in point 5 “Phasor is not simply “*a decomposition into Fourier components*” of our previous answer to this same question for this reviewer)
3. Complex Spectrally Encoded Enhanced Representation, Reference Maps, with multiple modes which adapt to the statistics/distribution of spectra on the phasor plot (the histogram).

We believe this has been extensively explained to this reviewer in the previous rebuttal as well as in the text

- (page 18-22 Second_Response_to_Referees_Letters_2019-03-30).

We also show the SEER maps are quite far from a “*simple representation*”. Figures 1, 2 and 3 very explicitly show that the technique here presented is not just a “*single Fourier component*”.

With regards to SEER vs ICA:

SEER includes denoising. ICA does not.

SEER has maps with multiple enhanced representations (see Figure 2 for reference), while ICA has a single output.

SEER includes statistical weighting with 4 different modes (Figure 3) which “can provide improved contrast during visualization” while ICA provides a 1-step solution.

Manuscript references were reported in the previous rebuttal in response to this reviewer’s question in paragraph 5 “Phasor is not simply “*a decomposition into Fourier components*””.

It is unclear whether the reviewer is skeptical about the SEER visualization approach presented in this manuscript or the Phasor approach to fluorescence microscopy in general. The former has been addressed extensively above and in previous replies to the reviewer.

If the skepticism is toward the Phasor approach to fluorescence microscopy, there is a large body of literature that points to this as a well-established approach and in specific cases, outperforms conventional ICA even without the phasor-denoising which we showed in our work (Ref.29).

Supporting literature on the underperformance of ICA/PCA compared to phasor:

1. Phasor analysis of multiphoton spectral images distinguishes autofluorescence components of in vivo human skin. Fereidouni, Bader, Colonna, Gerritsen. (Biophotonics 7, No. 8, 589–596 (2014) / DOI 10.1002/jbio.201200244)

In the concluding paragraph:

“ Conventional spectral analyses methods like linear unmixing and principal component analyses methods have been used to analyze conventional spectral imaging data from fluorescently labeled specimens. These techniques are, however, less suitable for analyzing AF and SHG images. In the case of autofluorescence, the number of components is large and often the spectral differences are only minor.”

2. Phasor analysis for nonlinear pump-probe microscopy. Robles, Wilson, Fischer, Warren (Optics Express Vol. 20, Issue 15, pp. 17082-17092 (2012) <https://doi.org/10.1364/OE.20.017082>) .

In the results section “*Advantages of using this [phasor] approach compared to PCA and multi-exponential fitting for these applications are discussed.*”

The work has a long list of examples where PCA fails in separating closely related signals, where phasors succeed.

“For example, if the data are analyzed using orthogonal projections, such as in PCA, some contributions from Hb will typically be projected onto eumelanin, and similarly some contributions from surgical ink will be projected onto pheomelanin. This results from the fact that only small portions of the signals are actually orthogonal to one another. In contrast, phasor analysis utilizes the whole trace to identify its position in a two-dimensional space defined by g and s , and does not require the signals to be orthogonal.”

“Secondly, surgical ink produces phasors that are much more scattered, which indicates that it does not have a consistent transient absorption spectrum, further emphasizing the difficulty in eliminating these contributions using PCA (or any projection based method).”

“This can also be done using PCA, however, this tends to omit more data owing to the fact that these signals are not inherently orthogonal, and thus manual segmentation of the surgical ink is usually preferred over PCA.”

“Interestingly, the cumulative, intensity weighted histogram phasor plot of five gray-blue lapis lazuli images (Fig. 8(c)) not only shows the presence of lapis, but also suggests the presence of at least three distinct impurities (highlighted by numbers 1-3). This type of analysis is difficult (if not impossible) to achieve with PCA or multi-exponential fitting and provides a promising tool for future work in this application.”

The authors mount a prolonged argument against my description of “a decomposition into Fourier components” in reference to the phasor representation used in SEER. I’m not sure why: the output of the Fourier-transform used in SEER is an array of complex frequencies, of which each component, including the subset use by SEER can be depicted as a phasor.

It is evident that we have not provided a clear enough explanation of our method.

The “*output of the Fourier-transform*” is not directly “*used in SEER*”. We are not mapping to color directly the Fourier-transform at one harmonic and the phasor plot is obtained before SEER is applied.

The steps that results in the SEER visualization comprise (in this order):

1. Sine/Cosine Fourier Transform: provides the Fourier components at one harmonic (usually 1st or 2nd owing to Riemann surfaces)
2. Histogram representation (the phasor plot): which provides insight on the spectral population distribution and improvement in the signal through summation of spectra in

the histogram bins. This property of the phasor plot was explained in the original work of Gratton group (ref 23) and in that of Gerritsen group (ref 26).

3. Denoising: in our previous work we presented a spatially lossless approach for denoising spectral fluorescent data which decrease spectral error up to ~40% at low signal to noise (ref 29, supplementary figure 4, also reported in point 5 “Phasor is not simply “a decomposition into Fourier components” of our previous answer to this same question for this reviewer)
4. Complex Spectrally Encoded Enhanced Representation, Reference Maps, with multiple modes which adapt to the statistics/distribution of spectra on the phasor plot (the histogram).

We could not find any part of the manuscript where we state that SEER is applied directly after the Sine/Cosine Fourier transform. The process is depicted in Figure 1 of the manuscript.

However, it is evident that this crucial steps are not sufficiently clear in the manuscript so we made the following further edits (in addition to the edits made in previous rebuttal for this same questions from this reviewer):

In the introduction:

“The solution we propose extracts from both the whole **denoised** phasor **plot** and image to reconstruct a “one shot” view of the data and its intrinsic spectral information “

In Figure 1 caption:

“Spectra for each voxel within the dataset are represented as a two-dimensional histogram of their Fourier coefficients S and G, known as the phasor plot **and denoised**. SEER provides a choice of several color reference maps that encode positions on the phasor into predetermined color palettes. ”

Multiple aspects of the answers to this question were already given to this reviewer in :

- Pages 58-60 - Response_to_Referees_Letters_2018-12-11
- Pages 70-71 - Response_to_Referees_Letters_2018-12-11
- Pages 18-22 - Point 5 - Second_Response_to_Referees_Letters_2019-03-30

It is strange to offer refs 22-25 to describe the properties of a phasor when it can be found in a text book on elementary maths

This point was answered in our previous answer to this reviewer, page 18-22 starting from “As is mentioned here, references 22-25 guide the reader through a large variety of literature (limited by the number of citations in the journal) with regards to the properties of the phasor.”
(Second_response_to_Referees_Letters_2019-03-30_FC)

This statement either:

1. Suggests we did not clarify the approach we are utilizing or
2. might appear to be unintentionally diminishing the past decade of work of multiple groups.

As we expect to publish this useful commentary, to avoid any misunderstanding for any future reader, we will address both cases.

1. We have revised the introductory paragraph to clarify that we are using the Phasor approach to Fluorescence Microscopy, as opposed to the more commonly used Phasor for electrical circuitry (described below):

"In this work, we build maps based on Phasors (Phase Vectors). The Phasor approach to fluorescence microscopy has multiple advantages deriving from its properties for fluorescent signals²²⁻²⁵."

This comment is, however, fairly confusing as:

- a) this reviewer has used these same references in a previous comment to highlight the lack of novelty of this work, hence suggesting the work we present in this manuscript is not found in "*text book on elementary maths*". This point was answered thoroughly in our first response.

"[Reviewer 3] *The fundamental principles of SEER are described in ref 26 by Fereidouni, Bader and Gerritson of Utrecht (Optics Express, 2012) and its application to in vivo imaging of zebrafish was described in ref 27 (Nature Methods by the authors of this manuscript) in 2017.*"

- (Response_to_Referees_Letters_2018-12-11).

- b) The title already states "Pre-processing visualization of hyperspectral fluorescent data".

- c) The first words of the manuscript introduction are "Fluorescence Hyperspectral Imaging (fHSI)"

2. With over a hundred cumulative publications on the subject, we believe the phasor approach to fluorescence microscopy and its properties, as applied to a specific field (example: Fluorescence Lifetime, Spectral, Magnetic Resonance, non-linear pump probe, FCS) is not something that "*can be found in a text book on elementary maths*".

To bring proof of this point, we consulted a few of the "*text book[s] on elementary maths*", one in particular has a chapter about Phasors:

“Phasors are a means by which sinusoidal alternating voltages and currents can be specified in terms of a rotating radius and an angle, they behave like vectors. This chapter looks at how we can work with such quantities, considering both vector algebra and complex numbers.”

(Reference: Mathematics for Engineers and Technologists, IIE Core Textbooks Series 2002, Pages 57-84, Chapter: Vectors, phasors and complex numbers. Huw Fox; Bill Bolton.)

We believe it should be clear at this point of the review process and to any skeptical reader that we are not analyzing alternating voltages or currents.

The Phasors described in this textbook share name, polar plot and some mathematical properties with the approach we are using. There are a number of properties of phasors as applied to fluorescent signals that have been characterized by other groups and us (Ref 22-25), which are not referenced anywhere in these books:

- i) the basic representation of phasor as a histogram of points and how this histogram affects our understanding of large (10^6) fluorescent (lifetime, spectral etc) signals at a glance. The book above (and all of the others we consulted) does not show a single phasor plot of the likes in this manuscript. There also is no mention of how this histogram representation affects the encoding of fluorescent signals in either the spectral or lifetime domain. An in-detail discussion is already reported in:
 - Pages 13-22 - Point 2, 3 and 5 -
Second_Response_to_Referees_Letters_2019-03-30
- ii) the denoising properties of the phasor, a fundamental step of our previous work, cannot be found in these “*text book on elementary maths*”. This was the topic of our previous work and we have described answers to this point for this reviewer already in:
 - Pages 70-71 - Response_to_Referees_Letters_2018-12-11
 - Pages 14-15 - Point 2 - Second_Response_to_Referees_Letters_2019-03-30
- iii) the interpretation of vectorial logic for fluorescent signals: see figure 2.20 of the above referenced book.

Reprinted from Mathematics for Engineers and Technologists. A volume in IIE Core Textbooks Series. Fox & Bolton. *Vectors, phasors and complex numbers*, 57-84, <https://doi.org/10.1016/B978-075065544-6/50003-5>
 Copyright © 2002 Huw Fox and Bill Bolton. Published by Elsevier Ltd All rights reserved. (2002), with permission from Elsevier <http://www.elsevier.com>

Download full-size image

Figure 2.20. Adding phasors

This vectorial mathematics is appropriate for electrical signals, but how does it translate for fluorescent spectra? “Summing” fluorescent spectra results in a combination of the fluorescent signals, appearing on a point adjoining the two “endmembers”. These details are more specific to the application of phasors to fluorescence microscopy and, again, cannot be found in these books.

- iv) the separation of intrinsically noisy signals (in our case fluorescent signals): these books do not describe any aspects of large amount, highly multiplexed signal separation in noisy conditions.

– indeed the authors state in their Nature Methods paper “SP uses the Fourier transform to depict the spectrum of every pixel in an image as a point on the phasor plane (Fig. 1a), providing a density plot of the ensemble of pixels.” The rationale for the lengthy rebuttal on this simple point is not clear to me. It is a natural question for a reviewer and reader to ask why such a simple representation should work better than more complex analyses.

While the sentence is taken partially out of context, as the reviewer quotes “*SP uses the Fourier transform to depict the spectrum of every pixel in an image as a point on the phasor plane (Fig. 1a)*”, “a careful and skeptical reviewer would notice the words “*every pixel in an image*” and the “*providing a density plot of the ensemble of pixels*” where the histogram is described. This sentence shows that the method uses a Fourier decomposition but that this is only a single portion of the method itself. How does this representation affect fluorescent signals? We refer the reviewer to point (i) in the previous answer with references in the text.

On the partially out of context aspect of this statement: the sentence before the one reported by this reviewer states:

“*Conventional Spectral Phasor (SP)^{11,12,13} is an efficient processing and rendering tool for multispectral data.*”

(somehow not referenced here) indicates the origin of the approach (Spectral Phasors - for fluorescence microscopy) and not the conclusion of our previous work.

A careful reader would need to advance 2 sentences into the short Nature Methods abstract to discover there is more to phasors and fluorescence than just a Fourier Transform. The second sentence of the abstract states:

“Here, we report software called Hyper-Spectral Phasors (HySP) for denoising and unmixing multiple spectrally overlapping fluorophores in a low signal-to-noise regime with fast analysis.”

This sentence, less out-of-context, does give an insight into the conclusion of our previous work.

On the “*lengthy rebuttal*” comment, we believe it is our duty to provide a complete, clear answer for the reviewer (and readers, as these answers will be published). As our analysis is far from “*a simple representation*” the answers become more extensive. The discussion and explanation now described as a “*lengthy rebuttal*” was encouraged by this reviewer’s comment “*This merits some discussion and explanation of why a decomposition into Fourier components, (which are not in general an accurate match to actual spectra), produces a higher accuracy than a decomposition into an optimised basis set.*”

- (page 12 - Second_Response_to_Referees_Letters_2019-03-30).

We believe a skeptical reviewer should welcome thorough, elucidating responses rather than defining them as a “*lengthy rebuttal*”.

On the statement, “*why such a simple representation should work better than more complex analyses*”, we believe the point that SEER is far from “*a simple representation*” was answered for this reviewer in the following places:

- Above, Pages 4-6 - this document.
- Pages 59-60 - Response_to_Referees_Letters_2018-12-11
- Pages 69-71 - Response_to_Referees_Letters_2018-12-11
- Pages 12-22 - Second_Response_to_Referees_Letters_2019-03-30

The authors suggest that I am somehow confused by the term ‘accuracy’ and suggest that ‘spectral separation accuracy’ is a more suitable phrase. Accuracy refers to closeness to the correct value – I believe that in these data produced in the paper, there is no ground truth from which the correct value is known and so accuracy is not known. I refer the authors to my previous comment, that the measure seems to refer instead to contrast.

On the comment, “*there is no ground truth from which the correct value is known and so accuracy is not known,*” we respectfully disagree.

We simulated three concentric squares (Supplementary Figure 3) each with a unique spectrum. The spectra are reported in Supplementary Figure 3, and while closely overlapping, they are different because we have simulated them.

Example of three different spectra from Supplementary figure 3:

The question we try to answer here is “how well can SEER visually separate closely overlapping spectra” and this point was raised by this reviewer in the first round of review “*the main contribution of the paper seems to be about enhanced visualisation of SEER images rather the enhancement provided by SEER compared to traditional techniques*”

- (page 69 - Response_to_Referees_Letters_2018-12-11).

The ground truth is that the 3 spectra are different. Any method used for visually separating the 3 concentric squares spectra should ideally show three maximally distinct values, which we normalized to (R,G,B) vectors and show as colors. Datasets are reported in the “Data Availability” section at the end of the manuscript. A skeptical reader could download the simulated datasets and verify that the spectra (already represented in Supplementary Figure 3) are different.

The 3 independent components returned from the method used (SEER, ICA etc) are normalized and used as R, G, B components for visualization (the topic of this manuscript) and analysis of spectral separation accuracy (as defined in the manuscript and asked from this reviewer).

A perfect result should show three perfectly separate squares, which would result in three separate colors red (1,0,0), green (0,1,0), and blue (0,0,1). Again, this is the ground truth; the spectra are different, and the method used to distinguish them (SEER, ICA, etc) should show distinct (R,G,B) vectors.

The euclidean distance of the color vectors (R,G,B) represents a measurement of how well the three concentric squares are separated.

This aspect is described in the manuscript section titled “Spectral separation accuracy calculation” and equations 3, 4, 5 clearly show this point:

$$l_{12} = \sqrt{\sum_{i=R}^B (p_{Q1} - p_{Q2})_i^2} \quad (\text{eq. 3})$$

$$Sp. sep. acc = \frac{(l_{12}+l_{13}+l_{23})}{l_{red-green} + l_{red-blue} + l_{green-blue}} \quad (\text{eq. 4})$$

$$l_{red-green} + l_{red-blue} + l_{green-blue} = 3\sqrt{2} \quad (\text{eq. 5})$$

These equations address the point that “*Accuracy refers to closeness to the correct value.*” Particularly, “*the correct value*” is the denominator of eq.4 (reported as a numerical value in eq.5); the “*closeness*” is the ratio in eq. 4.

Extensive answers to this question and edits to the text were reported in:

- Pages 22-24 - Response_to_Referees_Letters_2018-12-11

In response to my comment:

“Also the results of the ICA comparison are not as I would expect: (1) my understanding is that higher values of ‘accuracy’ would correspond to greater differences in color contrast, but that is not apparent from the curves: for example for the two left graphs of supplementary Fig 23, the highest ‘accuracy’ occurs at highest SNR according to the graphs – as is expected – but the ICA images actually seem to have the lowest contrast for the highest SNR”

the rebuttal seemed to provide a long explanation of how ICA was calculated but I could not find a straightforward explanation of why there is lowest contrast at high SNR – contrary to expectations and the comments by the authors that ICA is suboptimal only for low SNR.

Figure 23 was corrected in:

- Pages 31-32 - Second_Response_to_Referees_Letters_2019-03-30

The “*straightforward explanation of why there is lowest contrast at high SNR*” is part of the “*long explanation*”:

- Pages 26 - Second_Response_to_Referees_Letters_2019-03-30

“The ICA results contain negative values, so for each independent component, we subtracted the minimum value of the ICA result if this was negative. We made sure the data

type was float64 to avoid memory overflow for the high SNR simulations, which was the reason for the confusing previous Supp. Fig. 23.”

Followed by the step-by-step demonstration of how the ICA was calculated, providing this reviewer (and any reader) all the data and functions used for calculations and the resulting updated figure 23:

On my comment

“(2) More importantly, the results for ICA are not what I expect from ICA – I would expect that each of the different spectra to approximate pure R, G or B as for SEER – however they all look greyish suggesting little discrimination. This suggests that ICA may [not*] be being achieved. Also the hue varies systematically with SNR, which is also unexpected. Overall, I believe that either these results suggest an incorrect implementation of ICA or they require careful explanation – at least to convince this reviewer of their validity.”

*My apologies I accidentally omitted the all-important “not” in my original 2nd review

the main argument used in the rebuttal for ICA being achieved is that two implementations gave similar responses. Another explanation is that for some reason the ICA is incorrectly implemented in both cases. I could not see an explanation for each of the factors above that one would expect to see in a correct implementation – that would be more compelling than a description of the mechanics of implementation.

We respectfully disagree. We have provided a thorough explanation of the reason for ICA’s lower performance as well as a step-by-step analysis with links to each of the intermediate results:

- Pages 24-32 - Second_Response_to_Referees_Letters_2019-03-30

On the “*for some reason the ICA is incorrectly implemented in both cases*” we used both publicly available (python scikit-learn) and commercial software (ENVI) with 10 different variations, establishing the consistency of our results.

- Page 27 - Second_Response_to_Referees_Letters_2019-03-30

With results in agreement between different implementations, the incorrect implementation would then be on the python scikit-learn package and the ENVI software side with both having incorrect implementations. Such a major error in code implementation would have been discovered by all the scientific groups and companies utilizing these functions, including this reviewer, who previously stated using “*ENVI for microscopy data – this is not a problem that we have experienced*”

- Page 33 - Second_Response_to_Referees_Letters_2019-03-30

This is also unlikely as, ENVI’s speed, simplicity and widespread use was suggested by this reviewer “*compared to a traditional technique (such as ICA), which may already be implemented quite easily and computed quickly (<1s) using standard software such as ENVI,*”

- Pages 52 - Response_to_Referees_Letters_2018-12-11

Particularly, the ENVI loading of data is shown in:

- Page 25 ‘ICA Calculations’ - Second_Response_to_Referees_Letters_2019-03-30

The parameters on ENVI leave very little room for user mistake, mainly the shape of data loading (i.e. accidentally loading x or y dimensions as wavelength). The correct loading is shown in the screenshots of the ENVI processing, in our previous rebuttal.

Both data and results are provided at

- Page 24 and 26 ‘ICA Calculations’ - Second_Response_to_Referees_Letters_2019-03-30

For the python scikit-learn analysis, we have linked the python scripts for review in our previous rebuttal:

- Page26 ‘second LINK’ - Second_Response_to_Referees_Letters_2019-03-30

Finally, on the “*I could not see an explanation for each of the factors above that one would expect to see in a correct implementation – that would be more compelling than a description of the mechanics of implementation*”, the explanation of the factors above have been summarized in the following parts of our previous response:

- Pages 12-22 - Second_Response_to_Referees_Letters_2019-03-30
- Pages 32-33 - Second_Response_to_Referees_Letters_2019-03-30

We believe over 12 pages of explanation should be sufficient to describe the underperformance of ICA in the simulations presented in this manuscript.

The “denoising” aspect of SEER, an important point in the underperformance of ICA in this work, is mentioned 24 times in `Second_Response_to_Referees_Letters_2019-03-30` and 6 times in the main manuscript with the corresponding references.

I find the long discussion about the usability of ENVI for microscopy data to be irrelevant to the acceptability of the paper for publication.

We completely agree with the reviewer. The discussion of the usability of ENVI for microscopy should be irrelevant for the acceptability of the paper for publication. Our answers on the topic arise from our belief that it is our duty to address the concerns of the reviewers.

This comment, however, is confusing as this reviewer in particular used ENVI as a comparison for discussing the acceptability of our paper for publication.

Reviewer 3: *“I believe that the manuscript as it stands is marginal on both counts, but could be sufficiently improved, particularly in terms of clarity and rigour to justify publication”*

Using as motivation:

“SEER may be considered as an alternative to established spectral decomposition techniques, such as PCA, (or better, ICA),[...] which may already be implemented quite easily and computed quickly (<1s) using standard software such as ENVI”

- Pages 51-52 - `Response_to_Referees_Letters_2018-12-11`

Reviewer 3: *“On the issue of speed, it is unfortunate that the authors seem to find some additional challenges in using ENVI for microscopy data – this is not a problem that we have experienced and indeed ENVI is also recommended for HSI microscopy by the distributors.”*

- Page 33 - `Second_Response_to_Referees_Letters_2019-03-30`

Reviewer 3: *“Another explanation is that for some reason the ICA is incorrectly implemented in both cases.”* [on Python scikit-learn and ENVI].

And:

“I find the long discussion about the usability of ENVI for microscopy data to be irrelevant to the acceptability of the paper for publication.”

- Pages 13-14- this document.

As we stated in our previous response to this question, “Given these motivations, we are quite confused about this comment from the reviewer. “

- Page 34 - `Second_Response_to_Referees_Letters_2019-03-30`

Reviewers' comments:

Reviewer #4 (Remarks to the Author):

I have been asked to provide feedback on the overall package of manuscript, supplementary information and responses to reviewers for this manuscript.

Overall, I found the manuscript surprisingly poorly written. Given the long discussion in the reviews, I was surprised that there wasn't a simple explanation such as that provided in p4/5 of the most recent review, highlighting the exact steps required and advance provided by SEER. Figure 1 is not particularly illustrative. The approach and benefit of SEER feels somewhat buried until you reach Figures 2 and 3, and even then is not clearly written. There has been a long discussion about the relationship of the method to Fourier decomposition; authors have just added the phrase "denoised" in various places rather than explicitly giving details of how / why denoising is achieved, again, not helping the reader in their understanding. Arguments about improved speed are also obscure, because speed data is not presented for the individual data sets, only summarised finally in the discussion. These issues with presentation of the manuscript have probably led to several of the points that the reviewer has raised, and may also have arisen during the multiple rounds of review.

I was asked to comment only on the technical aspects under debate, so that I will do now (with the above given for context) with some recommendations that I think would help see the paper through to an acceptance decision.

Having "lightly" reviewed the manuscript, I do find several of the points that the reviewer is making well justified, namely: a) the question of image contrast vs spectral separation accuracy as a metric of relevance for defining a visualisation and b) the question of the Fourier decomposition and how clearly the steps of the method are explained. While the manuscript does not, in my view, appear to be technically incorrect, it does make the reader work very hard to find the information they need (see commentary in above paragraph).

For a) the authors are presenting a method for improved visualisation of hyperspectral data. This implies that the contrast provided by SEER images should exceed that provided by alternative visualisations, rather than that the spectral accuracy be higher. As the authors themselves note, there are many methods available for quantification of hyperspectral data but they wanted to develop a method that would enable "quick" visualisation for users of hyperspectral microscopes. Statements are made in that indicate "improved contrast" but this metric is not quantified. The authors should really provide a quantitative comparison of the image contrast achieved for the visualisations given in the main text of the paper; this would be expected in a medical imaging publication reporting enhanced visualisation.

For b), I find the various visualisation modes from the phasor map relatively well described but the exact steps taken to reach the SEER visualisation rather opaque. The closest description is that given on p4/5 of the response to reviewers, but even then the steps are not precisely specified. As this seems like quite a crucial point of contention, I think the authors should be encouraged to write up-front at either the end of the introduction or beginning of results when they are introducing SEER, the steps that they take (cf p4/5) but importantly, precisely how they take them e.g. 3. Denoising using XX method to generate YY improvement in the data. To my view this is necessary in a paper reporting a new method, as readers should be able to simply understand the steps involved reproduce the method themselves.

Also, in a new method that seeks to reduce computation speed, all data presented in visualisation should really state the computation speed of the new and existing methods.

Finally, I was asked to comment on benchmarking in relation to ICA. Of course, appropriate benchmarking is an important topic during the introduction of a new method. The benchmarking method chosen should take account of the signal, noise and size of the data set, among other parameters. ICA requires statistical independence of components, and in the case of noisy data may display a non-linear relationship between the components and the original data. ICA can be

notoriously difficult to apply in biomedical imaging hyperspectral data at low signal-to-noise, which may make it challenging to benchmark against it.

Reviewers' comments:

Reviewer #4 (Remarks to the Author):

I have been asked to provide feedback on the overall package of manuscript, supplementary information and responses to reviewers for this manuscript.

Overall, I found the manuscript surprisingly poorly written. Given the long discussion in the reviews, I was surprised that there wasn't a simple explanation such as that provided in p4/5 of the most recent review, highlighting the exact steps required and advance provided by SEER. Figure 1 is not particularly illustrative. The approach and benefit of SEER feels somewhat buried until you reach Figures 2 and 3, and even then is not clearly written. There has been a long discussion about the relationship of the method to Fourier decomposition; authors have just added the phrase "denoised" in various places rather than explicitly giving details of how / why denoising is achieved, again, not helping the reader in their understanding. Arguments about improved speed are also obscure, because speed data is not presented for the individual data sets, only summarised finally in the discussion. These issues with presentation of the manuscript have probably led to several of the points that the reviewer has raised, and may also have arisen during the multiple rounds of review.

I was asked to comment only on the technical aspects under debate, so that I will do now (with the above given for context) with some recommendations that I think would help see the paper through to an acceptance decision.

We thank the reviewer for the constructive comments. We answered the specific points raised by this reviewer in line.

Having "lightly" reviewed the manuscript, I do find several of the points that the reviewer is making well justified, namely: a) the question of image contrast vs spectral separation accuracy as a metric of relevance for defining a visualisation and b) the question of the Fourier decomposition and how clearly the steps of the method are explained. While the manuscript does not, in my view, appear to be technically incorrect, it does make the reader work very hard to find the information they need (see commentary in above paragraph).

For a) the authors are presenting a method for improved visualisation of hyperspectral data. This implies that the contrast provided by SEER images should exceed that provided by alternative visualisations, rather than that the spectral accuracy be higher. As the authors themselves note, there are many methods available for quantification of hyperspectral data but they wanted to develop a method that would enable "quick" visualisation for users of hyperspectral microscopes. Statements are made in that indicate "improved contrast" but this metric is not quantified. The authors should really provide a quantitative comparison of the image contrast

achieved for the visualisations given in the main text of the paper; this would be expected in a medical imaging publication reporting enhanced visualisation.

We have added a quantitative comparison of the contrast achieved for the visualizations given in the main text of the paper. Due to the color nature of the images, we included in the comparison colorfulness, sharpness, contrast and an established measure of the overall color quality enhancement.

The results are reported in the text in a new paragraph on quantification of image colorfulness, contrast and sharpness and overall image quality. The text now reads:

“Quantification of RGB images by colorfulness, contrast and sharpness show that SEER generally performs better than standard visualization approaches (Supplementary Figure 25, Methods). SEER’s average enhancement was 2%-19% for colorfulness, 11%-27% for sharpness and 2%-8% for contrast (Supplementary Table 3) for the datasets of Figures 4-7. We then performed a measure of the Color Quality Enhancement (CQE)⁵⁸, a metric of the human visual perception of color image quality, (Supplementary Table 4). The CQE score of SEER was higher than the standard, with improvement of 11%-26% for Figure 4, 7%-98% for Figure 5, 14%-25% for Figure 6 and 12%-15% for Figure 7 (Supplementary Figure 25, Supplementary Note 3, Methods). “

We edited the methods section to define how colorfulness, sharpness, contrast and color quality enhancement were performed:

“Color image quality calculations

Colorfulness

Due to the inherent lack of “ground truth” in experimental fluorescence microscopy images, we utilized an established model for calculating the color quality of an image without a reference⁶³. The colorfulness is one of three parameters, together with sharpness and contrast, utilized by Panetta et al⁵⁸ to quantify the overall quality of a color image. Two opponent color spaces are defined as:

$$\alpha = R - G \quad (\text{eq. 25})$$

$$\beta = 0.5 (R + G) - B \quad (\text{eq. 26})$$

Where R, G, B are the red, green and blue channels respectively, α and β are red-green and yellow-blue spaces. The colorfulness utilized here is defined as:

$$Colorfulness = 0.02 \log \left(\frac{\sigma_{\alpha}^2}{|\mu_{\alpha}|^{0.2}} \right) \log \left(\frac{\sigma_{\beta}^2}{|\mu_{\beta}|^{0.2}} \right) \quad (\text{eq. 27})$$

With σ_{α}^2 , σ_{β}^2 , μ_{α} , μ_{β} respectively as the variances and mean values of the α and β spaces⁵⁸.

Sharpness

We utilize EME⁶⁴, a Weber based measure of enhancement. EME is defined as follows:

$$EME_{sharp} = \frac{2}{k_1 k_2} \sum_{l=1}^{k_1} \sum_{l=1}^{k_2} \log \left(\frac{I_{max,k,l}}{I_{min,k,l}} \right) \quad (\text{eq. 28})$$

Where k_1 , k_2 are the blocks used to divide the image and $I_{max,k,l}$ and $I_{min,k,l}$ are the maximum and minimum intensities in the blocks. EME has been shown to correlate with a human observation of sharpness in color images⁵⁸ when associated with a weight λ_c for each color component.

$$Sharpness = \sum_{c=1}^3 \lambda_c EME_{sharp} \quad (\text{eq. 29})$$

Where the weights for different color components used in this article are $\lambda_R = 0.299$, $\lambda_G = 0.587$, $\lambda_B = 0.114$ in accordance with NTSC standard and values reported in literature⁵⁸.

Contrast

We utilize Michelson-Law measure of enhancement AME⁶⁵, an effective evaluation tool for contrast in grayscale images, designed to provide larger metric value for larger contrast images. AME is defined as:

$$AME_{contrast} = \frac{1}{k_1 k_2} \sum_{l=1}^{k_1} \left(\sum_{l=1}^{k_2} \log \left(\frac{I_{max,k,l} + I_{min,k,l}}{I_{max,k,l} - I_{min,k,l}} \right) \right)^{-0.5} \quad (\text{eq. 30})$$

Where k_1 , k_2 are the blocks used to divide the image and $I_{max,k,l}$ and $I_{min,k,l}$ are the maximum and minimum intensities in the blocks. The value of contrast for color images was then calculated as:

$$Contrast = \sum_{c=1}^3 \lambda_c AME_{contrast} \quad (\text{eq. 31})$$

With the same weights λ_c utilized for sharpness.

Color Quality Enhancement

We utilize Color Quality Enhancement (CQE) a polling method to combine colorfulness, sharpness and contrast into a value that has both strong correlation and linear correspondence with human visual perception of quality in color images⁵⁸. CQE is calculated as:

$$CQE = c_1 \text{colorfulness} + c_2 \text{sharpness} + c_3 \text{contrast} \quad (\text{eq. 31})$$

Where the linear combination coefficients for CQE measure were set to evaluate contrast change according to values reported in literature⁵⁸, $c_1 = 0.4358$, $c_2 = 0.1722$ and $c_3 = 0.3920$. “

Values for each figure are reported in Supp. Figure 25:

“**Supplementary Figure 25. Quantification of enhancement for Figures 4-7.** The scores of (a) colorfulness, (b) contrast, (c) sharpness and (d) Color Quality Enhancement (CQE) are calculated according to Methods section for multiple visualization strategies. Average values are reported in Supplementary Table 2 and 3. (a) Colorfulness values for SEER were generally higher than other approaches, made exception for Figure 7 Peak Wavelength visualization (reported in Supplementary Figure 19d), owing to a very low average intensity in the red channel ($\langle I_R \rangle = 840$), and an almost double average green to blue intensity ($\langle I_G \rangle / \langle I_B \rangle = 1.7$), which makes the β parameter used in colorfulness small in average and the denominator of the second logarithm in the colorfulness equation (Methods) approximately equal to 1, producing a factor of 10 larger than usual ratio of variance β to average β . This combination of intensities results in a colorfulness 1.03-fold higher than the SEER h=2, however in this case the value of colorfulness does not correspond to human observation (Supplementary Figure 19d) suggesting this score could be an outlier due to a special combination of intensities. The values of (b)

contrast, (c) Sharpness show higher performance for SEER. (d) The CQE score of SEER was higher than the standard, with improvement of 11%-26% for Figure 4, 7%-98% for Figure 5, 14%-25% for Figure 6 and 12%-15% for Figure 7.”

We report the actual numbers and average values in Supplementary Tables 3 and 4:

Supplementary Table 3: Average colorfulness, contrast and sharpness score across figures 4-7 for different visualization methods

	Average Score		
	Colorfulness	Contrast	Sharpness
Gauss. Def.	2.11	53.84	10.83
Gauss r=.1	2.06	52.19	10.60
Gauss r=.2	1.97	59.57	11.17
Gauss r=.3	1.96	60.00	11.20
Peak Wav.	2.29	58.61	11.14
SEER h=1	2.34	66.32	11.40
SEER h=2	2.34	65.15	11.40

Supplementary Table 4: Color Quality Enhancement score for datasets in figures 4-7. Parameters calculations are reported in methods section.

	Color Quality Enhancement			
	Figure 4	Figure 5	Figure 6	Figure 7
Gauss. Def.	39.47	22.72	58.37	46.64
Gauss r=.1	38.26	16.39	61.29	46.34
Gauss r=.2	43.55	30.11	62.53	46.71
Gauss r=.3	43.07	30.26	63.89	46.89
Peak Wav.	43.15	29.09	61.70	47.27
SEER h=1	45.72	31.08	72.87	53.18
SEER h=2	48.38	32.37	69.33	49.58

We added a Supplementary Note 3 discussing the calculation of contrast and quality of color images.

“Supplementary Note 3

Measuring color contrast in fluorescent images

There is an inherent difficulty in determining an objective method to measure the image quality of color images in relation to fluorescent images. The main challenge for fluorescent images is that for the majority of fluorescence microscopy experiments, a reference image does not exist because there is an inherent uncertainty related to the image acquisition. Therefore, any kind of color image quality assessment will need to be based solely on the distribution of colors within an image.

This type of assessment has its own further challenges. Although there have been a variety of quantitative methods formulated to determine the quality of intensity distributions in grayscale images, such methods for color images are still being debated and tested⁹⁻¹². This lack of suitable methods for color images mainly comes from the divide between the mathematical representation of the composition of different colors and human perception of those same colors. This divide occurs because human color perception varies widely and is nonlinear for different colors; whereas the quantitative representation of any color is usually a linear combination of base colors such as Red, Green, and Blue. This nonlinear human perception of color is closely related to the concept of hue. Loosely speaking, hue is the dominant wavelength that reflects the light. Hues perceived as blue tend to reflect light at the left end of the spectrum and red at the right end of the spectrum. Generally, each individual color has a unique holistic trait which is determined by its distinctive spectrum. Discretization of the spectrum into multiple components cannot fully describe the original richness in color.

The current methods for determining the quality of an RGB image usually adapt grayscale methods in two different ways. The first method involves either converting the three channel color image into a single channel grayscale image before measuring the quality. The second method measures the quality of each channel individually and then combines those measurements with different weights. However, both methods face limitations in providing a value of quality that correlates well with human perception. The first method loses information when converting the color image to grayscale. The second method tries to interpret the nonlinear human perception of the quality of a color image by separating it into three channels and measuring them individually. The inherent hue of a color, however, is more than the sum of the individual component colors, since each channel taken individually is not necessarily as colorful as the combined color. A more complete color metric should take hue into account, such as by measuring the colorfulness loss between the original and processed images¹⁰. In conclusion, as a consequence of this limitation in measuring colorfulness for current methods, there is currently no established “true measure of contrast” within fluorescent color images. “

For b), I find the various visualisation modes from the phasor map relatively well described but the exact steps taken to reach the SEER visualisation rather opaque. The closest description is that given on p4/5 of the response to reviewers, but even then the steps are not precisely specified. As this seems like quite a crucial point of contention, I think the authors should be encouraged to write up-front at either the end of the introduction or beginning of results when they are introducing SEER, the steps that they take (cf p4/5) but importantly, precisely how they take them e.g. 3. Denoising using XX method to generate YY improvement in the data. To my view this is necessary in a paper reporting a new method, as readers should be able to simply understand the steps involved reproduce the method themselves.

We edited Figure 1 and its caption to provide a more thorough description of the steps involved in SEER.

“Figure 1. Spectrally Encoded Enhanced Representations (SEER) conceptual representation. (a) A multispectral fluorescent dataset is acquired using a confocal instrument in spectral mode (32-channels). Here we show a *Tg(ubi:Zebrawow)*³⁴ dataset where cells contain a stochastic combination of cyan, yellow and red fluorescent proteins. (b) Average spectra within six regions of interest (colored boxes in a) show the level of overlap resulting in the sample. (c) Standard multispectral visualization approaches have limited contrast for spectrally similar fluorescence. (d) Spectra for each voxel within the dataset are represented as a two-dimensional histogram of their Sine and Cosine Fourier coefficients S and G, known as the phasor plot. (e) Spatially lossless spectral denoising is performed in phasor space to improve signal²⁹. (f) SEER provides a choice of several color reference maps that encode positions on the phasor into predetermined color palettes. The

reference map used here (magenta selection) is designed to enhance smaller spectral differences in the dataset. (g) Multiple contrast modalities allow for improved visualization of data based on the phasor spectra distribution, focusing the reference map on the most frequent spectrum, on the statistical spectral center of mass of the data (magenta selection), or scaling the map to the distribution. (h) Color is assigned to the image utilizing the chosen SEER reference map and contrast modality. (i) Nearly indistinguishable spectra are depicted with improved contrast, while more separated spectra are still rendered distinctly. “

We deeply reworked the initial portion of the results, introducing the same scheme we presented in previous response pages 4-5.

“SEER was designed to create usable spectral contrast within the image by accomplishing five main steps. First, the Sine and Cosine Fourier transforms of the spectral dataset at one harmonic (usually 1st or 2nd owing to Riemann surfaces) provide the components for a 2D phasor plot (Figure 1D). The phasor transformation compresses and normalizes the image information, reducing a multi-dimensional dataset into a 2D-histogram representation and normalizing it to the unit circle.

Second, the histogram representation of the phasor plot provides insight on the spectral population distribution and improvement of the signal through summation of spectra in the histogram bins. Pixels with very similar spectral features, for example expressing only a single fluorophore, will fall within the same bin in the phasor plot histogram. Because of the linear property of the phasor transform, if an image pixel contains a mixture of two fluorophores, its position on the phasor plot will lie proportionally along the line connecting the phasor coordinates of those two components. This step highlights importance of geometry and distribution of bins in the phasor representation.

Third, spatially lossless spectral denoising, previously presented²⁹, is performed 1-2 times in phasor space to reduce spectral error. In short, median filters are applied on both the Sine and Cosine transformed images, reducing the spectral scatter error on the phasor plot, while maintaining the coordinates of the spectra in the original image (Figure 1E). Filters affect only the phasor space, producing an improvement of the signal.

Fourth, we designed multiple SEER maps exploiting the geometry of phasors. For each bin, we assign RGB colors based on the phasor position in combination with a reference map (Figure 1F). Subtle spectral variations can be further enhanced with multiple contrast modalities, focusing the map on the most frequent spectrum, the

statistical center of mass of the distribution or scaling the colors to the extremes of the phasor distribution (Figure 1G).

Finally, colors in the original dataset are remapped based on SEER results (Figure 1H). This permits a dataset in which spectra are visually indistinguishable (Figure 1A-C) to be rendered so that even these subtle spectral differences become readily discernible (Figure 1I). SEER rapidly produces 3 channel color images (Supplementary Figure 1) that approximate the visualization resulting from a more complete spectral unmixing analysis (Supplementary Figure 2)."

Also, in a new method that seeks to reduce computation speed, all data presented in visualisation should really state the computation speed of the new and existing methods.

We added timing of SEER for the images in the main manuscript. The results are summarized in the main text and are in accordance with the processing time we previously reported in Supplementary Figure 1:

"Processing speed comparison between SEER and fastICA for the multispectral fluorescent data shown in Figures 4-7 is presented in Supplementary Table 2. SEER's computation time ranged between 0.44 (for Figure 4) and 6.27 seconds (for Figure 5) where the corresponding timing for fastICA was 3.45 and 256.86 seconds respectively, with a speed up in the range of 7.9 to 41 folds (Supplementary Figure 20), in accordance with the trend shown in Supplementary Figure 1. "

Values are reported in Supplementary Table 2 and Supplementary Figure 20.

"Supplementary Table 2: Processing time comparison SEER vs Independent Component Analysis (scikit-learn implementation) for Figures 4-7."

	Processing Time		
	SEER [sec]	ICA (3c) [sec]	Speed Up [folds]
Figure 4	0.44	3.45	7.9
Figure 5	6.27	256.86	41.0
Figure 5 subset	2.89	77.82	26.9
Figure 6	1.49	33.58	22.6
Figure 7	0.52	10.19	19.5

Supplementary Figure 20. Processing speed comparison SEER vs Independent

Component Analysis for the datasets of Figures 4-7. Here we compare the processing time between SEER and the FastICA submodule of the python module, scikit-learn. With the same measurement strategy used in Supplementary Figure 1, timers using the perf_counter function within the python module, time, were placed around specific functions corresponding to the calculations required for the creation of SEER maps in HySP and with FastICA. (a) Run time for SEER (magenta) was considerably lower than ICA (3 components) (cyan) in all Figures and their subsets. (b) The speed up was higher for larger z-stack spectral datasets (Figure 5, 41-fold improvement) and reduced for smaller, single spectral images (Figure 4, 7.9-fold improvement).

Numerical values for these plots are reported in Supplementary Table 2.

Finally, I was asked to comment on benchmarking in relation to ICA. Of course, appropriate benchmarking is an important topic during the introduction of a new method. The benchmarking method chosen should take account of the signal, noise and size of the data set, among other parameters. ICA requires statistical independence of components, and in the case of noisy data may display a non-linear relationship between the components and the original data. ICA can be notoriously difficult to apply in biomedical imaging hyperspectral data at low signal-to-noise, which may make it challenging to benchmark against it.

We thank the reviewer for confirming our point.

REVIEWERS' COMMENTS:

Reviewer #4 (Remarks to the Author):

The authors have appropriately and thoroughly addressed the key areas that I raised and I am therefore happy to see the manuscript published.

REVIEWERS' COMMENTS:

Reviewer #4 (Remarks to the Author):

The authors have appropriately and thoroughly addressed the key areas that I raised and I am therefore happy to see the manuscript published.

We thank the reviewer for her/his time reviewing and improving our work.